# FeDaL: Federated Dataset Learning for General Time Series Foundation Models

**Shengchao Chen, Guodong Long, Michael Blumenstein, Jing Jiang**
Australian AI Institute, Faculty of Engineering and IT, University of Technology Sydney
`shengchao.chen.uts@gmail.com`
`{Guodong.Long,Michael.Blumenstein,Jing.Jiang}@uts.edu.au`

## Abstract

Dataset-level heterogeneity introduces significant domain biases that fundamentally degrade generalization on general Time Series Foundation Models (TSFMs), yet this challenge remains underexplored. This paper rethinks the from-scratch training of TSFMs using the paradigm of federated learning. We propose a novel Federated Dataset Learning (**FeDaL**) approach to tackle heterogeneous time series by learning dataset-agnostic temporal representations. Specifically, the distributed architecture of federated learning is a nature solution to decompose heterogeneous TS datasets into shared generalized knowledge and preserved personalized knowledge. Moreover, based on the TSFM architecture, FeDaL explicitly mitigates both local and global biases by adding two complementary mechanisms: Domain Bias Elimination (DBE) and Global Bias Elimination (GBE). FeDaL's cross-dataset generalization has been extensively evaluated in real-world datasets spanning eight tasks (including various regression and classification), against 54 baselines. We further analyze federated scaling behavior, showing how data volume, client count, and join rate affect model performance under decentralization. Our code is publicly available at https://github.com/shengchaochen82/FeDaL.

## 1 Introduction

Time series analysis plays critical roles in decision systems such as weather forecasting (Bi et al., 2023), medical symptom classification (Wang et al., 2024c), industrial anomaly detection (Xu et al., 2022), and imputing missing data in wearable sensors (Wu et al., 2020). Recent advances follow two main directions. Time Series Foundation Models (TSFMs) pretrain on large-scale, multi-domain datasets to capture transferable representations, achieving strong zero-shot or few-shot performance but remaining largely task-specific, for example in forecasting (Liu et al., 2024c; Shi et al., 2024; Ansari et al., 2024; Das et al., 2023) or classification (Feofanov et al., 2025). In contrast, Time Series Pattern Machines (TSPMs) pursue architecture-level unification across tasks (Zhou et al., 2023; Wu et al., 2022a; Wang et al., 2024a). However, TSPMs are typically trained per dataset, restricting zero-shot generalization across unseen domains. Moreover, similar to TSFMs, they often assume centralized access to diverse datasets, which is unrealistic in practice since time series data are usually siloed and heterogeneous (Chen et al., 2025). This gap motivates the need for a general TSFM that combines architecture-level flexibility with data-level generalization, without centralizing data.

Federated FMs (FFMs) (Zhuang et al., 2023) provide a promising decentralized paradigm for training FMs by leveraging both public and private datasets. This strategy mitigates privacy concerns and alleviates the computational burden of centralized training, and has been extended to train general-purpose models by treating each domain or dataset as an independent client. Despite this promise, existing works (Chen et al., 2025) face two major challenges. First, they adopt a coarse-grained treatment of heterogeneity, focusing only on broad domain-level shifts (e.g., climate vs. healthcare) while overlooking dataset-specific structural biases within each domain. Consequently, the lack of robust cross-domain invariance limits scalability, as downstream tasks still require fine-grained tuning to adapt to individual datasets, preventing these models from functioning as true FMs.

To make this concrete, we illustrate three representative dataset-level biases in **Figure 1**: **(1) Temporal resolution bias**, where sequences with different sampling rates encode inconsistent contexts

under a fixed window (e.g., 120 steps cover five days of hourly weather but only two hours of minute-level energy data); **(2) Physical constraint bias**, where unrelated physical laws (e.g., temperature variation vs. electric current) reduce cross-domain transferability; and **(3) Pattern transition bias**, where initially similar trends diverge due to exogenous events (e.g., traffic vs. website visits), breaking assumptions of shared temporal structures. These biases illustrate that heterogeneity arises at both the client and global levels: temporal resolution and physical constraint biases primarily distort client-specific representations, while pattern transition bias amplifies during global aggregation.

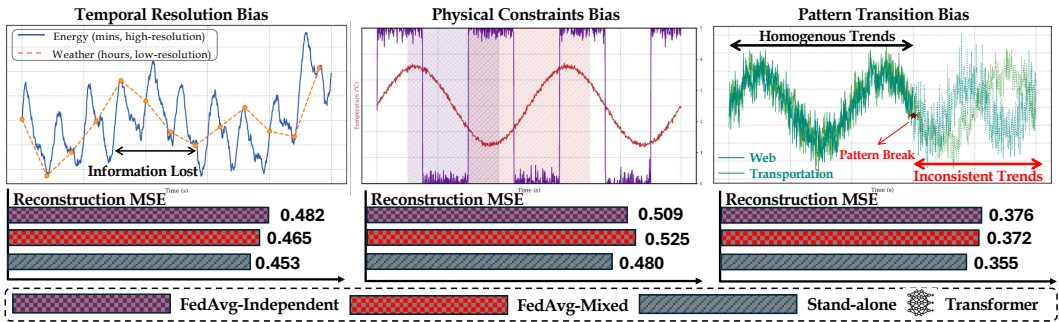

Figure 1: Dataset-level biases. **Information Lost**: Low-resolution sequences capture fewer details under a fixed window. **Pattern Break**: Abrupt structural changes across time. **Settings**: *FedAvg-Independent* (each dataset as a client), *FedAvg-Mixed* (datasets pooled then split), and *Stand-alone* (no aggregation). 'Transformer' means that results from the local Transformer model. Setup follow Sec. 4. Lower MSE is better.

This paper introduces **Federated Dataset Learning (FeDaL)**, a federated framework for training general TSFMs from scratch, designed to capture dataset-level patterns from heterogeneous datasets. FeDaL moves beyond coarse domain-level treatment to address dataset-level heterogeneity stemming from differences in sampling resolution, physical constraints, and transition patterns. Specifically, FeDaL integrates two complementary components: Domain Bias Elimination (DBE) and Global Bias Elimination (GBE), operating at the client and server levels respectively. DBE approximates and disentangles client-specific biases in a context-agnostic manner by modeling local bias representations. GBE improves global generalization by aligning client updates via gradient-level correction and applying fine-grained server-side tuning to mitigate residual biases during aggregation. DBE and GBE enable FeDaL to produce TSFMs with domain-invariant representations that generalize across diverse datasets while preserving privacy and scalability. Our contributions are:

1. We propose Federated Dataset Learning (FeDaL), a FL framework for training general TSFMs from scratch, explicitly addressing dataset heterogeneity through context-agnostic Domain Bias Elimination (DBE) on clients and Global Bias Elimination (GBE) at server.

2. We provide the first systematic study of TSFM pretraining scaling behaviors across various tasks under federated learning, offering empirical insights into how data volume, client population, and participation rate shape model generalization in decentralized settings.

3. Extensive experiments show that FeDaL-trained models outperforms advanced baselines across forecasting, imputation, anomaly detection, and classification, while achieving highly competitive performance against centralized TSFMs with far fewer parameters.

## 2 PRELIMINARIES

**Pre-trained Time Series Models.** Pretraining has become central to time series modeling, driving both task-specific models (Nie et al., 2022; Jin et al., 2023) and TSFMs (Goswami et al., 2024; Woo et al., 2024; Shi et al., 2024). Early efforts typically rely on masked reconstruction at the time-point (Zeng et al., 2023) or patch level (Liu et al., 2024c; Garza & Mergenthaler-Canseco, 2023), achieving strong zero-shot performance. More recent models such as Time-MoE (Shi et al., 2024), Moment (Goswami et al., 2024), Moirai (Woo et al., 2024), TimeFM (Das et al., 2024), and Chronos (Ansari et al., 2024) replace reconstruction with long-sequence prediction, further boosting zero-shot forecasting. However, these models remain task-oriented rather than universally applicable, unlike TSPMs that pursue architecture-level generality. In addition, they assume centralized

access to massive datasets, an unrealistic requirement since time series are often siloed and heterogeneous (Chen et al., 2025; Zhuang et al., 2023). Federated learning (FL) (McMahan et al., 2017) offers a practical alternative by enabling decentralized training with privacy preservation.

**Federated Foundation Models.** FFMs have emerged as a promising direction for scaling FMs without centralizing data (Zhuang et al., 2023). In TS, research on FFMs has begun to emerge. Time-FFM (Liu et al., 2024a) and PeFAD (Xu et al., 2022) leverages pretrained LLMs and applies lightweight tuning to achieve personalized predictions. In contrast to achieve multiple customized models, FFTS (Chen et al., 2025) takes a different approach by pretraining a TSFM on cross-domain datasets and then performing task-specific adaptation on downstream. It reduces local-global discrepancies to mitigate domain-wise heterogeneity. However, FFTS still falls short of being a true FM, since it don't support zero-shot inference and considers heterogeneity only at a coarse level, leaving dataset-level biases largely unresolved. To bridge this gap, we propose FeDaL, an FL framework TSFM training framework that explicitly addresses dataset-level heterogeneity and enables training a general TSFM from scratch with multiple downstream TS analysis task support.

**Problem Definition.** We consider an FFM setting with a server and $N$ clients, where each client $i$ holds a local time series dataset $\mathcal{D}_i$. These datasets are heterogeneous due to differences in sampling resolution, physical constraints, and temporal dynamics, which induce dataset-specific biases that may not transfer across clients. The objective is to collaboratively train a general TSFM that generalizes that supports diverse downstream tasks. Formally, the global objective is $F(\theta) = \arg\min_\theta \sum_{i=1}^N \frac{n_i}{n} F_i(\theta_i; \mathcal{D}_i)$, where $F_i(\cdot)$ denotes the local loss, and $n_i$, $n$ are the local and total sample counts. Our key question is: *Can train a general TSFM from scratch under FL constraints that captures dataset-invariant temporal pattern while mitigating dataset-induced biases?*

## 3 METHODOLOGY

Heterogeneous datasets introduce dataset-level biases, causing local models to overfit client-specific patterns that are not transferable across domains. For example, energy data may encode biases from grid cycles, while health data may reflect hospital-specific sampling protocols; such patterns are valid locally but break down in cross-domain settings. When aggregated, this misalignment produces a globally biased TSFM with limited generalization, a problem further amplified in large and structurally diverse datasets. To address this, we propose FeDaL (**Figure 2**) with two complementary mechanisms: Domain Bias Elimination (DBE) on clients and Global Bias Elimination (GBE) on server, where "domain" is dataset-level heterogeneity rather than broad application classes. FeDaL mitigate dataset-induced biases and align temporal patterns into a unified, transferable space.

### 3.1 DOMAIN BIAS ELIMINATION

DBE (as shown in **Figure 2**) addresses dataset-induced biases by decomposing latent representations of masked input patches into trend and seasonal components, from which a trainable bias vector is derived. This bias vector is injected back during reconstruction to suppress dataset-specific artifacts (see Right, **Figure 2**), while a regularization term aligns it with a global bias reference, encouraging the model to disentangle non-transferable local patterns from generalizable temporal structures.

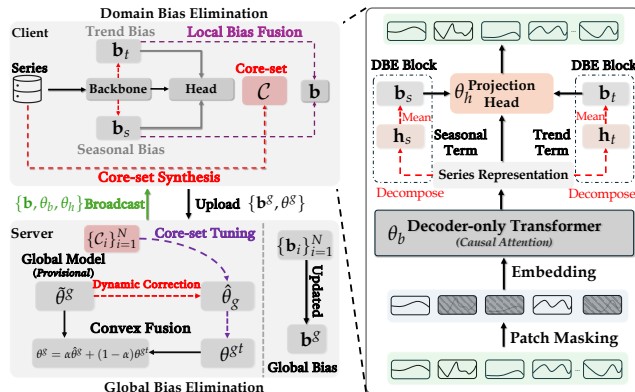

Figure 2: **FeDaL**: DBE reduces dataset-induced biases on the client side, whereas GBE improves alignment on the server. Locally, the plug-and-play DBE block (instead of DBE mechanism) captures trend and seasonal biases from latent representations.

Following (Nie et al., 2022), we adopt an unsupervised patch reconstruction for local updating. For clarity, the client index $i$ is omitted in the following derivations. Given a masked sequence $\tilde{X}$ obtained by patch-wise masking from original sequence $X$, its latent representation is obtained via the lo-

cal backbone $f_{\theta^b}(\cdot)$. We introduce a plug-and-play DBE block to explicitly approximate dataset-induced biases. Instead of directly averaging latent features, we apply decomposition (Wu et al., 2021) to split them into trend and seasonal components to capture context-independent patterns as:

$$\mathbf{h}_{i,t}, \mathbf{h}_{i,s} = \text{TimeDecomp}(f_{\theta^b}(\tilde{\boldsymbol{X}}), \tau), \tag{1}$$

where $\tau$ is the decomposition granularity, $\mathbf{h}_t$ denotes the trend component, and $\mathbf{h}_s$ the seasonal component. We then average each component to extract persistent deviations from compact representations, producing bias vectors that capture stable dataset-specific shifts. This process as:

$$\mathbf{b}_t = \text{Mean}(\mathbf{h}_t) \odot \boldsymbol{\gamma}_t, \quad \mathbf{b}_s = \text{Mean}(\mathbf{h}_s) \odot \boldsymbol{\gamma}_s. \tag{2}$$

Then we define the local bias vector $\mathbf{b} = \mathbf{b}_t + \mathbf{b}_s$. Compared with a plain mean of features, decomposition introduces an inductive bias: $\mathbf{b}_t$ captures low-frequency shifts, while $\mathbf{b}_s$ captures higher-frequency periodicities. Both are learnable, where the element-wise scaling factors $\boldsymbol{\gamma}_t$ and $\boldsymbol{\gamma}_s$ enable adaptation to evolving dataset patterns. To encourage disentanglement, the model reconstructs the input by injecting $\mathbf{b}$ into the latent features, the process can be formulated as:

$$\mathcal{L}(\theta) = \mathbb{E}[\|f_{\theta_h}(f_{\theta_b}(\tilde{\boldsymbol{X}}) + \mathbf{b}) - \boldsymbol{X}\|^2], \tag{3}$$

where $\theta = [\theta_b, \theta_h, \mathbf{b}]$. This design ensures that dataset-specific deviations are absorbed by $\mathbf{b}$, forcing $f_{\theta_b}$ to focus on transferable structure. Directly reconstructing $\mathbf{b}$ alone would not constrain the backbone, whereas this additive formulation encourages disentanglement between invariant features and dataset-bound biases. To align local biases across clients, we rewrite local optimization objective Eq. 3 via introducing an explicit penalty with controlling coefficient $\lambda$ as:

$$\mathcal{L}(\theta) = \mathbb{E}[\|f_{\theta_h}(f_{\theta_b}(\tilde{\boldsymbol{X}}) + \mathbf{b}) - \boldsymbol{X}\|^2] + \lambda \|\mathbf{b} - \mathbf{b}^g\|^2, \tag{4}$$

Here $\mathbf{b}^g$ is initialized from aggregated client statistics and serves as a dynamic global prior, preventing drift. Although the penalty is applied to $\mathbf{b}$, its effect propagates to the backbone by constraining the separation of spurious and transferable patterns. Since bias depend on full-dataset statistics but training proceeds on mini-batches, we adopt an exponential moving average (EMA) (Zhang et al., 2015) to stabilize updates by gradually blending the old estimates with new batch-wise estimates:

$$\mathbf{b}_t \leftarrow (1 - \mu)\mathbf{b}_t^{\text{old}} + \mu \mathbf{b}_t^{\text{new}}, \quad \mathbf{b}_s \leftarrow (1 - \mu)\mathbf{b}_s^{\text{old}} + \mu \mathbf{b}_s^{\text{new}}. \tag{5}$$

where $\mu$ is a smoothing factor. DBE can explicitly approximate dataset-specific biases, disentangle them from generalizable temporal patterns, and reduce inter-client divergence during aggregation.

## 3.2 GLOBAL BIAS ELIMINATION

While DBE alleviates dataset-induced biases at clients, residual misalignment persists during aggregation, leading to divergence in global pattens. To mitigate this, we propose GBE with two components: (1) Gradient-level Dynamic Correction, which compensates for client-server drifts, and (2) Core-set Tuning, which refines the global model with privacy-preserving client summaries. GBE reduce cross-client discrepancies and guide the global model toward domain-invariant patterns.

**Gradient-level Dynamic Correction.** The intuition is that naive aggregation (McMahan et al., 2017) assumes client updates are unbiased estimates of the global gradient. However, when client datasets are heterogeneous, this assumption fails, and the aggregated model drifts toward client-specific directions. Inspired by (Acar et al., 2021), we maintain a server-side state vector $\mathbf{s}_r$ that records the accumulated drift between client and global models across communication rounds:

$$\mathbf{s}^r = \mathbf{s}^{r-1} - \beta \sum_i (\theta_i^r - \theta_g^{r-1}), \tag{6}$$

where $\theta_r^i$ denotes the local model from client $i$ at round $r$, $\theta_{r-1}^g$ is the global model from the previous round, and $\beta$ is a scaling factor. This state serves as a correction memory: instead of directly adopting the naive FedAvg result $\tilde{\theta}_g^r$, we correct it as $\hat{\theta}_g^r = \tilde{\theta}_g^r - (1/\beta) \cdot \mathbf{s}^r$. Here, $\tilde{\theta}_g^r$ is the weighted average of client models, and $\hat{\theta}_r^g$ is the corrected global model used for the next round. By subtracting accumulated drifts, this mechanism prevents client-specific deviations (e.g., hospital-specific rhythms in healthcare or region-specific seasonality in energy data) from dominating the shared model.

**Core-set Tuning** Even with gradient correction, global bias may persist if clients encode different structural patterns. To refine the global model further, we perform server-side fine-tuning using compact core-sets generated by clients. Each client first samples a small batch $\tilde{\mathcal{X}} \subset \mathcal{X}$ of size $K \ll |\mathcal{X}|$ for efficiency and privacy. From this, we construct an initial core-set $\mathcal{C}_{\text{init}}$, parameterized as a set of learnable vectors rather than raw samples, so that it can be directly optimized via gradient descent. To approximate the client-specific pattern, the core-set is optimized by minimizing gradient discrepancies against the sampled batch, using the gradient matching (Killamsetty et al., 2021) as:

$$\mathcal{L}_{\text{match}} = \sum\nolimits_{x \in \tilde{\mathcal{X}}} ||\nabla_\theta f_\theta(\mathcal{C}_{\text{init}}) - \nabla_\theta f_\theta(x)||_2^2, \tag{7}$$

Here the loss is well-defined because gradients are aggregated at the model-parameter level; the optimization updates $\mathcal{C}_{\text{init}}$ rather than $\theta$ (model parameters) with learning rate $\eta$. To protect raw data, we perturb $\mathcal{C}_{\text{init}}$ in the Fourier domain. Specifically, we perturb only the amplitude while keeping the phase, since the phase encodes semantic temporal patterns (e.g., periodicity), whereas the amplitude carries fine-grained details as $\mathcal{C}' = \mathcal{F}^{-1}(\mathcal{F}(\mathcal{C}_{\text{init}}) + \epsilon \mathcal{N}(0,1))$, where $\mathcal{F}$ and $\mathcal{F}^{-1}$ denote the forward and inverse Fourier transforms, and $\epsilon$ controls the noise intensity. Perturbation may distort information, so we further align $\mathcal{C}'$ with the sampled batch $\tilde{\mathcal{X}}$ in latent space like Eq. 7:

$$\mathcal{L}_{\text{align}} = \sum\nolimits_{x \in \tilde{\mathcal{X}}} ||\nabla_\theta f_\theta(\mathcal{C}') - \nabla_\theta f_\theta(x)||_2^2, \tag{8}$$

The final aligned core-set $\mathcal{C}$ is uploaded to the server. The server fine-tunes the corrected global model $\hat{\theta}^{g,r}$ on aggregated $\mathcal{C}_i$ from clients, producing $\theta^{gt,r}$. To prevent catastrophic forgetting, we apply convex fusion as $\theta^{g,r} = \alpha \hat{\theta}^{g,r} + (1-\alpha)\theta^{gt,r}$, where $\alpha$ balances stability and adaptability.

**FedDaL.** As shown in Algorithm 1, Fed-DaL proceeds in three phases. In the initialization stage, the server broadcasts the model and establishes a global bias reference. In local updating, each client applies DBE to decompose features, estimate dataset-specific biases, and train the backbone with bias-regularized reconstruction, while also constructing a compact core-set for knowledge summarization. In the server aggregation stage, the server aggregates local models, applies dynamic correction strategy to counter client–server drifts, and refines the global model using compact core-sets. After each round, the server sends the updated model $\theta^g$ and the global bias $\mathbf{b}^g$ to selected clients, ensuring consistency across rounds. Through the interplay of DBE and GBE, our FedDaL progressively mitigates dataset-induced biases and aligns temporal patterns into a unified general TSFM.

---

**Algorithm 1** Workflow of Federated Dataset Learning

**Require:** Clients $\{c_i\}_{i=1}^N$, total rounds $R$, core-set size $K$, decomposition period $\tau$, hyperparameters $\lambda, \alpha, \epsilon$
**Ensure:** General TSFM $\theta^g = \{\theta_b, \theta_h\}$
    *Initialization Phase:*
    Server initializes $\theta^{g,0} = \{\theta_b^0, \theta_h^0\}$
    Warm-up 1 epoch, broadcasts global bias $\mathbf{b}^{g,0} = \sum_i \frac{n_i}{n} \mathbf{b}_i^0$
    Initialize server state $s = 0$
    **for** round $r = 1$ to $R$ **do**
        Sample client subset $\mathcal{S}_r$ and send $\{\theta_g^{r-1}, \mathbf{b}^{g,r-1}\}$
        **Local Update (each $c_i \in \mathcal{S}_r$ in parallel):**
        Decompose features into trend/seasonal terms (Eq. 1)
        Estimate local bias $\mathbf{b}_i^r = \text{Mean}(\mathbf{h}_{t,i}^r) + \text{Mean}(\mathbf{h}_{s,i}^r)$
        Train $\{\theta_{b,i}^r, \theta_{h,i}^r, \mathbf{b}_i^r\}$ by minimizing $\mathcal{L}_i$ (Eq. 4)
        Construct core-set $C_i^r$ (Eqs. 7, 8)
        Upload $\{\theta_{b,i}^r, \theta_{h,i}^r, C_i^r\}$ to the server
        **Server Aggregation:**
        Aggregate models to obtain provisional global $\tilde{\theta}^{g,r}$
        Update server state $\mathbf{s}^r$ (Eq. 6)
        Apply gradient correction: $\hat{\theta}^{g,r} = \tilde{\theta}^{g,r} - (1/\beta) \cdot \mathbf{s}^r$
        Refine $\hat{\theta}^{g,r} \to \theta^{gt,r}$ by fine-tuning on core-sets
        Fuse models: $\theta^{g,r} = \alpha \hat{\theta}^{g,r} + (1-\alpha)\theta^{gt,r}$
        Update global bias $\mathbf{b}^{g,r} = \frac{1}{|\mathcal{S}_r|} \sum_{i \in \mathcal{S}_r} \mathbf{b}_i^r$
    **end for**

---

## 4 MAIN RESULTS

In this section, we comprehensively evaluate FeDaL across three dimensions: federated representation learning, downstream generalization, and federated scaling behavior. Our experiments demonstrate that FeDaL (i) learns domain-agnostic representations in highly heterogeneous settings, (ii) enables strong generalization across forecasting, imputation, classification, and anomaly detection tasks, and (iii) exhibits favorable scaling properties under increasing data, client count and join ratio.

## 4.1 FEDERATED REPRESENTATION LEARNING

**Setup.** We evaluate FeDaL from an FL perspective, focusing on its ability to handle heterogeneous data. Experiments are conducted on two multi-domain datasets: UTSD (Liu et al., 2024c) with domain-mixed (DM) partitioning, where each client accesses data from multiple domains, and CTSD (Chen et al., 2025) with domain-independent (DI) partitioning, where each client is tied to a single domain. This allows us to examine FeDaL under both mixed-domain and domain-specific heterogeneity. We compare against five FL baselines: FedAvg (McMahan et al., 2017), FedProx (Li et al., 2020), FedPer (Arivazhagan et al., 2019), FedRep (Collins et al., 2021), and TSFM-oriented FFTS (Chen et al., 2025). All trained with patch-wise masked reconstruction (Nie et al., 2022) under 75% masking ratio with length of 512. Detailed implementation are provided in **Appendix B**.

**Main Results.** The main results are shown in **Table 1**, FeDaL consistently outperforms baselines across datasets and settings. Compared to the state-of-the-art federated TSFM pretraining method FFTS, FeDaL reduces reconstruction MSE by an average of **4.16%** on USTD and **8.86%** on CTSD. Notably, FeDaL adds about **2 MB** per round due to transmitting core-sets, yet achieves significantly stronger generalization. These demonstrate FeDaL's ability to balance client-specific pattern learning with server-side alignment, yielding more transferable temporal patterns.

**Bias Mitigation.** To further assess its bias-mitigation capacity, we visualize the evolution of local bias features from clients in **Figure 3**. At R1, the biases show substantial variation across clients, reflecting strong intra- and inter-client discrepancies. As training progresses, these discrepancies shrink, then toward a common space. This provides intuitive evidence that FeDaL effectively reduces dataset-induced biases and promotes alignment toward domain-invariant features.

Table 1: Federated TS representation learning results averaged over masking ratios $\{20\%, 35\%, 50\%, 75\%, 90\%\}$. For UTSD, two heterogeneity levels (H1 and H2) are simulated as described in **Appendix B**, while in CTSD each dataset is treated as an independent client. **Bold**: the best, Underline: the second best. † indicates evaluation with personalized models after server averaging, and ‡ denotes client-side evaluation without aggregation. Full results in Table 16.

| Method | USTD (DM) H1 | USTD (DM) H2 | CTSD (DI) | Comm. Param\# |
|---|---|---|---|---|
| FedAvg | 0.586 | 0.592 | 0.455 | 108.41 MB |
| FedProx | 0.583 | 0.586 | 0.444 | 108.41 MB |
| FedPer† | 0.565 | 0.588 | 0.430 | **106.43 MB** |
| FedRep† | 0.569 | 0.592 | 0.430 | **106.43 MB** |
| FFTS | 0.562 | 0.531 | 0.416 | 118.94 MB |
| Standalone‡ | 0.571 | 0.567 | 0.447 | – |
| **FeDaL (Ours)** | **0.551** | **0.511** | **0.387** | 110.41 MB |

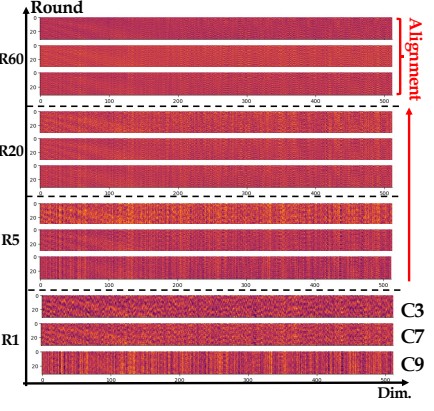

Figure 3: Local bias change across rounds (R1 - R60) for select clients (Clients 3, 7, 9).

**Ablation & Hyperparameter Sensitivity Results.** We analyze the contribution of five key components and the robustness of FeDaL to hyperparameter (**Table 2**). Ablations reveal that removing DBE or GBE significantly harms representation quality, confirming their role in addressing domain heterogeneity. Disabling DBE's alignment step (Eq. 4) further degrades performance, emphasizing the importance of bias alignment. Removing Core-set or Correction (Eq. 6) impairs global adaptation, showing their necessity for model refinement under heterogeneity. Sensitivity results (as shown in **Figure 4**) show: (i) overly large alignment weight $\lambda$ over-constrains local representations; (ii) larger core-sets bring marginal gains at the cost of privacy and communication; (iii) extreme fusion weights $\alpha$ weaken generalization; (iv) unstable $\beta$ values impair client-server representation blending.

Table 2: Ablation results. UTSD values are averaged over H1 and H2. "Vars." indicates the relative performance change compared with refees.

| Model | UTSD MSE | UTSD Vars. | CTSD MSE | CTSD Vars. | Average MSE | Average Vars. |
|---|---|---|---|---|---|---|
| **FeDaL (Original)** | **0.573** | – | **0.405** | – | **0.489** | – |
| *w/o* Alignment | 0.602 | ↓ 5.06% | 0.434 | ↓ 7.16% | 0.518 | ↓ 6.11% |
| *w/o* DBE | 0.637 | ↓ 11.17% | 0.452 | ↓ 7.16% | 0.545 | ↓ 9.17% |
| *w/o* Core-set Tuning | 0.590 | ↓ 2.97% | 0.430 | ↓ 6.17% | 0.510 | ↓ 4.57% |
| *w/o* Correction | 0.600 | ↓ 4.71% | 0.431 | ↓ 6.42% | 0.516 | ↓ 5.57% |
| *w/o* GBE | 0.610 | ↓ 6.46% | 0.444 | ↓ 9.63% | 0.527 | ↓ 8.05% |

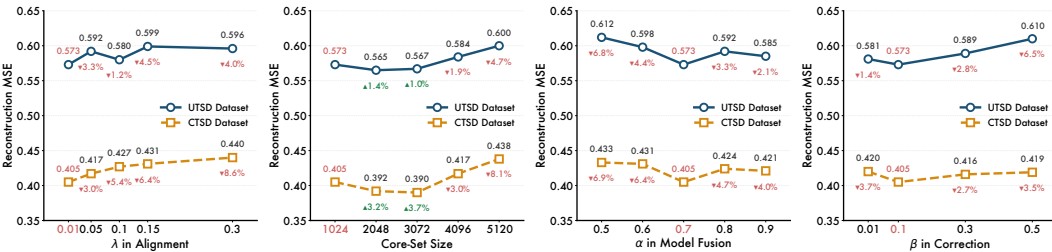

Figure 4: Hyperparameter sensitivity. Results on UTSD are averaged over H1 and H2. ↓: performance drop; ↑: improvement relative to the original FeDaL. *Best viewed in color and with zoom.*

## 4.2 ADAPTION AS GENERAL TIME SERIES FOUNDATION MODELS

To evaluate the generality of FeDaL-pretrained TSFMs, we test on diverse downstream tasks including long-/short-term deterministic forecasting, imputation, classification, and anomaly detection. Pretraining is conducted on LOTSA (Woo et al., 2024), which consists of 231B time points partitioned into 174 dataset-specific clients, naturally aligning with the federated setting. For non–zero-shot task (expect long-term forecasting), we fine-tune the trained model for only one epoch. Data preprocessing details are provided in **Appendix B**, and full results are included in **Appendix C**.

Table 3: Full-shot Long-term forecasting results (averaged across horizons $\{96, 192, 336, 720\}$ for ETT-series and Weather, and $\{24, 36, 48, 60\}$ for ILI). **Bold**: best; Underline: second best.

| Models | FeDaL | | FFTS | | FedAvg | | TimeMixer | | Time-LLM | | GPT4TS | | PatchTST | | TimesNet | | DLinear | | Fedformer | | Autoformer | | Stationary | | LightTS | |
|---|---|---|---|---|---|---|---|---|---|---|---|---|---|---|---|---|---|---|---|---|---|---|---|---|---|---|
| Metrics | MSE | MAE | MSE | MAE | MSE | MAE | MSE | MAE | MSE | MAE | MSE | MAE | MSE | MAE | MSE | MAE | MSE | MAE | MSE | MAE | MSE | MAE | MSE | MAE | MSE | MAE |
| ETTh1 | **0.380** | **0.409** | 0.391 | 0.412 | 0.412 | 0.431 | 0.448 | 0.443 | 0.408 | 0.423 | 0.465 | 0.455 | 0.413 | 0.430 | 0.458 | 0.450 | 0.422 | 0.437 | 0.440 | 0.460 | 0.496 | 0.487 | 0.570 | 0.537 | 0.491 | 0.479 |
| ETTh2 | 0.334 | **0.377** | 0.334 | 0.389 | 0.340 | 0.382 | 0.364 | 0.394 | 0.334 | 0.383 | 0.381 | 0.412 | **0.330** | 0.379 | 0.414 | 0.427 | 0.431 | 0.446 | 0.437 | 0.449 | 0.450 | 0.459 | 0.526 | 0.516 | 0.602 | 0.543 |
| ETTm1 | **0.319** | **0.365** | 0.323 | 0.374 | 0.333 | 0.367 | 0.381 | 0.395 | 0.329 | 0.372 | 0.388 | 0.400 | 0.406 | 0.406 | 0.357 | 0.378 | 0.448 | 0.452 | 0.588 | 0.517 | 0.481 | 0.456 | 0.435 | 0.437 | | |
| ETTm2 | 0.261 | 0.319 | 0.253 | 0.314 | 0.254 | 0.316 | 0.275 | 0.323 | **0.251** | **0.313** | 0.284 | 0.339 | 0.255 | 0.315 | 0.291 | 0.333 | 0.267 | 0.333 | 0.305 | 0.439 | 0.327 | 0.371 | 0.306 | 0.347 | 0.409 | 0.436 |
| Weather | **0.213** | **0.255** | 0.217 | 0.256 | 0.226 | 0.260 | 0.241 | 0.272 | 0.225 | 0.257 | 0.237 | 0.270 | 0.255 | 0.264 | 0.259 | 0.287 | 0.248 | 0.300 | 0.309 | 0.360 | 0.338 | 0.382 | 0.288 | 0.314 | 0.261 | 0.312 |
| ILI | **1.355** | **0.773** | 1.389 | 0.798 | 1.410 | 0.800 | 2.039 | 0.899 | 1.435 | 0.801 | 1.925 | 0.903 | 1.443 | 0.797 | 2.139 | 0.931 | 2.169 | 1.041 | 2.847 | 1.144 | 3.006 | 1.161 | 2.077 | 0.914 | 7.382 | 2.003 |
| **1st Count** | **9** | | 0 | | 0 | | 0 | | 2 | | 0 | | 1 | | 0 | | 0 | | 1 | | 2 | | 2 | | 2 | |

Table 4: Average zero-shot forecasting performance across horizons $\{96, 192, 336, 720\}$ for observation lengths $\{512, 1024, 2048, 3072\}$. **Bold**: best; Underline: second best. Full results in **Table 19, Appendix C**.

| Models | FL Pretraining | | | | | | Centralized Time Series Foundation Models | | | | | | | | | | | | | | | | |
|---|---|---|---|---|---|---|---|---|---|---|---|---|---|---|---|---|---|---|---|---|---|---|---|
| Metrics | FeDaL (Ours) | | FFTS | | FedAvg | | Moirai$_b$ | | Moirai$_l$ | | TimesFM | | Moment | | Chronos$_b$ | | Chronos$_l$ | | Time-MoE$_b$ | | Time-MoE$_l$ | | Time-MoE$_u$ | |
| Metrics | MSE | MAE | MSE | MAE | MSE | MAE | MSE | MAE | MSE | MAE | MSE | MAE | MSE | MAE | MSE | MAE | MSE | MAE | MSE | MAE | MSE | MAE | MSE | MAE |
| ETTh1 | 0.407 | 0.429 | 0.425 | 0.437 | 0.438 | 0.427 | 0.417 | **0.419** | 0.480 | 0.439 | 0.473 | 0.443 | 0.683 | 0.566 | 0.591 | 0.468 | 0.588 | 0.466 | 0.400 | 0.424 | **0.394** | **0.419** | 0.412 | 0.426 |
| ETTh2 | **0.361** | 0.382 | 0.370 | **0.374** | 0.390 | 0.401 | 0.362 | 0.382 | 0.367 | 0.377 | 0.392 | 0.406 | **0.361** | 0.409 | 0.405 | 0.410 | 0.455 | 0.427 | 0.366 | 0.404 | 0.405 | 0.415 | 0.371 | 0.399 |
| ETTm1 | 0.360 | 0.390 | 0.364 | 0.420 | 0.378 | 0.410 | 0.406 | 0.385 | 0.422 | 0.391 | 0.433 | 0.418 | 0.670 | 0.536 | 0.645 | 0.500 | 0.555 | 0.465 | 0.394 | 0.415 | 0.376 | 0.405 | **0.356** | 0.391 |
| ETTm2 | 0.292 | 0.341 | 0.317 | 0.362 | 0.322 | 0.365 | 0.311 | **0.337** | 0.329 | 0.343 | 0.328 | 0.343 | 0.316 | 0.365 | 0.310 | 0.350 | 0.295 | 0.338 | 0.317 | 0.365 | 0.316 | 0.361 | **0.288** | 0.344 |
| Weather | **0.255** | 0.284 | 0.262 | 0.300 | 0.277 | 0.305 | 0.287 | 0.281 | 0.264 | **0.273** | – | – | 0.294 | 0.326 | 0.292 | 0.315 | 0.279 | 0.306 | 0.265 | 0.297 | 0.270 | 0.300 | 0.256 | 0.288 |
| Avg. | **0.335** | 0.365 | 0.348 | 0.379 | 0.361 | 0.382 | 0.357 | **0.361** | 0.372 | 0.365 | 0.407 | 0.403 | 0.465 | 0.440 | 0.449 | 0.409 | 0.434 | 0.400 | 0.348 | 0.381 | 0.352 | 0.380 | 0.337 | 0.370 |
| **1st Count** | 3 | | 1 | | 0 | | **4** | | 1 | | 0 | | 1 | | 0 | | 1 | | 0 | | 0 | | 2 | |
| **2nd Count** | **4** | | 0 | | 0 | | 2 | | 2 | | 0 | | 0 | | 0 | | 1 | | 2 | | 2 | | 2 | |
| **Data Scale** | 231B | | 231B | | 231B | | 231B | | 231B | | 100B | | 1.13B | | 84B | | 84B | | 300B | | 300B | | 300B | |
| **Params#** | **28.42M** | | 31.18M | | **28.42M** | | 84M | | 91M | | 200M | | 385M | | 200M | | 710M | | 113M | | 453M | | 2.4B | |

**Long-Term Forecasting.** Time series forecasting remains a critical yet challenging task in practice. **(1) Full-shot.** We follow the setup from (Jin et al., 2023), evaluating on ETT, Weather, and Illness, excluding Traffic and ECL as they are used in pretraining. All models use a look-back window of 512, and we fine-tune the pretrained TSFM via FeDaL for five epochs. Results in **Table 3** show that FeDaL consistently outperforms both state-of-the-art deep models and LLM-based TSFMs in full-shot and few-shot scenarios, yielding significant MSE reductions. **(2) Zero-shot.** To further assess generalization, we perform zero-shot forecasting following the evaluation protocol of Time-MoE (Shi et al., 2024). As shown in **Table 4**, FeDaL achieves highly competitive zero-shot performance, surpassing FFTS by **3.8%** and Moirai$_{large}$ by **6.2%** in average MSE. Remarkably, FeDaL attains these gains with only **28M parameters**, far fewer than all centralized TSFMs such as Moirai (84M-91M), Chronos (200M-710M), and Time-MoE (113M-2.4B). This efficiency demon-

strates that our federated pretraining strategy FeDaL can deliver strong domain-agnostic temporal representations without relying on the massive model scales typical of centralized approaches.

**Short-Term Forecasting.** To evaluate the effectiveness of FeDaL-trained TSFM in short-term forecasting tasks, we conduct experiments on M4 dataset, following the protocols of GPT4TS (Zhou et al., 2023). As shown in **Table 5**, FeDaL significantly outperforms baselines. While it performs slightly below FFTS on SMAPE by a narrow margin of 0.07%, FeDaL reduces MASE by **2% to 38%** and achieves **5% to 22%** improvements in SMAPE and OWA. These further confirm the ability of FeDaL to learn cross-dataset representations that generalize effectively in forecasting.

Table 5: Average short-term forecasting results on M4 dataset. **Bold**: best; Underline: second best. None of these datasets were included in pretraining. * denotes a "former" suffix. Full results are provide in **Table 21**.

| | Model | FeDaL | FFTS | FedAvg | MOMENT | Time-LLM | GPT4TS | TimesNet | PatchTST | N-HiTS | N-BEATS | ETS.* | LightTS | DLinear | FED. | Stationary | Auto.* |
|---|---|---|---|---|---|---|---|---|---|---|---|---|---|---|---|---|---|
| Average | SMAPE | 11.412 | **11.404** | 12.342 | 14.593 | 11.983 | 12.69 | 12.88 | 12.059 | 12.035 | 12.25 | 14.718 | 13.525 | 13.639 | 13.16 | 12.78 | 12.909 |
| | MASE | **1.489** | 1.522 | 1.753 | 2.161 | 1.595 | 1.808 | 1.836 | 1.623 | 1.625 | 1.698 | 2.408 | 2.111 | 2.095 | 1.775 | 1.756 | 1.771 |
| | OWA | **0.818** | 0.831 | 0.926 | 1.103 | 0.859 | 0.94 | 0.955 | 0.869 | 0.869 | 0.896 | 1.172 | 1.051 | 1.051 | 0.949 | 0.93 | 0.939 |

**Imputation.** Imputation evaluates a model's ability to reconstruct missing values. We conduct experiments on five widely used datasets, including four ETT and Weather, where missingness is common. Following FFTS (Chen et al., 2025), we simulate varying corruption levels by randomly masking points. As shown in **Table 6**, FeDaL consistently outperforms all baselines. Compared to the strongest FL baselines, it reduces MSE by **12.64%** over FFTS and **27.62%** over FedAvg. Against centralized TSFMs, FeDaL achieves **22.84%** lower error than GPT4TS and also surpasses the general-purpose TSFM Moment, despite Moment being trained on a much larger centralized corpus. These confirm that FeDaL not only leverages distributed data effectively but also achieves competitive or superior generalization to centralized TSFMs under missing-data conditions.

Table 6: Average imputation performance for randomly masked time series, evaluated across four mask ratios $\{12.5\%, 25\%, 37.5\%, 50\%\}$. **Bold**: best; Underline: second best. Full results in **Table 23**.

| Model | FeDaL | | FFTS | | FedAvg | | Moment | | TimeMixer | | GPT4TS | | PatchTST | | TimesNet | | DLinear | | Fedformer | | Autoformer | | NS | | LightTS | |
|---|---|---|---|---|---|---|---|---|---|---|---|---|---|---|---|---|---|---|---|---|---|---|---|---|---|---|
| Metrics | MSE | MAE | MSE | MAE | MSE | MAE | MSE | MAE | MSE | MAE | MSE | MAE | MSE | MAE | MSE | MAE | MSE | MAE | MSE | MAE | MSE | MAE | MSE | MAE | MSE | MAE |
| ETTh1 | **0.022** | **0.090** | 0.024 | **0.090** | 0.034 | 0.120 | 0.024 | 0.094 | 0.036 | 0.123 | 0.028 | 0.105 | 0.047 | 0.140 | 0.120 | 0.253 | 0.027 | 0.107 | 0.104 | 0.218 | 0.093 | 0.206 | 0.062 | 0.177 | 0.051 | 0.150 |
| ETTh2 | 0.018 | **0.071** | **0.017** | 0.074 | 0.026 | 0.098 | 0.021 | 0.080 | 0.028 | 0.102 | 0.021 | 0.084 | 0.029 | 0.102 | 0.208 | 0.327 | 0.022 | 0.088 | 0.046 | 0.151 | 0.096 | 0.208 | 0.101 | 0.215 | 0.029 | 0.105 |
| ETTm1 | **0.054** | **0.147** | 0.058 | 0.160 | **0.054** | 0.154 | 0.062 | 0.167 | 0.073 | 0.192 | 0.069 | 0.173 | 0.115 | 0.224 | 0.202 | 0.329 | 0.078 | 0.187 | 0.284 | 0.373 | 0.201 | 0.306 | 0.117 | 0.246 | 0.103 | 0.214 |
| ETTm2 | **0.034** | **0.106** | 0.046 | 0.135 | 0.062 | 0.154 | 0.042 | 0.130 | 0.038 | 0.120 | 0.048 | 0.141 | 0.065 | 0.163 | 0.367 | 0.436 | 0.049 | 0.146 | 0.119 | 0.250 | 0.142 | 0.259 | 0.163 | 0.279 | 0.055 | 0.156 |
| Weather | **0.024** | **0.048** | 0.029 | 0.059 | 0.034 | 0.050 | 0.029 | 0.061 | 0.039 | 0.083 | 0.031 | 0.056 | 0.030 | 0.054 | 0.076 | 0.171 | 0.030 | 0.054 | 0.055 | 0.117 | 0.052 | 0.110 | 0.099 | 0.203 | 0.031 | 0.057 |
| $1^{st}$ Count | 9 | | 2 | | 1 | | 0 | | 0 | | 0 | | 0 | | 0 | | 0 | | 0 | | 0 | | 0 | | 0 | |

**Anomaly Detection.** We evaluate FeDaL-trained TSFM on five widely used multivariate datasets (SMD, MSL, SMAP, SwaT, PSM), following Moment (Goswami et al., 2024) protocols for fair comparison. As shown in **Table 7**, FeDaL achieves the best overall performance across all datasets. It surpasses strong centralized baselines such as ModernTCN and TSFM Moment by **2.40%** and **5.17%**, respectively, and further outperforms FL methods including FedAvg and FFTS by **0.96%** and **3.69%**. These results highlight our FeDaL's effectiveness in capturing global temporal invariance and its superior generalization to complex anomaly detection tasks.

Table 7: Anomaly detection results. We calculate the F1-score (%) for each dataset and statics the average F1-score. **Bold**: the best, Underline: the second best. * denotes a "former" suffix. Full results are in **Table 24**.

| Model | FeDaL | FFTS | FedAvg | Moment | GPT4TS | MTCN | TimesNet | FED.* | LightTS | DLinear | NS.* | Auto.* | Pyra.* | Anomaly | In.* | Re.* |
|---|---|---|---|---|---|---|---|---|---|---|---|---|---|---|---|---|
| SMD | 88.46 | 89.88 | 88.44 | 84.94 | 86.89 | 85.81 | 85.81 | 85.08 | 82.53 | 77.10 | 84.62 | 85.11 | 83.04 | 85.49 | 81.64 | 75.32 |
| MSL | **89.05** | 88.42 | 82.32 | 81.45 | 82.45 | 84.92 | 85.15 | 78.57 | 78.95 | 84.88 | 77.50 | 79.05 | 84.86 | 83.31 | 84.06 | 84.40 |
| SMAP | 71.70 | 71.38 | 70.78 | 69.43 | **72.88** | 71.26 | 71.52 | 70.76 | 69.21 | 69.26 | 71.09 | 71.12 | 71.09 | 71.18 | 69.92 | 70.40 |
| SwaT | **95.40** | 91.12 | 90.23 | 91.90 | 94.23 | 93.86 | 91.74 | 93.19 | 93.33 | 87.52 | 79.88 | 92.74 | 91.78 | 83.10 | 81.43 | 82.80 |
| PSM | **98.88** | 98.54 | 95.86 | 93.96 | 97.13 | 97.23 | 97.47 | 97.23 | 97.15 | 93.55 | 97.29 | 93.29 | 82.08 | 79.40 | 77.10 | 73.61 |
| $1^{st}$ Count | 3 | 0 | 0 | 0 | 1 | 0 | 0 | 0 | 0 | 0 | 0 | 0 | 0 | 0 | 0 | 0 |

**Classification.** Following the standard evaluation protocol from (Zhang et al., 2025a), We evaluate the FeDaL-trained TSFM on time series classification using 10 UEA (Bagnall et al., 2018) and

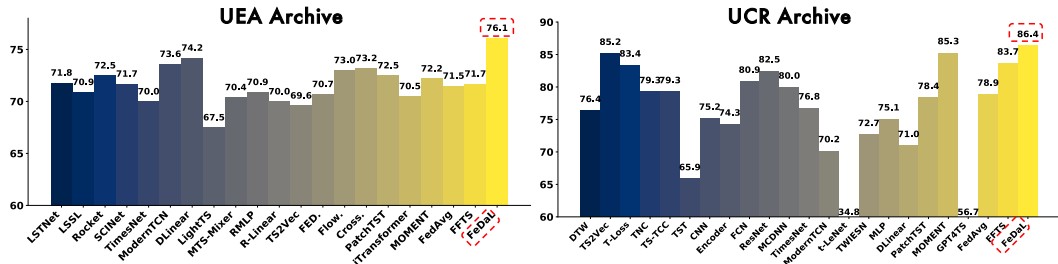

Figure 5: Classification results. Full results in **Table 25 & Table 26**. *Best viewed in color and with zoom.*

91 UCR (Dau et al., 2019) subsets spanning diverse domains. We adopt Linear Probing-based adaption method by attaching a linear classifier to the frozen TSFM, directly measuring the quality of learned representations. As shown in **Figure 5**, our FeDaL-trained TSFM consistently outperforms all baselines, including task-specific models, GPT4TS (fine-tuned), and Moment. Notably, FeDaL surpasses FL baselines such as FedAvg and FFTS by **6.4%/6.1%** on UEA and **9.5%/3.2%** on UCR, highlighting its ability to learn domain-invariant features and mitigate cross-client biases.

**Core-Set Analysis.** To better understand the effect of core-set tuning, we visualize the representation spaces of three key stages using t-SNE in **Figure 6**. The initial core-sets cluster closely around the original client data, indicating that they effectively approximate local distributions, but may also retain sensitive fine-grained patterns. After applying Fourier perturbation and semantic alignment, the refined core-sets move away from the raw data distribution while remaining semantically aligned. This suggests that our perturbation strategy can partially obfuscate fine-grained details and reduce direct data leakage risks, while preserving high-level temporal semantics that are useful for global alignment. As a result, the uploaded core-sets serve as compact, privacy-aware summaries that enable effective server-side tuning without requiring access to raw client data. More detailed analysis of the t-SNE visualizations for our proposed core-set tuning is provided in **Figure 8**.

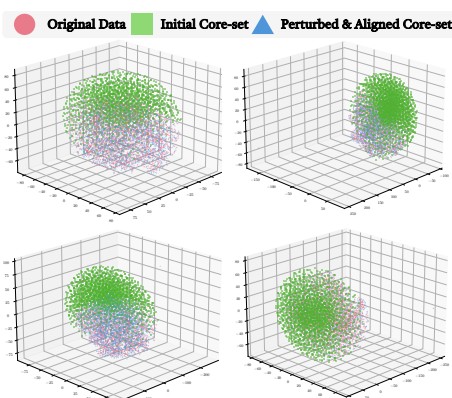

Figure 6: T-SNE visualization of Core-set.

### 4.3 FEDERATED SCALING BEHAVIORS

While prior work has examined scaling laws of centralized TSFMs (Yao et al., 2024), we investigate how federated pretraining scales with respect to data size, number of clients, and client participation rate. Specifically, we vary: (i) data size from 40B to 231B with fixed 174 clients; (ii) client count from 30 to 174 under fixed total data; and (iii) participation rate from 10% to 100%. As shown in **Figure 7**, larger data consistently improves performance, more clients yield better representations (even under fixed total data), and higher participation enhances aggregation and mitigates drift. These results indicate that federated TSFM pretraining benefits from scaling in data and client diversity, emphasizing coverage and participation over model size for improved generalization.

### 4.4 DISCUSSION ON LARGER TIME SERIES FORECASTING MODELS

To further contextualize performance, we compare our FeDaL-trained TSFM with Time-MoE (Shi et al., 2024), a recent large-scale TSFM designed specifically for forecasting. Time-MoE variants include models with up to 2.4B parameters trained on 300B time series data. We present results on five standard long-horizon forecasting benchmarks in **Table 20**, including each model's parameter count and training data size for visual comparison. Our findings show that FeDaL consistently outperforms Time-MoE$_{base}$ (113M) and Time-MoE$_{large}$ (453M), and achieves comparable results to the largest variant, Time-MoE$_{ultra}$ (2.4B). Notably, FeDaL reaches this level of performance with only

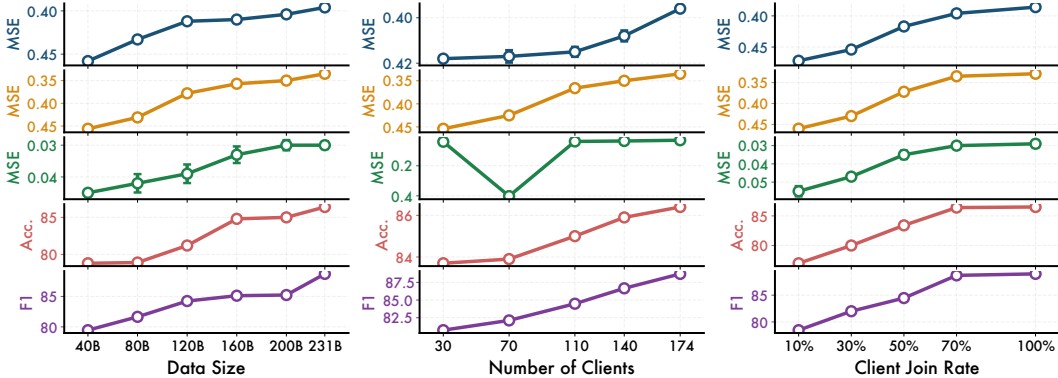

Figure 7: Scaling behaviors across tasks. Color codes: Blue – Avg. full/few-shot forecasting, Orange – Zero-shot forecasting, Green – Imputation, Red – Classification, Purple – Anomaly Detection. Y-axis for forecasting/imputation is inverted (lower is better). Full details and plots in **Appendix C.7**.

1.8% of the parameters and ∼70B fewer training samples, indicating substantially better efficiency. To quantify this tradeoff between performance and resource cost inspired by (Kaplan et al., 2020; Tan & Le, 2019), we define the Information Gain per Cost (IGC) metric as:

$$\text{IGC} = \frac{1}{\text{MSE} \times \text{Parameters Count}^{\alpha} \times \text{Training Data Size}^{\beta}}, \quad (9)$$

where $\alpha = \beta = 1$ by default. A higher IGC indicates a model training with better efficiency.

As shown in **Table 8**, our FeDaL achieves the highest IGC, outperforming all Time-MoE variants in terms of cost-effectiveness: **FeDaL > Time-MoE_base > Time-MoE_large > Time-MoE_ultra**. This underscores that FeDaL not only delivers strong performance, but does so with superior parameter and data efficiency, making it a more scalable and practical choice for real-world deployment. In addition, the TSFM trained with FeDaL demonstrates strong generalization across diverse tasks beyond forecasting, including classification, imputation, and anomaly detection.

Table 8: Zero-shot long-term forecasting performance comparison with Time-MoE. **Bold**: the best, Underline: the second best. Full results are provided in **Table 20**.

| Models | | FeDaL (Ours) | | Time-MoE_base | | Time-MoE_large | | Time-MoE_ultra | |
|---|---|---|---|---|---|---|---|---|---|
| **Metrics** | | **MSE** | **MAE** | **MSE** | **MAE** | **MSE** | **MAE** | **MSE** | **MAE** |
| ETTh1 | 96 | **0.347** | 0.381 | 0.357 | 0.381 | 0.350 | 0.382 | 0.349 | **0.379** |
| | 192 | 0.398 | 0.410 | **0.384** | **0.404** | 0.388 | 0.412 | 0.395 | 0.413 |
| | 336 | 0.425 | 0.452 | **0.411** | 0.434 | **0.411** | **0.430** | 0.447 | 0.453 |
| | 720 | 0.457 | 0.469 | 0.449 | 0.477 | **0.427** | **0.455** | 0.457 | 0.462 |
| | **Avg.** | 0.407 | 0.429 | 0.400 | 0.424 | **0.394** | **0.419** | 0.412 | 0.426 |
| ETTh2 | 96 | 0.307 | 0.355 | 0.305 | 0.359 | 0.302 | 0.354 | **0.292** | **0.352** |
| | 192 | 0.349 | **0.372** | 0.351 | 0.386 | 0.364 | 0.385 | **0.347** | 0.379 |
| | 336 | **0.387** | **0.395** | 0.391 | 0.418 | 0.417 | 0.425 | 0.406 | 0.419 |
| | 720 | **0.401** | **0.406** | 0.419 | 0.454 | 0.537 | 0.496 | 0.439 | 0.447 |
| | **Avg.** | **0.361** | **0.382** | 0.366 | 0.404 | 0.405 | 0.415 | 0.371 | 0.399 |
| ETTm1 | 96 | 0.289 | 0.346 | 0.338 | 0.368 | 0.309 | 0.557 | **0.281** | **0.341** |
| | 192 | 0.317 | 0.369 | 0.353 | 0.388 | 0.346 | 0.381 | **0.305** | **0.358** |
| | 336 | 0.370 | 0.420 | 0.381 | 0.413 | 0.373 | 0.408 | **0.369** | **0.395** |
| | 720 | **0.464** | **0.426** | 0.504 | 0.493 | 0.475 | 0.477 | 0.469 | 0.472 |
| | **Avg.** | 0.360 | **0.390** | 0.394 | 0.415 | 0.376 | 0.405 | **0.356** | 0.391 |
| ETTm2 | 96 | 0.207 | **0.283** | 0.201 | 0.291 | **0.197** | 0.286 | 0.198 | 0.288 |
| | 192 | 0.248 | 0.333 | 0.258 | 0.334 | 0.250 | 0.322 | **0.235** | **0.312** |
| | 336 | 0.316 | **0.340** | 0.324 | 0.373 | 0.337 | 0.375 | **0.293** | 0.348 |
| | 720 | **0.397** | **0.408** | 0.488 | 0.464 | 0.480 | 0.461 | 0.427 | 0.423 |
| | **Avg.** | 0.292 | **0.341** | 0.317 | 0.365 | 0.316 | 0.361 | **0.288** | 0.344 |
| Weather | 96 | 0.159 | 0.212 | 0.160 | 0.214 | 0.159 | 0.213 | **0.157** | **0.211** |
| | 192 | 0.217 | 0.264 | 0.210 | 0.260 | 0.215 | 0.266 | **0.208** | **0.256** |
| | 336 | 0.285 | 0.312 | 0.274 | 0.309 | 0.291 | 0.322 | **0.255** | **0.290** |
| | 720 | **0.359** | **0.348** | 0.418 | 0.405 | 0.415 | 0.400 | 0.405 | 0.397 |
| | **Avg.** | **0.255** | **0.284** | 0.265 | 0.297 | 0.270 | 0.300 | 0.256 | 0.288 |
| **Average** | | **0.335** | 0.370 | 0.343 | 0.382 | 0.355 | 0.387 | 0.342 | **0.369** |
| **Total Param.#** | | 28.42 M | | 113 M | | 453 M | | 2.4 B | |
| **Training Data** | | ∼ 231B | | 300B | | 300B | | 300B | |
| **IGC** | | $4.545 \times 10^{-19}$ | | $8.605 \times 10^{-20}$ | | $2.073 \times 10^{-20}$ | | $4.061 \times 10^{-21}$ | |

## 5 CONCLUSION

We introduced FeDaL, a federated learning framework for *training TSFMs from scratch* that explicitly targets dataset-level heterogeneity. By integrating DBE at the client side and GBE at server, FeDaL mitigates both local and global biases, enabling the learning of robust, domain-invariant temporal patterns. Extensive experiments across diverse time series analysis tasks (various regression and classification) demonstrate its superior cross-domain learning capabilities and show that FeDaL-trained models remain highly competitive against centralized TSFMs with far larger parameter scales.

## REPRODUCIBILITY STATEMENT

We pretrain on publicly available datasets UTSD (Liu et al., 2024c), CTSD (Chen et al., 2025), and LOTSA (Woo et al., 2024). Experiments are conducted on widely used benchmarks, including TSLib (Wu et al., 2022a) (ETT, Weather, and M4 for forecasting; ETT and Weather for imputation; SMD, MSL, SMAP, SwaT, and PSM for anomaly detection), as well as the UEA (Bagnall et al., 2018) and UCR (Dau et al., 2019) archives. All datasets are publicly accessible. To ensure reproducibility, we detail training setups, model configurations, and hyperparameter choices in Appendix B and Table 9. Furthermore, the code implementation of our proposed FeDaL is released at https://github.com/shengchaochen82/FeDaL.

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

# APPENDIX

This Appendix provides supplementary information and implementation details omitted from the main text, including:

- More Related Work (**Appendix A**): A comprehensive review of relevant literature, covering time series foundation models, FL for heterogeneous data, and foundation models.

- Implementation Details (**Appendix B**): Detailed descriptions of the techniques employed, training configurations, experimental setup, including benchmark procedures, data processing pipelines, and baseline methods.

- Full Results (**Appendix C**): Complete presentation of all results discussed in the main text.

## A  MORE RELATED WORK

**Pre-trained Time Series Models.**  Large Language Models (LLMs) have demonstrated strong capabilities such as zero-shot adaptation (Jin et al., 2023). Yet, due to the heterogeneity of time series, GPT-style TSFMs remain underexplored. Existing research can be grouped into two categories. The first leverages LLMs for time series analysis. For example, FPT (Zhou et al., 2023) uses GPT-2 as a representation extractor, while LLM4TS (Chang et al., 2023) encodes series into numerical tokens, showing scalability on forecasting tasks. Time-LLM (Jin et al., 2023) employs prompting techniques to enhance cross-modality reasoning, and UniTime (Liu et al., 2024b) introduces domain adaptation to reduce prediction bias. However, these approaches largely depend on the LLM backbone and cross-modal design. In contrast, our model is pre-trained natively on time series, avoiding extra modality alignment. The second line pre-trains directly on large-scale time series datasets. ForecastFPN (Dooley et al., 2024) trains on synthetic series for zero-shot forecasting, while CloudOps (Woo et al., 2023) applies masked modeling for domain-specific prediction. TimeGPT-1 (Garza & Mergenthaler-Canseco, 2023) represents the first commercial zero-shot API, and PreDcT (Das et al., 2023) shows strong results with a decoder-only Transformer. More recently, models such as MOMENT (Goswami et al., 2024), Moirai (Woo et al., 2024), TimeFM (Das et al., 2024), Chronos (Ansari et al., 2024), and Timer (Liu et al., 2024c) have been trained on ultra-large datasets and publicly released, achieving competitive cross-task performance. These efforts collectively highlight both the potential and the resource-intensive nature of training TSFMs.

**Federated Learning for Heterogeneous Data.**  FL (McMahan et al., 2017; Feng & Chen, 2025; Chen & Shu, 2025; Yan et al., 2025) enables decentralized model training without raw data sharing, but suffers from statistical heterogeneity (non-IID data), where distributional shifts across clients degrade performance. Early approaches reduce client–server divergence via alignment or regularization, yet typically assume mild heterogeneity and shared representation spaces, an assumptions that collapse in highly diverse domains. Personalized Federated Learning (PFL) extends FL by tailoring models to individual clients through regularization (Hanzely et al., 2020; Li et al., 2021a), partial parameter sharing (Li et al., 2021b; Collins et al., 2021), adaptive aggregation (Zhang et al., 2020), or meta-learning (Fallah et al., 2020). While effective for vision or text tasks where low-level features transfer well (Chen et al., 2025), PFL is less suitable for time series, where dataset-level disparities in resolution, semantics, and physical context are more severe. Moreover, PFL emphasizes personalization over unified generalization, conflicting with the goal of pretraining TSFMs, which require cross-domain invariance.

## B  IMPLEMENTATION DETAILS

**Decoder-only Transformer as Local Models**  We employ the Transformer as the model backbone due to its excellent scalability. Inspired by the significant advancements in Decoder-only LLMs capable of iterative generation (Garza & Mergenthaler-Canseco, 2023; Zhou et al., 2023), and recognizing the need for processing variable-length time series (Liu et al., 2024c; Zhang et al., 2025b), we adopt an auto-regressive approach for generative pre-training using standard Decoder-only Trans-

former architectures. The next-token prediction can be formulated as:

$$P(U) = \prod_{i=1}^{N} p(u_i \mid u_{<i}), \quad \text{where } U = \{u_1, u_2, ..., u_N\}. \tag{10}$$

For the tokenization of a given input time series $\boldsymbol{X}$, we employed a segment-wise tokenization strategy, representing $\boldsymbol{X}$ as $\{x_1, x_2, \ldots, x_{NS}\}$ with a unified context length $NS$. In this approach, a time series token is defined as a consecutive segment of length $S$, covering the series variations: $\mathbf{s}_i = \{x_{(i-1)S+1}, \ldots, x_{iS}\} \in \mathbb{R}^S$, where $i = 1, \ldots, N$. Subsequently, a time series segment is incorporated into the learnable position encoding, followed by the standard autoregressive Transformer update step. Utilizing the causal attention mechanism of the Decoder-only Transformer, the model autoregressively generates the subsequent segment $\mathbf{s}_{i+1}$ based on the previous segment $\mathbf{s}_i$. Consequently, generative pre-trained models are endowed with the flexibility to handle variable context lengths during inference and excel at multi-step generation through iterative sliding and enlarging of input tokens.

**Fourier-based Perturbation for Privacy Preservation**   To ensure privacy during core-set transmission, we introduce a Fourier-based perturbation mechanism that obfuscates sensitive raw temporal patterns while preserving high-level semantic structure. This builds on the observation that in time series, the phase of the Fourier transform captures semantic trends (e.g., shape, rhythm), while the amplitude encodes finer-grained details (e.g., scale, local fluctuations). Formally, given a core-set $\mathcal{C}$, we apply a discrete Fourier transform (DFT) to obtain its frequency-domain representation $\mathcal{F}(\mathcal{C}) = A + iP$, where $A$ and $P$ denote amplitude and phase, respectively. We then perturb only the amplitude component:

$$\hat{\mathcal{C}} = \mathcal{F}(\mathcal{C}_{\text{init}}) = A \odot e^{iP}, \tag{11}$$

where $A$ and $P$ denote amplitude and phase, and $\odot$ is the elementwise product. We then add Gaussian noise only to the amplitude:

$$\tilde{A} = A + \epsilon \, \Xi, \qquad \Xi \sim \mathcal{N}(0, I), \tag{12}$$

and reconstruct the perturbed core-set by

$$\mathcal{C}' = \mathcal{F}^{-1}\big(\tilde{A} \odot e^{iP}\big), \tag{13}$$

where $\epsilon > 0$ controls the perturbation strength and $\mathcal{F}^{-1}$ is the inverse DFT. To keep $\mathcal{C}'$ semantically faithful to the client data while remaining privacy-preserving, we align its latent representations to those of the sampled mini-batch $\tilde{\mathcal{X}}$ (both of size $K$) via

$$\mathcal{L}_{\text{align}}(\mathcal{C}') = \left\| \frac{1}{K} \sum_{c \in \mathcal{C}'} f_\theta(c) \;-\; \frac{1}{K} \sum_{x \in \tilde{\mathcal{X}}} f_\theta(x) \right\|_2^2, \tag{14}$$

where $f\theta(\cdot)$ denotes the encoder that outputs latent embeddings. The semantically aligned core-set $\mathcal{C}$, obtained by optimizing $\mathcal{L}_{\text{align}}$ w.r.t. $\mathcal{C}'$, is then uploaded to the server for core-set tuning.

**T-SNE Visualization on Core-set**   To further examine the semantic fidelity and privacy robustness of our core-set pipeline, we visualize the t-SNE embeddings of three data subsets for several clients: (i) the original local mini-batch $\tilde{\mathcal{X}}$ used for core-set construction, (ii) the gradient-matched core-set $\mathcal{C}$init, and (iii) the final perturbed and aligned core-set $\mathcal{C}'$ sent to the server. As shown in **Figure 8**, the initial core-set (green) largely overlaps with the original batch (pink), indicating that $\mathcal{C}$init effectively approximates local gradients but may still retain fine-grained information if shared directly. After applying Fourier-based perturbation and semantic alignment, the refined core-set (blue) becomes more separated from the raw data distribution while preserving overall cluster structure. This suggests that our perturbation mechanism reduces exposure of fine-grained data characteristics while maintaining high-level semantic consistency required for server-side alignment. Overall, the results indicate that the proposed pipeline achieves a practical utility–privacy trade-off without claiming formal privacy guarantees. We note that our perturbation mechanism mitigates but does not eliminate potential information leakage, and formal differential privacy guarantees remain future work.

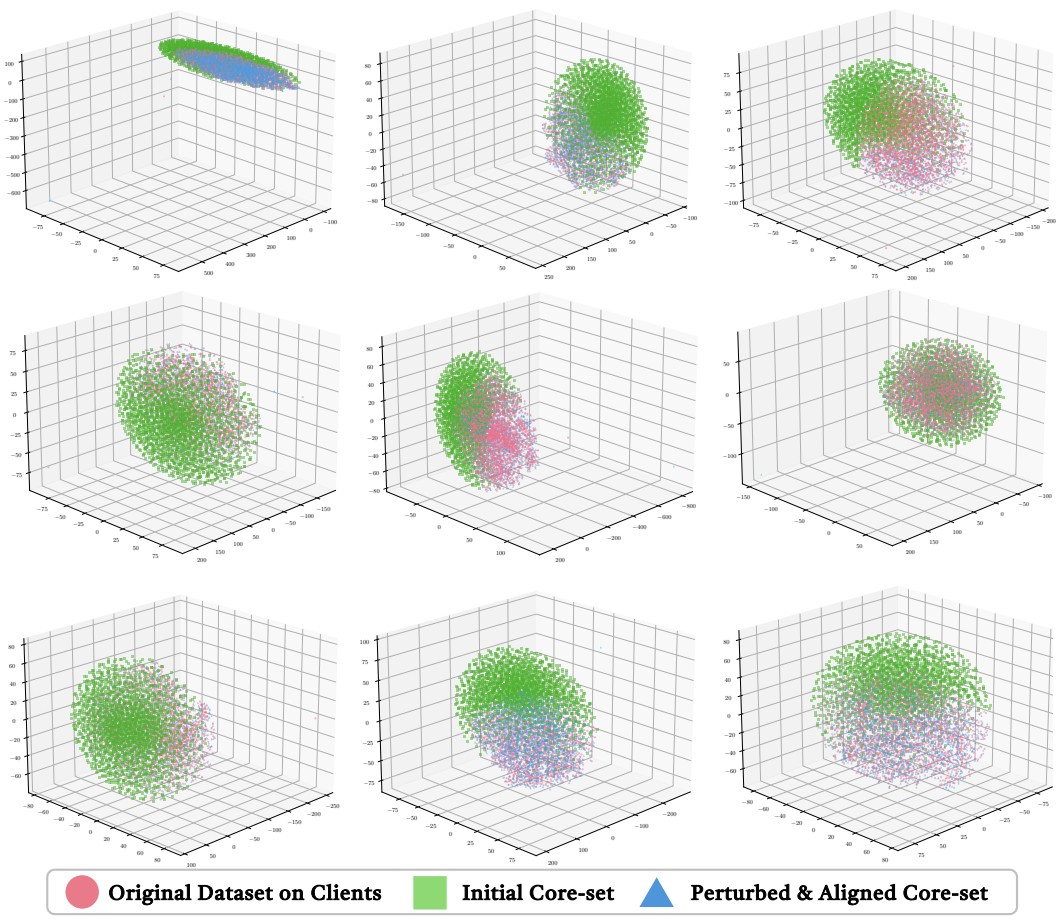

Figure 8: t-SNE visualization of core-set across clients. Each subfigure corresponds to a different client. The initial core-set closely aligns with the local data, while the perturbed core-set shows semantic similarity with added privacy-preserving shifts. *Best viewed in color and zoom-in*.

**Time Series Decomposition** Time series decomposition separates a sequence into interpretable components, typically trend, seasonal, and residual parts, enabling models to capture structured temporal patterns while isolating noise or biases. In our framework, we adopt the decomposition strategy from (Wu et al., 2021), but apply it directly in the latent representation space instead of the raw input domain. Given the latent representation $\mathbf{h} = f_{\theta^b}(\tilde{\boldsymbol{X}}) \in \mathbb{R}^{L \times d}$, we decompose it into trend and seasonal parts using a $\tau$-point moving average operator:

$$\mathbf{h}_\text{t} = \text{MA}_\tau(\mathbf{h}), \quad \mathbf{h}_\text{s} = \mathbf{h} - \mathbf{h}_\text{t}. \tag{15}$$

Here, $\mathbf{h}_\text{t}$ captures low-frequency temporal dynamics, while $\mathbf{h}_\text{s}$ represents higher-frequency structured variations. The residual component is omitted to avoid modeling unstructured noise. To estimate persistent dataset-specific deviations, we average each component over the temporal dimension:

$$\mathbf{b}_\text{t} = \text{Mean}(\mathbf{h}_\text{t}), \quad \mathbf{b}_\text{s} = \text{Mean}(\mathbf{h}_\text{s}), \tag{16}$$

where $\mathbf{b}_\text{t}$ and $\mathbf{b}_\text{s}$ are compact approximations of dataset-level biases arising from heterogeneous time series distributions.

**Training Configuration** We evaluate FeDaL from two complementary perspectives: (1) its ability to learn generic representations in heterogeneous, distributed settings (referred to as ***Federated Representation Learning***), and (2) the transferability of the learned TSFM to various downstream time series tasks (***Downstream Adaptation***). The detailed configurations for both tasks

are summarized in **Table 9**. For federated representation learning, we use the UTSD and CTSD datasets to assess FeDaL under two forms of cross-domain heterogeneity: domain-mixed (DM) and domain-independent (DI). For downstream adaptation, we pre-train the Decoder-only Transformer via FeDaL on LOTSA, a large-scale cross-domain dataset comprising 231 billion time points, to ensure broad coverage of temporal patterns prior to adaptation.

Table 9: Detailed training configuration across various tasks.

| Category | Configuration
Federated Representation Learning | Configuration
Pretraining for TSFMs |
|---|---|---|
| Optimizer | SGD | SGD |
| Number of Clients | 10 (UTSD) / 18 (CTSD) | 174 (LOTSA) |
| Batch Size | 2048 | 2048 |
| Client Join Ratio ($\rho$) | 70% | 70% |
| Local Epochs | 10 | 10 |
| Global Rounds | 200 | 1000 |
| Learning Rate (Local Updating) | 0.01 | 0.01 |
| Core-set Generation Epochs | 5 | 5 |
| Learning Rate (Core-set Generation) | 0.005 | 0.01 |
| Server-side Tuning | 5 | 15 |
| Learning Rate (Server-side Tuning) | 0.01 | 0.01 |
| Input Sequence Length | 1024 | 3072 |
| Learning Scheduler | StepLR | StepLR |
| Computation Devices | $2 \times$ Nvidia RTX A5500-24GB GPUs | $16\times$ Nvidia A100-80G GPUs |
| Smoother Factor ($\mu$) | 0.2 | 0.1 |
| $\alpha$ | 0.7 | 0.7 |
| $\beta$ | 0.1 | 0.1 |
| $\lambda$ (Alignment) | 0.01 | 0.1 |
| Core-set Size ($K$) | 1024 ($< 1\%$) | 3072 ($< 0.01\%$) |
| Decomposition Period ($\tau$) | 4 | 8 |
| Model Dimension | 512 | 768 |
| Number of Heads | 4 | 8 |
| Decoder Layers | 4 | 8 |
| Feedforward Dimension | 512 | 768 |
| Dropout | 0.1 | 0.35 |
| Activation Function | GELU | GELU |
| Masking Ratio | 75% | 50% |
| Patch Length | 32 | 64 |
| Stride | 32 | 32 |
| Evaluation Pipeline | Masked Patches Reconstruction | Downstream Adaption |
| Downstream Tasks | N/A | Long-term Forecasting (Full-/Zero-shot)
Short-term Forecasting
Classification
Anomaly Detection |

**Benchmark Details** We evaluate FeDaL across two major settings: **Federated Representation Learning** and **Downstream Adaptation**. For *Federated Representation Learning*, we use two large-scale cross-domain time series datasets: (1) **UTSD**[1], containing over 1 billion time points across 7 domains (e.g., energy, environment, health, IoT, nature); and (2) **CTSD**[2], sampled from the Monash Time Series Forecasting Repository, comprising 500 million time points from 6 domains including ergonomics, transportation, health, energy, nature, and web. For *Downstream Adaptation*, we pretrain on the large-scale LOTSA dataset (Woo et al., 2024) [3], which contains 231 billion observations spanning multiple domains. We evaluate the pretrained TSFM from FeDaL on four downstream tasks: long- and short-term forecasting, imputation, classification, and anomaly detection. For **long-term forecasting and imputation**, we follow the GPT4TS (Zhou et al., 2023) protocol, using ETTh1, ETTh2, ETTm1, ETTm2, Weather, and Illness datasets (details in **Table 11**). For **short-term forecasting**, we adopt the M4 dataset, again following GPT4TS, covering various temporal granularities (details in **Table 12**). For **classification**, we follow the TimesBERT (Zhang et al., 2025b) setup, using 10 representative datasets from UEA and 91 from UCR to cover diverse domains (see **Table 13**). For **anomaly detection**, we adopt the FFTS (Chen et al., 2025) benchmark,

---

[1] https://huggingface.co/datasets/thuml/UTSD

[2] https://forecastingdata.org/

[3] https://huggingface.co/datasets/Salesforce/lotsa_data

evaluating on SMD, MSL, SMAP, SWaT, and PSM datasets (details in **Table 14**). Any datasets overlapping with LOTSA have been excluded from the pretraining phase.

Table 10: Dataset statistics about the CTSD dataset. The channels indicates the number of time series. 'Min' and 'Max' denote the shortest and longest sequence lengths, respectively, while 'Fixed' signifies datasets with uniform sequence lengths.

| Domain | Dataset | # Channels | Frequency | Length |
|---|---|---|---|---|
| Economic | Bitcoin | 18 | Daily | Min: 2659 Max: 4581 |
| | FRED-MD | 107 | Monthly | Fixed: 728 |
| | NN5 | 111 | Daily | Fixed: 791 |
| Transport | Pedestrain Counts | 66 | Hourly | Min: 576 Max: 96424 |
| | Rideshare | 2304 | Daily | Fixed: 541 |
| | San Francisco Traffic | 862 | Hourly/Weekly | Fixed: 17544 |
| Health | COVID Deaths | 266 | Daily | Min: 212 Max: 212 |
| | Hospital | 767 | Monthly | Fixed: 84 |
| Energy | London Smart Meters | 5560 | Daily | Min: 288 Max: 39648 |
| | Wind Farms | 339 | Minutely | Min: 6345 Max: 527040 |
| | Wind Power | 1 | Second | Fixed: 7397147 |
| | Electricity | 321 | Hourly/Weekly | Fixed: 26304 |
| Nature | KDD Cup 2018 | 270 | Hourly | Min: 9504 Max: 10920 |
| | Oikolab Weather | 3010 | Daily | Min: 1332 Max: 65981 |
| | Temperature Rain | 32072 | Daily | Fixed: 725 |
| Web | Web Traffic | 145063 | Daily | Fixed: 803 |

Table 11: Dataset statistics about long-term forecasting (imputation) dataset. The channels indicates the number of time series (i.e., variables), and the size is organized in (training, validation, testing). Note that Illness dataset is not included in the imputation task.

| Dataset | Domain | # Channels | Frequency | Size | Forecast Length |
|---|---|---|---|---|---|
| ETTh1 | Power | 7 | Hourly | (8545, 2881, 2881) | $\{96, 192, 336, 720\}$ |
| ETTh2 | Power | 7 | Hourly | (8545, 2881, 2881) | $\{96, 192, 336, 720\}$ |
| ETTm1 | Power | 7 | 15 Minute | (34465, 11521, 11521) | $\{96, 192, 336, 720\}$ |
| ETTm2 | Power | 7 | 15 Minute | (34465, 11521, 11521) | $\{96, 192, 336, 720\}$ |
| Illness | Epidemiology | 7 | Weekly | (617, 74, 170) | $\{24, 36, 48, 60\}$ |
| Weather | Weather | 21 | 10 Minute | (36792, 5271, 10540) | $\{96, 192, 336, 720\}$ |

Table 12: Dataset statistics about short-term forecasting dataset. The channels indicates the number of time series (i.e., variables), and the size is organized in (training, validation, testing).

| Dataset | Domain | # Channels | Frequency | Size | Forecast Length |
|---|---|---|---|---|---|
| M4-Yearly | Demographic | 1 | Yearly | (23000, 0, 23000) | 6 |
| M4-Quarterly | Finance | 1 | Quarterly | (24000, 0, 24000) | 8 |
| M4-Monthly | Industry | 1 | Monthly | (48000, 0, 48000) | 18 |
| M4-Weekly | Macro | 1 | Weekly | (359, 0, 359) | 13 |
| M4-Daily | Macro | 1 | Daily | (4227, 0, 4227) | 14 |
| M4-Hourly | Other | 1 | Hourly | (414, 0, 414) | 48 |

Table 13: Dataset statistics about classification dataset. The channels indicates the number of time series (i.e., variables), and the size is organized in (training, validation, testing).

| Dataset | # Channels | Series Length | Dataset Size | Information (Frequency) |
|---|---|---|---|---|
| EthanolConcentration | 3 | 1751 | (261, 0, 263) | Alcohol Industry |
| FaceDetection | 144 | 62 | (5890, 0, 3524) | Face (250Hz) |
| Handwriting | 3 | 152 | (150, 0, 850) | Handwriting |
| Heartbeat | 61 | 405 | (204, 0, 205) | Heart Beat |
| JapaneseVowels | 12 | 29 | (270, 0, 370) | Voice |
| PEMS-SF | 963 | 144 | (267, 0, 173) | Transportation (Daily) |
| SelfRegulationSCP1 | 6 | 896 | (268, 0, 293) | Health (256Hz) |
| SelfRegulationSCP2 | 7 | 1152 | (200, 0, 180) | Health (256Hz) |
| SpokenArabicDigits | 13 | 93 | (6599, 0, 2199) | Voice (11025Hz) |
| UWaveGestureLibrary | 3 | 315 | (120, 0, 320) | Gesture |
| UCR Archive | 1 | * | (*, 0, *) | * |

Table 14: Dataset statistics about anomaly detection dataset. The channels indicates the number of time series (i.e., variables), and the size is organized in (training, validation, testing).

| Dataset | # Channels | Series Length | Dataset Size | Information (Frequency) |
|---|---|---|---|---|
| SMD | 38 | 40 | (566724, 141681, 708420) | Server Machine |
| MSL | 55 | 40 | (44653, 11664, 73729) | Spacecraft |
| SMAP | 25 | 40 | (108146, 27037, 427617) | Spacecraft |
| SWaT | 51 | 40 | (396000, 99000, 449919) | Infrastructure |
| PSM | 25 | 40 | (105984, 26497, 87841) | Server Machine |

**Data Processing Pipeline** Given the varying lengths of time series in the pretraining dataset, some sequences are too short to support the full sequence length used during pretraining (3072 points). Moreover, despite the large scale of LOTSA, we aim to expose the model to as diverse a set of temporal patterns as possible. To address these challenges, we introduce an efficient dataset processing pipeline for client-side pretraining. This pipeline (1) filters out invalid or insufficiently long time series samples, (2) augments the original time series to enrich temporal diversity, and (3) flexibly handles multi-domain time series with varying numbers of channels or variables in a unified manner. The source code for the unified client-side data processing pipeline is provided below.

```python
class FederatedDataset_Representation(Dataset):
    def __init__(self, args, series_data: dict,
                 mode='train',
                 ls='normal',
                 scale=True,
                 split_ratio=0.7):
        self.initialize_parameters(args, mode, ls, scale)
        self.process_data(series_data, split_ratio)

    def initialize_parameters(self, args, mode, ls, scale):
        self.args = args
        self.mode = mode
        self.ls = ls
        self.scale = scale

        self.subset_size = args.subset_size
        self.window_stride = args.window_stride
        self.seq_len = args.seq_len
        self.pred_len = args.pred_len
        self.total_len = self.seq_len + self.pred_len

    def process_data(self, series_data: dict, split_ratio: float):
```

```python
23          processed_series = []
24          skipped_series = 0
25
26          items_to_process = list(series_data.items())
27          for key, series in items_to_process:
28              required_len = self.seq_len
29              if len(series) < required_len:
30                  skipped_series += 1
31                  continue
32
33              if self.scale:
34                  scaler = StandardScaler()
35                  series = scaler.fit_transform(series)
36
37              n_features = series.shape[1]
38              for feature_idx in range(n_features):
39                  feature_series = series[:, feature_idx]
40
41                  windows = self.create_windows(
42                      feature_series.reshape(-1, 1),
43                      required_len,
44                      self.window_stride
45                  )
46                  processed_series.extend(windows)
47
48          if len(processed_series) == 0:
49              raise ValueError("No valid sequences found after length
               ↪ filtering!")
50
51          self.data = np.array(processed_series)
52          split_idx = int(len(self.data) * split_ratio)
53
54      def __len__(self):
55          return len(self.data)
56
57      def __getitem__(self, index):
58          seq_x = self.data[index]
59          return torch.from_numpy(seq_x)
60
61      @staticmethod
62      def create_windows(series, window_size, stride=32):
63          windows = []
64          for i in range(0, len(series) - window_size + 1, stride):
65              windows.append(series[i:i + window_size])
66          return windows
```

**Data Partitioning in Federated Time Series Representation Learning**  Learning domain-invariant representations from unlabeled, heterogeneous time series is crucial for advancing TSFMs in federated settings. To evaluate FeDaL in this context, we adopt two large-scale cross-domain benchmarks: UTSD (Woo et al., 2024) (1B time points across 7 domains) with domain-mixed (DM) partitioning, and CTSD (Chen et al., 2025) (500M time points across 6 domains) with domain-independent (DI) partitioning. In the DM setting, each client receives data drawn from multiple domains, while in the DI setting, each client is restricted to a single domain. For better illustration, examples of both partitioning strategies are shown in **Figure 9**. To further assess FeDaL under varying degrees of heterogeneity, we construct two additional DM-based configurations on UTSD, denoted H1 and H2. These are generated using a customized data-loading pipeline that controls heterogeneity through an imbalance factor (parameter: `imbalance_factor`) (set to 0.1 for H1 and 0.3 for H2) in the static method `_split_series`. This allows evaluation of FeDaL's robustness across environments with increasing domain skew.

```python
class FedRepresentationDatasetConfig:
    DATASET_REGISTRY: Dict[str, str] = {
    "UTSD": "UTSD"
    }
    SINGLE_DATASET_CONFIGS: Dict[str, Dict] = {
        "UTSD_unbalance_H2":{
            "dataset": "UTSD",
            "max_clients": 10,
            "split_method": "random",
            "balanced": False,
            "imbalance_factor": 0.3 # by default
        },
        "UTSD_unbalance_H1":{
            "dataset": "UTSD",
            "max_clients": 10,
            "split_method": "random",
            "balanced": False,
            "imbalance_factor": 0.1 # by default
        }
    }

    @classmethod
    def get_max_clients(cls, dataset_name: str) -> int:
        if dataset_name not in cls.SINGLE_DATASET_CONFIGS:
            raise ValueError(f"Dataset {dataset_name} not found in
            ↪  configurations")
        return cls.SINGLE_DATASET_CONFIGS[dataset_name]["max_clients"]

    @classmethod
    def get_total_series(cls, dataset_name: str) -> int:
        if dataset_name not in cls.SINGLE_DATASET_CONFIGS:
            raise ValueError(f"Dataset {dataset_name} not found in
            ↪  configurations")

        dataset_key =
        ↪  cls.SINGLE_DATASET_CONFIGS[dataset_name]["dataset"]
        file_path = os.path.join('datasets', 'crossdomain',
        ↪  f'{cls.DATASET_REGISTRY[dataset_key]}.npz')

        with np.load(file_path) as data:
            return len(data.keys())

    @classmethod
    def get_series_client_data(cls, dataset_name: str, client_id: int):
        if dataset_name not in cls.SINGLE_DATASET_CONFIGS:
            raise ValueError(f"Dataset {dataset_name} not configured
            ↪  for series-based splitting")

        config = cls.SINGLE_DATASET_CONFIGS[dataset_name]
        if client_id >= config["max_clients"]:
            raise ValueError(f"Client ID {client_id} exceeds maximum
            ↪  clients {config['max_clients']}")

        dataset_key =
        ↪  cls.SINGLE_DATASET_CONFIGS[dataset_name]["dataset"]
        file_path = os.path.join('datasets', 'crossdomain',
        ↪  f'{cls.DATASET_REGISTRY[dataset_key]}.npz')
        split_method = config["split_method"]

        return cls._split_series(file_path, split_method, client_id,
        ↪  config["max_clients"], config["balanced"],
        ↪  config["imbalance_factor"])

    @staticmethod
```

```python
55    def _split_series(file_path: str, split_method: str, client_id:
      ↪    int, max_clients: int, balanced: bool = True, imbalance_factor:
      ↪    float = 0.3):
56        with np.load(file_path) as data:
57            keys = list(data.keys())
58            n_series = len(keys)
59
60            if split_method == "random":
61                np.random.seed(42)
62                shuffled_keys = np.random.permutation(keys)
63
64                if balanced:
65                    series_per_client = n_series // max_clients
66                    extra_series = n_series % max_clients
67                    start_idx = client_id * series_per_client +
                      ↪    min(client_id, extra_series)
68                    end_idx = start_idx + series_per_client + (1 if
                      ↪    client_id < extra_series else 0)
69
70                else:
71                    alpha = np.ones(max_clients) * (1.0 /
                      ↪    imbalance_factor)
72                    proportions = np.random.dirichlet(alpha)
73                    series_counts = (proportions *
                      ↪    n_series).astype(int)
74                    series_counts[-1] = n_series -
                      ↪    np.sum(series_counts[:-1])
75                    start_idx = np.sum(series_counts[:client_id])
76                    end_idx = start_idx + series_counts[client_id]
77                client_series = {}
78                for key in shuffled_keys[start_idx:end_idx]:
79                    client_series[key] = data[key]
80            else:
81                raise ValueError(f"Unknown split method:
                  ↪    {split_method}")
82
83            return client_series
```

**Baseline** We compare our proposed FeDaL against 54 different baselines in our experiments to demonstrate its effectiveness and superiority in federated time series representation learning and downstream time series analysis tasks. In this section, we introduce some of these representative baselines from both federated learning and deep time series modeling. Full information of baselines can be found in **Table 15**. The key federated learning baselines include:

- FedAvg (McMahan et al., 2017): A decentralized approach that enables devices to collaboratively learn a shared model by aggregating locally-computed updates, allowing for the training of deep networks on private and large datasets while reducing communication costs.

- FedProx (Li et al., 2020): A federated learning framework that generalizes and reparameterizes FedAvg to tackle systems and statistical heterogeneity, providing convergence guarantees and demonstrating more robust and accurate convergence behavior.

- FedPer (Arivazhagan et al., 2019): A federated learning baseline that personalizes models by keeping client-specific layers local while training shared base layers collaboratively.

- FedRep (Collins et al., 2021): A federated learning baseline that decouples learning into global feature representation (shared) and client-specific heads (local) for better personalization.

- FFTS (Chen et al., 2025) A FL-based decentralized framework for training time series foundation models from scratch, which incorporates a Mixture-of-Experts mechanism on

Table 15: Baseline information pertains to federated representation learning and its applications in forecasting, imputation, anomaly detection, and classification for downstream adaptation. Regarding the Foundation Model class, all are employed exclusively for the zero-shot long-term forecasting task, except for Moment, which is applied across the entire scenario.

| | Baseline | Venue | Task | | | | |
|---|---|---|---|---|---|---|---|
| | | | Representation | Forecasting | Imputation | Anomaly | Classification |
| **FL Methods** | FedAvg | AISTATS'17 (2017) | ✔ | ✔ | ✔ | ✔ | ✔ |
| | FedProx | MLSys'20 (2020) | ✔ | ✗ | ✗ | ✗ | ✗ |
| | FedPer | arXiv'19 (2019) | ✔ | ✗ | ✗ | ✗ | ✗ |
| | FedRep | ICML'21 (2021) | ✔ | ✗ | ✗ | ✗ | ✗ |
| | FFTS | AAAI'25 (2025) | ✔ | ✔ | ✔ | ✔ | ✔ |
| | **FeDaL (Ours)** | **This paper** | ✔ | ✔ | ✔ | ✔ | ✔ |
| **Deep Models and Traditional Machine Learning Methods** | TimeMixer (2024b) | ICLR'24 (2024b) | ✗ | ✔ | ✔ | ✗ | ✗ |
| | GPT4TS | NeurIPS'23 (2023) | ✗ | ✔ | ✔ | ✔ | ✔ |
| | PatchTST | ICLR'23 (2022) | ✗ | ✔ | ✔ | ✔ | ✔ |
| | TimesNet | ICLR'23 (2022a) | ✗ | ✔ | ✔ | ✔ | ✔ |
| | DLinear | AAAI'23 (2023) | ✗ | ✔ | ✔ | ✔ | ✔ |
| | Fedformer | ICML'22 | ✗ | ✔ | ✔ | ✔ | ✔ |
| | Autoformer | NeurIPS'21 | ✗ | ✔ | ✔ | ✔ | ✔ |
| | Stationary | NeurIPS'22 (2022) | ✗ | ✔ | ✔ | ✔ | ✔ |
| | LightTS | arXiv'22 (2022) | ✗ | ✔ | ✔ | ✔ | ✔ |
| | Informer | AAAI'21 (2021) | ✗ | ✔ | ✔ | ✔ | ✔ |
| | Reformer | ICLR'20 (2020) | ✗ | ✔ | ✔ | ✔ | ✔ |
| | Time-LLM | ICLR'24 (2023) | ✗ | ✔ | ✗ | ✗ | ✗ |
| | N-HiTS | AAAI'23 (2022) | ✗ | ✔ | ✗ | ✗ | ✗ |
| | N-BEATS | ICLR'20 (2020) | ✗ | ✔ | ✗ | ✗ | ✗ |
| | ETSformer | arXiv'22 (2022) | ✗ | ✔ | ✗ | ✗ | ✔ |
| | MTCN | ICLR'24 (2024) | ✗ | ✗ | ✗ | ✔ | ✔ |
| | Pyraformer | ICLR'22 (2021) | ✗ | ✔ | ✔ | ✔ | ✔ |
| | Anomaly | ICLR'22 (2022) | ✗ | ✗ | ✗ | ✔ | ✗ |
| | DTW | ICML'17 (2017) | ✗ | ✗ | ✗ | ✗ | ✔ |
| | TS2Vec | AAAI'22 (2022) | ✗ | ✗ | ✗ | ✗ | ✔ |
| | T-Loss | NeurIPS'19 (2019) | ✗ | ✗ | ✗ | ✗ | ✔ |
| | TNC | ICLR'21 (2021) | ✗ | ✗ | ✗ | ✗ | ✔ |
| | TS-TCC | IJCAI'21 (2021) | ✗ | ✗ | ✗ | ✗ | ✔ |
| | TST | / | ✗ | ✗ | ✗ | ✗ | ✗ |
| | CNN | / | ✗ | ✗ | ✗ | ✗ | ✔ |
| | Encoder | / | ✗ | ✗ | ✗ | ✗ | ✔ |
| | FCN | / | ✗ | ✗ | ✗ | ✗ | ✔ |
| | MCDNN | CVPR'12 (2012) | ✗ | ✗ | ✗ | ✗ | ✔ |
| | ResNet | CVPR'16 (2016) | ✗ | ✗ | ✗ | ✗ | ✔ |
| | t-LeNet | ECML'16 (2016) | ✗ | ✗ | ✗ | ✗ | ✔ |
| | TWIESN | TR'06 (2006) | ✗ | ✗ | ✗ | ✗ | ✔ |
| | MLP | / | ✗ | ✗ | ✗ | ✗ | ✔ |
| | XGBoost | KDD'16 (2016) | ✗ | ✗ | ✗ | ✗ | ✔ |
| | Rocket | DMKD'20 (2020) | ✗ | ✗ | ✗ | ✗ | ✔ |
| | LSTM | NC'97 (1997) | ✗ | ✗ | ✗ | ✔ | ✔ |
| | LSTNet | SIGIR'18 (2018) | ✗ | ✗ | ✗ | ✗ | ✔ |
| | LSSL | ICLR'22 (2021) | ✗ | ✗ | ✗ | ✔ | ✔ |
| | Transformer | NeurIPS'17 (2017) | ✗ | ✔ | ✔ | ✔ | ✔ |
| | Flowformer | ICML'22 (2022b) | ✗ | ✗ | ✗ | ✗ | ✔ |
| | LogTransformer | NeurIPS'19 (2019) | ✗ | ✗ | ✗ | ✔ | ✗ |
| | TCN | NeurIPS'19 (2019) | ✗ | ✗ | ✗ | ✔ | ✔ |
| **Foundation Model** | Moment | ICML'24 (2024) | ✔ | ✔ | ✔ | ✔ | ✔ |
| | Moirai$_{small}$ | ICML'24 (2024) | ✗ | ✔ | ✗ | ✗ | ✗ |
| | Morial$_{base}$ | ICML'24 (2024) | ✗ | ✔ | ✗ | ✗ | ✗ |
| | Morial$_{large}$ | ICML'24 (2024) | ✗ | ✔ | ✗ | ✗ | ✗ |
| | TimesFM | ICML'24 (2024) | ✗ | ✔ | ✗ | ✗ | ✗ |
| | Chronos$_{small}$ | TMLR'24 (2024) | ✗ | ✔ | ✗ | ✗ | ✗ |
| | Chronos$_{base}$ | TMLR'24 (2024) | ✗ | ✔ | ✗ | ✗ | ✗ |
| | Chronos$_{large}$ | TMLR'24 (2024) | ✗ | ✔ | ✗ | ✗ | ✗ |

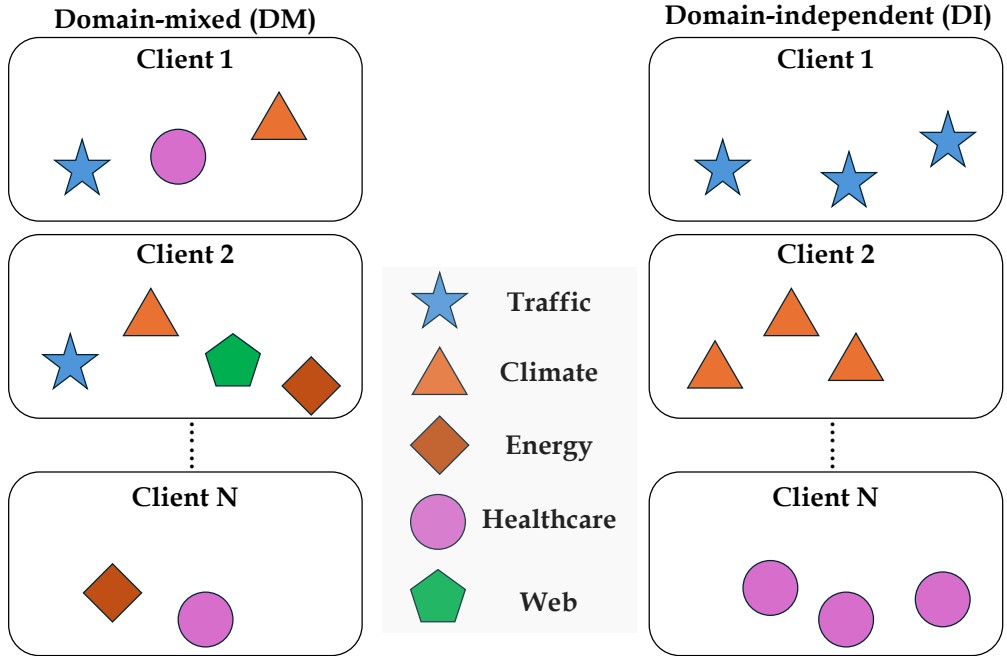

Figure 9: Comparison between Domain-Mixed (DM) and Domain-Independent (DI) settings. In DM, each client contains time series from multiple domains. In contrast, in DI, each client contains time series from a single domain, though the same domain may appear across different clients with non-overlapping sequences.

each client and employs a heterogeneous knowledge alignment strategy to address time series heterogeneity.

Details of deep time series model baseline is as follows:

- LogTransformer (Li et al., 2019): A modified Transformer architecture that addresses the locality-agnostics and memory bottleneck issues in time series forecasting by incorporating convolutional self-attention and log-sparse. attention, achieving improved forecasting accuracy with reduced memory cost.

- N-BEATS (Oreshkin et al., 2020): A deep neural architecture that achieves state-of-the-art performance in univariate time series point forecasting, using a stack of fully-connected layers with backward and forward residual links.

- Reformer (Kitaev et al., 2020): This model improves Transformer by using locality-sensitive hashing for attention and reversible residual layers. It offers better memory efficiency and speed for lengthy sequences without sacrificing performance.

- Informer (Zhou et al., 2021): An optimized Transformer-based model for long-range time series prediction. It uses ProbSparse self-attention for efficiency, processes long inputs effectively, and employs a fast prediction decoder.

- LightTS (Zhang et al., 2022): A lightweight MLP structure. It utilizes two downsampling strategies—spaced and sequential sampling—on the MLP structure, capitalizing on the fact that downsampled time series generally maintain most of their original information.

- ETSformer (Woo et al., 2022): A Transformer architecture that leverages the principle of exponential smoothing to improve traditional Transformers for time-series forecasting, offering better decomposition capability, interpretability, and long-term forecasting efficiency.

- Stationary (Liu et al., 2022): A framework that addresses the over-stationarization problem in time series forecasting by combining Series Stationarization and De-stationary Attention modules, which unify statistics for better predictability while recovering intrinsic non-stationary information.

- Autoformer (Wu et al., 2021): A decomposition architecture that leverages leveraging an Auto-Correlation mechanism for long-term forecasting, which efficiently discovers dependencies and aggregates representations at the sub-series level.

- FEDformer (Zhou et al., 2022): A forecasting method that combines seasonal-trend decomposition with a frequency-enhanced Transformer, capturing both the global profile and detailed structures of time series.

- Pyraformer (Liu et al., 2021): It features hierarchical pyramidal attention modules with binary trees to capture temporal dependencies across different ranges efficiently, both in time and memory complexity.

- AnomalyTransformer (Xu et al., 2022): A approach for unsupervised time series anomaly detection that leverages the self-attention mechanism to compute association discrepancy, which captures the adjacent-concentration bias of anomalies.

- TimesNet (Wu et al., 2022a): A framework that transforms 1D time series into 2D tensors to model complex temporal variations, leveraging the multi-periodicity of time series to adaptively discover and extract intraperiod- and interperiod-variations.

- PatchTST (Nie et al., 2022): This method divides the time series into patches at the sub-series level for input to the Transformer. Each channel holds a univariate time series, sharing the same embedding and Transformer weights across all series.

- DLinear (Zeng et al., 2023): DLinear integrates decomposition schemes from Autoformer and FEDformer with linear layers to model time series data tables. It effectively summarizes trend and seasonal components, enhancing performance on datasets rich in trends.

- N-HiTS (Challu et al., 2022): A forecasting model that addresses the challenges of long-horizon forecasting by incorporating hierarchical interpolation and multi-rate data sampling techniques.

- GPT4TS (Zhou et al., 2023): This model is designed for time series analysis across various scenarios, achieved by fine-tuning a pre-trained language model, specifically GPT2, for the time series domain.

- Time-LLM (Jin et al., 2023): A reprogramming framework that repurposes large language models (LLMs) for general time series forecasting by aligning time series data with natural language modalities through text prototypes and Prompt-as-Prefix (PaP) techniques.

- TimeMixer (Wang et al., 2024b): An advanced time series forecasting baseline that employs a multi-scale mixing architecture with Past Decomposition Mixing (PDM) and Future Multi-predictor Mixing (FMM) modules to effectively disentangle and integrate seasonal and trend patterns.

**Code Implementation** The code is available at `https://anonymous.4open.science/r/FeDaL-3DC2`. This anonymous code repository is intended for review purposes only.

**Evaluation Metrics** For evaluation metrics in forecasting and imputation tasks, we utilize the mean square error (MSE) and mean absolute error (MAE) for long-term forecasting. In terms of the short-term forecasting on M4 benchmark, we adopt the symmetric mean absolute percentage error (SMAPE), mean absolute scaled error (MASE), and overall weighted average (OWA) as in N-BEATS. Note that OWA is a specific metric utilized in the M4 competition. The calculations of

these metrics are as follows:

$$\text{MSE} = \frac{1}{H} \sum_{h=1}^{T} (\mathbf{Y}_h - \hat{\mathbf{Y}}_h)^2,$$

$$\text{MAE} = \frac{1}{H} \sum_{h=1}^{H} |\mathbf{Y}_h - \hat{\mathbf{Y}}_h|,$$

$$\text{SMAPE} = \frac{200}{H} \sum_{h=1}^{H} \frac{|\mathbf{Y}_h - \hat{\mathbf{Y}}_h|}{|\mathbf{Y}_h| + |\hat{\mathbf{Y}}_h|}, \tag{17}$$

$$\text{MAPE} = \frac{100}{H} \sum_{h=1}^{H} \frac{|\mathbf{Y}_h - \hat{\mathbf{Y}}_h|}{|\mathbf{Y}_h|},$$

$$\text{MASE} = \frac{1}{H} \sum_{h=1}^{H} \frac{|\mathbf{Y}_h - \hat{\mathbf{Y}}_h|}{\frac{1}{H-s} \sum_{j=s+1}^{H} |\mathbf{Y}_j - \mathbf{Y}_{j-s}|},$$

$$\text{OWA} = \frac{1}{2} \left[ \frac{\text{SMAPE}}{\text{SMAPE}_{\text{Naïve2}}} + \frac{\text{MASE}}{\text{MASE}_{\text{Naïve2}}} \right],$$

where $s$ is the periodicity of the time series data. $H$ denotes the number of data points (i.e., prediction horizon in our cases). $\mathbf{Y}_h$ and $\hat{\mathbf{Y}}_h$ are the $h$-th ground truth and prediction where $h \in \{1, \cdots, H\}$.

In addition, we used Precision (P), Recall (R), and F1-Score (F1) to simply quantify the performance of our FeDaL and baselines on the anomaly detection task, these can be formulated as:

$$\text{P} = \frac{\text{TP}}{\text{TP} + \text{FP}},$$

$$\text{R} = \frac{\text{TP}}{\text{TP} + \text{FN}}, \tag{18}$$

$$\text{F1} = 2 \times \frac{\text{P} \times \text{R}}{\text{P} + \text{R}},$$

where TP (True Positives), FP (False Positives), and FN (False Negatives) represent the number of samples correctly labeled as anomalous, the number of samples incorrectly labeled as anomalous, and the number of samples that were not labeled as anomalous by the model but were actually anomalous, respectively. For classification tasks, we used Accuracy to evaluate the performance.

## C  FULL RESULTS

This section presents the complete results across full-shot and zero-shot long-term forecasting, short-term forecasting, imputation, anomaly detection, and classification. It also includes detailed plots and analysis of federated scaling behaviors during general time series foundation model training.

### C.1  TIME SERIES REPRESENTATION LEARNING

The full federated time series representation learning results are shown in **Table 16**, which shows our proposed FeDaL outperform advanced FL methods across various datasets and setups in unsupervised representation learning.

**Representation Learning Under Increased Heterogeneity**  . We further investigated FeDaL's performance under increasingly challenging, high-heterogeneity configurations, specifically employing imbalance ratios of $\{0.7, 0.8, 0.9\}$. The results are shown in **Table 17**, illustrate FeDaL's superior generalization and robust performance over competitive baselines. This confirms its ability to effectively handle severe non-IID conditions and maintain its leading edge.

### C.2  LONG-TERM FORECASTING

The full long-term forecasting results are provided in **Table 18** and **Table 19**. **Table 18** demonstrates that FeDaL-trained TSFM consistently outperforms state-of-the-art deep time series models

Table 16: Representation learning results under varying patch masking ratios. For UTSD, we simulate two levels of domain heterogeneity (H1 and H2) as detailed in the **Appendix B**. For CTSD, each dataset is treated as a domain-independent client. **Bold** indicates the best result, Underline the second best. † denotes evaluation using personalized models after server averaging; ‡ indicates client-side evaluation without aggregation.

| Method | USTD Dataset (10 Clients, DM) | | | | | | | | | | CTSD Dataset (18 Clients, DI) | | | | |
| | 20% | | 35% | | 50% | | 75% | | 90% | | 20% | 35% | 50% | 75% | 90% |
| | H1 | H2 | H1 | H2 | H1 | H2 | H1 | H2 | H1 | H2 | | | | | |
|---|---|---|---|---|---|---|---|---|---|---|---|---|---|---|---|
| FedAvg | 0.387 | 0.404 | 0.473 | 0.489 | 0.550 | 0.565 | 0.636 | 0.602 | 0.882 | 0.901 | 0.350 | 0.388 | 0.405 | 0.480 | 0.652 |
| FedProx | 0.382 | 0.401 | 0.469 | 0.486 | 0.546 | 0.553 | 0.638 | 0.603 | 0.880 | 0.889 | 0.336 | 0.390 | 0.400 | 0.454 | 0.638 |
| FedPer† | 0.343 | 0.385 | 0.459 | 0.578 | 0.540 | 0.533 | 0.621 | 0.580 | 0.860 | 0.863 | 0.330 | 0.381 | 0.395 | 0.439 | 0.606 |
| FedRep† | 0.390 | 0.413 | 0.451 | 0.560 | 0.529 | 0.537 | 0.621 | 0.592 | 0.853 | 0.860 | 0.352 | 0.373 | 0.386 | 0.440 | 0.598 |
| FFTS | **0.333** | 0.327 | 0.450 | **0.410** | 0.526 | 0.510 | 0.630 | 0.584 | 0.870 | 0.823 | **0.300** | 0.357 | 0.379 | 0.436 | 0.610 |
| Stand-alone‡ | 0.376 | 0.380 | 0.454 | 0.450 | 0.535 | 0.521 | 0.643 | 0.616 | 0.875 | 0.870 | 0.342 | 0.381 | 0.395 | 0.470 | 0.646 |
| **FeDaL (Ours)** | 0.348 | **0.300** | **0.436** | 0.422 | **0.521** | **0.489** | **0.596** | **0.549** | **0.852** | **0.795** | 0.310 | **0.319** | **0.343** | **0.405** | **0.560** |

Table 17: Federated representation learning results across different masking ratios and imbalance ratios (H3: 0.7, H4: 0.8, H5: 0.9).

| Imbalance Ratio | Method | Masking Ratio | | | | |
| (H) | | 20% | 35% | 50% | 75% | 90% |
|---|---|---|---|---|---|---|
| | FedAvg | 0.427 | 0.505 | 0.571 | 0.623 | 0.912 |
| | FedProx | 0.409 | 0.500 | 0.568 | 0.627 | 0.900 |
| | FedPer | 0.399 | 0.612 | 0.544 | 0.582 | 0.877 |
| H3 (0.7) | FedRep | 0.415 | 0.579 | 0.540 | 0.603 | 0.868 |
| | FFTS | 0.339 | **0.423** | 0.525 | 0.599 | 0.831 |
| | **FeDaL (Ours)** | **0.310** | 0.429 | **0.501** | **0.557** | **0.809** |
| | FedAvg | 0.431 | 0.506 | 0.582 | 0.626 | 0.917 |
| | FedProx | 0.412 | 0.509 | 0.577 | 0.630 | 0.926 |
| | FedPer | 0.406 | 0.618 | 0.560 | 0.597 | 0.886 |
| H4 (0.8) | FedRep | 0.418 | 0.592 | 0.555 | 0.606 | 0.871 |
| | FFTS | 0.350 | 0.435 | 0.537 | 0.601 | 0.844 |
| | **FeDaL (Ours)** | **0.322** | **0.430** | **0.511** | **0.560** | **0.824** |
| | FedAvg | 0.451 | 0.523 | 0.598 | 0.637 | 0.924 |
| | FedProx | 0.420 | 0.514 | 0.586 | 0.642 | 0.933 |
| | FedPer | 0.416 | 0.620 | 0.577 | 0.603 | 0.878 |
| H5 (0.9) | FedRep | 0.432 | 0.604 | 0.569 | 0.610 | 0.878 |
| | FFTS | 0.362 | 0.441 | 0.550 | 0.612 | 0.846 |
| | **FeDaL (Ours)** | **0.330** | **0.437** | **0.523** | **0.571** | **0.825** |

specifically designed for long-term forecasting. Furthermore, **Table 19** shows that under zero-shot evaluation, our approach surpasses both advanced centralized pretrained TSFMs and federated pre-training baselines, highlighting its strong generalization across domains and tasks.

**Discussion on Larger Time Series Models** Zero-shot generalization is widely regarded as the core of strong foundation models. In **Table 19**, we evaluate this capability under long-term forecasting tasks, comparing our FeDaL-trained TSFM against a range of baseline models (most of which are tailored for forecasting). The results demonstrate the superior zero-shot generalization ability of our model. To further contextualize performance, we compare our FeDaL-trained TSFM with Time-MoE (Shi et al., 2024), a recent large-scale time series foundation model designed specifically for forecasting. Time-MoE variants include models with up to 2.4B parameters trained on 300B time series data. We present results on five standard long-horizon forecasting benchmarks in **Table 20**, including each model's parameter count and training data size for visual comparison. Our findings show that FeDaL consistently outperforms Time-MoE$_{base}$ (113M) and Time-MoE$_{large}$ (453M), and achieves comparable results to the largest variant, Time-MoE$_{ultra}$ (2.4B). Notably, FeDaL reaches this level of performance with only 1.8% of the parameters and ∼70B fewer training samples, indicating substantially better efficiency. To quantify this tradeoff between performance and resource cost inspired by (Kaplan et al., 2020; Tan & Le, 2019), we define the Information Gain per Cost

Table 18: Full long-term forecasting results comparing our proposed FeDaL with advanced deep time series models. **Bold**: the best, Underline: the second best.

| Dataset | Method Metric | FeDaL (Ours) MSE MAE | TimeMixer MSE MAE | Time-LLM MSE MAE | GPT4TS MSE MAE | DLinear MSE MAE | PatchTST MSE MAE | TimesNet MSE MAE | FED. MSE MAE | Auto. MSE MAE | Stationary MSE MAE | ETS. MSE MAE | LightTS MSE MAE | Informer MSE MAE | Re. MSE MAE |
|---|---|---|---|---|---|---|---|---|---|---|---|---|---|---|---|
| ETTh1 | 96 | **0.340** **0.371** | 0.373 0.401 | 0.362 0.392 | 0.376 0.397 | 0.375 0.399 | 0.370 0.399 | 0.384 0.402 | 0.376 0.419 | 0.449 0.459 | 0.513 0.491 | 0.494 0.479 | 0.424 0.432 | 0.865 0.713 | 0.837 0.728 |
| | 192 | **0.371** | **0.400** 0.436 0.430 | 0.398 0.418 | 0.416 0.418 | 0.405 0.416 | 0.413 0.421 | 0.436 0.429 | 0.420 0.448 | 0.500 0.482 | 0.534 0.504 | 0.538 0.504 | 0.475 0.462 | 1.008 0.792 | 0.923 0.766 |
| | 336 | **0.392** | **0.420** 0.484 0.458 | 0.430 0.427 | 0.442 0.433 | 0.439 0.443 | 0.422 0.436 | 0.491 0.469 | 0.459 0.465 | 0.521 0.496 | 0.588 0.535 | 0.574 0.521 | 0.518 0.488 | 1.107 0.809 | 1.097 0.835 |
| | 720 | **0.417** | **0.445** 0.497 0.482 | 0.442 0.457 | 0.477 0.456 | 0.472 0.490 | 0.447 0.466 | 0.521 0.500 | 0.506 0.507 | 0.514 0.512 | 0.643 0.616 | 0.562 0.535 | 0.547 0.533 | 1.181 0.865 | 1.257 0.889 |
| | Avg. | **0.380** | **0.409** 0.448 0.443 | 0.408 0.423 | 0.465 0.455 | 0.422 0.437 | 0.413 0.430 | 0.458 0.450 | 0.440 0.460 | 0.496 0.487 | 0.570 0.537 | 0.542 0.510 | 0.491 0.479 | 1.040 0.795 | 1.029 0.805 |
| ETTh2 | 96 | **0.267** | **0.325** 0.289 0.340 | 0.268 0.328 | 0.285 0.342 | 0.289 0.353 | 0.274 0.336 | 0.340 0.374 | 0.358 0.397 | 0.346 0.388 | 0.476 0.458 | 0.340 0.391 | 0.397 0.437 | 3.755 1.525 | 2.626 1.317 |
| | 192 | 0.333 | **0.369** 0.370 0.389 | **0.329** 0.375 | 0.354 0.389 | 0.383 0.418 | 0.339 0.379 | 0.402 0.414 | 0.429 0.439 | 0.456 0.452 | 0.512 0.493 | 0.430 0.439 | 0.520 0.504 | 5.602 1.931 | 11.12 2.979 |
| | 336 | 0.359 | 0.392 0.386 0.413 | 0.368 0.409 | 0.373 0.407 | 0.448 0.465 | **0.329** **0.380** | 0.452 0.452 | 0.496 0.487 | 0.482 0.486 | 0.552 0.551 | 0.485 0.479 | 0.626 0.559 | 4.721 1.835 | 9.323 2.769 |
| | 720 | 0.377 | **0.420** 0.412 0.432 | 0.372 **0.420** | 0.406 0.441 | 0.605 0.551 | 0.379 0.422 | 0.462 0.468 | 0.463 0.474 | 0.515 0.511 | 0.562 0.560 | 0.500 0.497 | 0.863 0.672 | 3.647 1.625 | 3.874 1.697 |
| | Avg | 0.334 | **0.377** 0.364 0.394 | 0.334 0.383 | 0.381 0.412 | 0.431 0.446 | **0.330** 0.379 | 0.414 0.427 | 0.437 0.449 | 0.450 0.459 | 0.526 0.516 | 0.439 0.452 | 0.602 0.543 | 4.431 1.729 | 6.736 2.191 |
| ETTm1 | 96 | **0.258** | **0.321** 0.320 0.357 | 0.272 0.334 | 0.292 0.346 | 0.299 0.343 | 0.290 0.342 | 0.338 0.375 | 0.379 0.419 | 0.505 0.475 | 0.386 0.398 | 0.375 0.398 | 0.374 0.400 | 0.672 0.571 | 0.538 0.528 |
| | 192 | **0.297** | **0.343** 0.361 0.380 | 0.310 0.358 | 0.332 0.372 | 0.335 0.365 | 0.332 0.369 | 0.374 0.387 | 0.426 0.441 | 0.553 0.496 | 0.459 0.444 | 0.408 0.410 | 0.400 0.407 | 0.795 0.669 | 0.658 0.592 |
| | 336 | **0.327** | **0.381** 0.392 0.404 | 0.352 0.384 | 0.366 0.394 | 0.369 0.386 | 0.366 0.392 | 0.410 0.411 | 0.445 0.459 | 0.621 0.537 | 0.495 0.464 | 0.435 0.428 | 0.438 0.438 | 1.212 0.871 | 0.898 0.721 |
| | 720 | 0.392 | 0.415 0.452 0.440 | **0.383** **0.411** | 0.417 0.421 | 0.425 0.421 | 0.416 0.420 | 0.478 0.450 | 0.543 0.490 | 0.671 0.561 | 0.585 0.516 | 0.499 0.462 | 0.527 0.502 | 1.166 0.823 | 1.102 0.841 |
| | Avg | **0.319** | **0.365** 0.381 0.395 | 0.329 0.372 | 0.388 0.403 | 0.357 0.378 | 0.351 0.380 | 0.400 0.406 | 0.448 0.452 | 0.588 0.517 | 0.481 0.456 | 0.429 0.425 | 0.435 0.437 | 0.961 0.734 | 0.799 0.671 |
| ETTm2 | 96 | **0.157** | **0.240** 0.175 0.258 | 0.161 0.253 | 0.173 0.262 | 0.167 0.269 | 0.165 0.255 | 0.187 0.267 | 0.203 0.287 | 0.255 0.339 | 0.192 0.274 | 0.189 0.280 | 0.209 0.308 | 0.365 0.453 | 0.658 0.619 |
| | 192 | 0.220 | **0.292** 0.235 0.298 | 0.219 0.293 | 0.229 0.301 | 0.224 0.303 | 0.220 0.292 | 0.249 0.309 | 0.269 0.328 | 0.281 0.340 | 0.280 0.339 | 0.253 0.319 | 0.311 0.382 | 0.533 0.563 | 1.078 0.827 |
| | 336 | **0.267** | **0.342** 0.298 0.341 | 0.271 **0.329** | 0.286 0.341 | 0.281 0.342 | 0.274 **0.329** | 0.321 0.351 | 0.325 0.366 | 0.339 0.372 | 0.334 0.361 | 0.314 0.357 | 0.442 0.466 | 1.363 0.887 | 1.549 0.972 |
| | 720 | 0.399 | 0.400 0.391 0.395 | 0.352 0.379 | 0.378 0.401 | 0.397 0.421 | 0.362 0.385 | 0.408 0.403 | 0.421 0.415 | 0.433 0.432 | 0.417 0.413 | 0.414 0.413 | 0.675 0.587 | 3.379 1.338 | 2.631 1.242 |
| | Avg | 0.261 | 0.319 0.275 0.323 | 0.251 **0.313** | 0.284 0.339 | 0.267 0.333 | 0.255 0.315 | 0.291 0.333 | 0.305 0.349 | 0.327 0.371 | 0.306 0.347 | 0.293 0.342 | 0.409 0.436 | 1.410 0.810 | 1.479 0.915 |
| Weather | 96 | **0.139** | **0.199** 0.162 0.209 | 0.147 0.201 | 0.162 0.212 | 0.176 0.237 | 0.149 0.198 | 0.172 0.220 | 0.217 0.296 | 0.266 0.336 | 0.173 0.223 | 0.197 0.281 | 0.182 0.242 | 0.300 0.384 | 0.689 0.596 |
| | 192 | **0.174** | **0.230** 0.208 0.250 | 0.189 0.234 | 0.204 0.248 | 0.220 0.282 | 0.194 0.241 | 0.219 0.261 | 0.276 0.336 | 0.307 0.367 | 0.245 0.285 | 0.237 0.312 | 0.227 0.287 | 0.598 0.544 | 0.752 0.638 |
| | 336 | **0.240** | **0.279** 0.252 0.287 | 0.262 0.279 | 0.254 0.286 | 0.265 0.319 | 0.245 0.282 | 0.280 0.306 | 0.339 0.380 | 0.359 0.395 | 0.321 0.338 | 0.298 0.353 | 0.282 0.334 | 0.578 0.523 | 0.639 0.596 |
| | 720 | **0.300** | **0.310** 0.340 0.343 | 0.304 0.316 | 0.326 0.337 | 0.333 0.362 | 0.314 0.334 | 0.365 0.359 | 0.403 0.428 | 0.419 0.428 | 0.414 0.410 | 0.352 0.288 | 0.352 0.386 | 1.059 0.741 | 1.130 0.792 |
| | Avg | **0.213** | **0.255** 0.241 0.272 | 0.225 0.257 | 0.237 0.270 | 0.248 0.300 | 0.225 0.264 | 0.259 0.287 | 0.309 0.360 | 0.338 0.382 | 0.288 0.314 | 0.271 0.334 | 0.261 0.312 | 0.634 0.548 | 0.803 0.656 |
| ILI | 24 | 1.265 | **0.714** 1.979 0.860 | 1.285 0.727 | 2.063 0.881 | 2.215 1.081 | 1.319 0.754 | 3.228 1.260 | 2.203 0.945 | 2.527 1.020 | 8.313 2.144 | 5.764 1.677 | 4.400 1.382 |
| | 36 | **1.329** | **0.800** 1.893 0.862 | 1.404 0.814 | 1.868 0.892 | 1.963 0.963 | 1.430 0.834 | 1.972 0.920 | 2.679 1.080 | 3.103 1.148 | 1.825 0.848 | 2.615 1.007 | 6.631 1.902 | 4.755 1.467 | 4.783 1.448 |
| | 48 | **1.409** | **0.768** 2.129 0.936 | 1.523 0.807 | 1.790 0.884 | 2.130 1.024 | 1.553 0.815 | 2.238 0.940 | 2.622 1.078 | 2.669 1.085 | 2.010 0.900 | 2.359 0.972 | 7.299 1.982 | 4.763 1.469 | 4.832 1.465 |
| | 60 | **1.418** | 0.810 2.155 0.938 | 1.531 0.854 | 1.979 0.957 | 2.368 1.096 | 1.470 **0.788** | 2.027 0.928 | 2.857 1.157 | 2.770 1.125 | 2.178 0.963 | 2.487 1.016 | 7.283 1.985 | 5.264 1.564 | 4.882 1.483 |
| | Avg | **1.355** | **0.773** 2.039 0.899 | 1.435 0.801 | 1.925 0.903 | 2.169 1.041 | 1.443 0.797 | 2.369 0.931 | 2.847 1.144 | 3.006 1.161 | 2.077 0.914 | 2.497 1.004 | 7.382 2.003 | 5.137 1.544 | 4.724 1.445 |
| 1st Count | | **46** | 0 | 13 | 0 | 0 | 5 | 0 | 0 | 0 | 0 | 0 | 0 | 0 | 3 |

Table 19: Full results of zero-shot forecasting experiments. A lower MSE or MAE indicates a better prediction. TimesFM, due to its use of Weather datasets in pretraining, is not evaluated on this dataset and is denoted by a dash ($-$). **Bold**: the best, Underline: the second best.

| Models | Metrics | Federated Learning Methods | | | Pretrained Time Series Foundation Models | | | | | | | | | |
|---|---|---|---|---|---|---|---|---|---|---|---|---|---|---|
| | | FeDaL (Ours) MSE MAE | FFTS MSE MAE | FedAvg MSE MAE | Moirai$_{small}$ MSE MAE | Moirai$_{base}$ MSE MAE | Moirai$_{large}$ MSE MAE | TimesFM MSE MAE | Moment MSE MAE | Chronos$_{small}$ MSE MAE | Chronos$_{base}$ MSE MAE | Chronos$_{large}$ MSE MAE |
| ETTh1 | 96 | 0.347 0.381 | **0.344** 0.382 | 0.412 0.409 | 0.401 0.402 | 0.376 0.392 | 0.349 **0.379** | 0.414 0.404 | 0.688 0.557 | 0.466 0.409 | 0.440 0.393 | 0.441 0.390 |
| | 192 | 0.398 **0.410** | 0.395 0.410 | 0.420 0.421 | **0.388** 0.412 | 0.412 0.413 | 0.434 0.415 | 0.465 0.434 | 0.688 0.560 | 0.530 0.450 | 0.492 0.426 | 0.502 0.424 |
| | 336 | 0.425 0.452 | 0.438 0.445 | 0.440 0.444 | 0.433 **0.428** | 0.433 0.428 | 0.495 0.445 | 0.503 0.456 | 0.675 0.563 | 0.570 0.486 | 0.550 0.462 | 0.576 0.467 |
| | 720 | 0.457 0.469 | 0.445 0.457 | 0.480 0.522 | **0.439** 0.454 | 0.447 **0.444** | 0.611 0.510 | 0.511 0.481 | 0.683 0.585 | 0.615 0.543 | 0.882 0.591 | 0.835 0.583 |
| | **Avg.** | **0.407** 0.429 | 0.425 0.437 | 0.438 0.449 | 0.428 0.427 | 0.417 0.419 | 0.480 0.439 | 0.473 0.443 | 0.683 0.566 | 0.545 0.472 | 0.591 0.468 | 0.588 0.466 |
| ETTh2 | 96 | 0.307 0.355 | 0.325 **0.332** | 0.340 0.350 | 0.297 0.356 | **0.294** 0.330 | 0.296 0.330 | 0.315 0.349 | 0.342 0.396 | 0.307 0.356 | 0.308 0.343 | 0.320 0.345 |
| | 192 | 0.349 0.372 | 0.355 0.359 | 0.378 0.388 | 0.368 0.381 | 0.365 0.375 | 0.361 0.371 | 0.388 0.395 | 0.354 0.402 | 0.376 0.401 | 0.384 0.392 | 0.406 0.399 |
| | 336 | 0.387 0.395 | 0.391 0.393 | 0.415 0.421 | 0.370 0.393 | 0.376 0.390 | 0.390 0.390 | 0.422 0.427 | **0.356** 0.407 | 0.408 0.431 | 0.429 0.430 | 0.492 0.453 |
| | 720 | **0.401** 0.406 | 0.409 0.412 | 0.427 0.441 | 0.411 0.426 | 0.416 0.433 | 0.423 0.418 | 0.443 0.454 | 0.395 0.434 | 0.604 0.533 | 0.501 0.477 | 0.603 0.511 |
| | **Avg.** | 0.361 0.382 | 0.370 **0.374** | 0.390 0.401 | 0.361 0.384 | 0.362 0.382 | 0.367 0.377 | 0.392 0.406 | 0.361 0.409 | 0.424 0.430 | 0.405 0.410 | 0.455 0.427 |
| ETTm1 | 96 | **0.289** **0.346** | 0.307 0.352 | 0.303 0.371 | 0.418 0.370 | 0.363 0.356 | 0.380 0.361 | 0.361 0.370 | 0.654 0.527 | 0.511 0.423 | 0.454 0.408 | 0.457 0.403 |
| | 192 | **0.317** 0.369 | 0.324 0.392 | 0.344 0.384 | 0.431 0.405 | 0.388 0.375 | 0.412 0.383 | 0.414 0.405 | 0.662 0.532 | 0.618 0.485 | 0.567 0.477 | 0.530 0.450 |
| | 336 | **0.370** 0.420 | 0.399 0.438 | 0.426 0.432 | 0.433 0.412 | 0.416 **0.392** | 0.460 0.418 | 0.445 0.429 | 0.672 0.537 | 0.683 0.524 | 0.662 0.525 | 0.577 0.481 |
| | 720 | 0.464 0.426 | 0.426 0.498 | 0.438 0.453 | 0.462 0.432 | 0.460 0.418 | 0.462 0.420 | 0.512 0.471 | 0.692 0.551 | 0.748 0.566 | 0.900 0.591 | 0.660 0.526 |
| | **Avg.** | **0.360** 0.390 | 0.364 0.420 | 0.378 0.410 | 0.436 0.410 | 0.406 0.385 | 0.422 0.391 | 0.433 0.418 | 0.670 0.536 | 0.640 0.499 | 0.645 0.500 | 0.555 0.465 |
| ETTm2 | 96 | 0.207 0.283 | 0.222 0.278 | 0.219 0.289 | 0.214 0.288 | 0.205 0.273 | 0.211 0.274 | 0.202 **0.270** | 0.260 0.335 | 0.209 0.291 | 0.199 0.274 | **0.197** 0.271 |
| | 192 | **0.248** 0.333 | 0.273 0.320 | 0.270 0.333 | 0.284 0.332 | 0.275 0.316 | 0.281 0.318 | 0.289 0.321 | 0.289 0.350 | 0.280 0.341 | 0.261 0.322 | 0.254 **0.314** |
| | 336 | **0.316** 0.340 | 0.320 0.327 | 0.321 0.340 | 0.331 0.362 | 0.329 0.350 | 0.341 0.355 | 0.360 0.366 | 0.324 0.369 | 0.354 0.390 | 0.326 0.366 | 0.313 0.353 |
| | 720 | **0.397** 0.408 | 0.453 0.523 | 0.478 0.498 | 0.402 **0.408** | 0.437 0.411 | 0.485 0.428 | 0.462 0.430 | **0.394** 0.400 | 0.553 0.499 | 0.455 0.439 | 0.416 0.415 |
| | **Avg.** | **0.292** 0.341 | 0.317 0.362 | 0.322 0.365 | 0.307 0.347 | 0.311 0.337 | 0.329 0.343 | 0.328 0.346 | 0.316 0.365 | 0.349 0.380 | 0.310 0.350 | 0.295 0.338 |
| Weather | 96 | **0.159** 0.212 | 0.172 0.218 | 0.201 0.207 | 0.198 0.222 | 0.220 0.217 | 0.199 0.211 | - - | 0.243 0.255 | 0.211 0.243 | 0.203 0.238 | 0.194 0.235 |
| | 192 | **0.217** 0.264 | 0.235 0.278 | 0.250 0.278 | 0.247 0.265 | 0.271 0.259 | 0.246 **0.251** | - - | 0.278 0.320 | 0.263 0.294 | 0.256 0.290 | 0.249 0.285 |
| | 336 | **0.285** 0.312 | 0.290 0.321 | 0.304 0.340 | 0.283 0.303 | 0.286 0.297 | 0.274 0.291 | - - | 0.306 0.346 | 0.321 0.339 | 0.314 0.336 | 0.302 0.327 |
| | 720 | **0.359** 0.348 | 0.351 0.383 | 0.353 0.395 | 0.373 0.354 | 0.373 0.354 | **0.337** 0.340 | - - | 0.350 0.374 | 0.404 0.397 | 0.397 0.396 | 0.372 0.378 |
| | **Avg.** | **0.255** 0.284 | 0.262 0.300 | 0.277 0.305 | 0.275 0.286 | 0.287 0.281 | 0.264 **0.273** | - - | 0.294 0.326 | 0.300 0.318 | 0.292 0.315 | 0.279 0.306 |
| 1st Count | | **17** | 6 | 1 | 4 | 9 | 8 | 1 | 3 | 0 | 0 | 3 |

(IGC) metric as:

$$\text{IGC} = \frac{1}{\text{MSE} \times \text{Parameters Count}^{\alpha} \times \text{Training Data Size}^{\beta}}, \tag{19}$$

where $\alpha = \beta = 1$ by default. A higher IGC indicates better efficiency. As shown in **Table 20**, FeDaL achieves the highest IGC, outperforming all Time-MoE variants in terms of cost-effectiveness: **FeDaL** > **Time-MoE$_{base}$** > **Time-MoE$_{large}$** > **Time-MoE$_{ultra}$**. This underscores that FeDaL not only delivers strong performance, but does so with superior parameter and data efficiency, making it a more scalable and practical choice for real-world deployment. In addition, the TSFM trained with FeDaL demonstrates strong generalization across diverse tasks beyond forecasting, including classification, imputation, and anomaly detection.

Table 20: Zero-shot long-term forecasting performance comparison with larger-scale time series forecasting foundation models. A lower MSE or MAE indicates a better prediction. A higher IGC indicates better efficiency. **Bold**: the best, Underline: the second best.

| Models | | Comparison with Larger Forecasting Models | | | | | | | |
| | | FeDaL (Ours) | | Time-MoE$_{base}$ | | Time-MoE$_{large}$ | | Time-MoE$_{ultra}$ | |
| Metrics | | MSE | MAE | MSE | MAE | MSE | MAE | MSE | MAE |
|---|---|---|---|---|---|---|---|---|---|
| ETTh1 | 96 | **0.347** | 0.381 | 0.357 | 0.381 | 0.350 | 0.382 | 0.349 | **0.379** |
| | 192 | 0.398 | 0.410 | **0.384** | **0.404** | 0.388 | 0.412 | 0.395 | 0.413 |
| | 336 | 0.425 | 0.452 | **0.411** | 0.434 | **0.411** | **0.430** | 0.447 | 0.453 |
| | 720 | 0.457 | 0.469 | 0.449 | 0.477 | **0.427** | **0.455** | 0.457 | 0.462 |
| | Avg. | 0.407 | 0.429 | 0.400 | 0.424 | **0.394** | **0.419** | 0.412 | 0.426 |
| ETTh2 | 96 | 0.307 | 0.355 | 0.305 | 0.359 | 0.302 | 0.354 | **0.292** | **0.352** |
| | 192 | 0.349 | **0.372** | 0.351 | 0.386 | 0.364 | 0.385 | **0.347** | 0.379 |
| | 336 | **0.387** | **0.395** | 0.391 | 0.418 | 0.417 | 0.425 | 0.406 | 0.419 |
| | 720 | **0.401** | **0.406** | 0.419 | 0.454 | 0.537 | 0.496 | 0.439 | 0.447 |
| | Avg. | **0.361** | **0.382** | 0.366 | 0.404 | 0.405 | 0.415 | 0.371 | 0.399 |
| ETTm1 | 96 | 0.289 | 0.346 | 0.338 | 0.368 | 0.309 | 0.557 | **0.281** | **0.341** |
| | 192 | 0.317 | 0.369 | 0.353 | 0.388 | 0.346 | 0.381 | **0.305** | **0.358** |
| | 336 | 0.370 | 0.420 | 0.381 | 0.413 | 0.373 | 0.408 | **0.369** | **0.395** |
| | 720 | **0.464** | **0.426** | 0.504 | 0.493 | 0.475 | 0.477 | 0.469 | 0.472 |
| | Avg. | 0.360 | **0.390** | 0.394 | 0.415 | 0.376 | 0.405 | **0.356** | 0.391 |
| ETTm2 | 96 | 0.207 | **0.283** | 0.201 | 0.291 | **0.197** | 0.286 | 0.198 | 0.288 |
| | 192 | 0.248 | 0.333 | 0.258 | 0.334 | 0.250 | 0.322 | **0.235** | **0.312** |
| | 336 | 0.316 | **0.340** | 0.324 | 0.373 | 0.337 | 0.375 | **0.293** | 0.348 |
| | 720 | **0.397** | **0.408** | 0.488 | 0.464 | 0.480 | 0.461 | 0.427 | 0.423 |
| | Avg. | 0.292 | **0.341** | 0.317 | 0.365 | 0.316 | 0.361 | **0.288** | 0.344 |
| Weather | 96 | 0.159 | 0.212 | 0.160 | 0.214 | 0.159 | 0.213 | **0.157** | **0.211** |
| | 192 | 0.217 | 0.264 | 0.210 | 0.260 | 0.215 | 0.266 | **0.208** | **0.256** |
| | 336 | 0.285 | 0.312 | 0.274 | 0.309 | 0.291 | 0.322 | **0.255** | **0.290** |
| | 720 | 0.359 | 0.348 | 0.418 | 0.405 | 0.415 | 0.400 | 0.405 | 0.397 |
| | Avg. | 0.255 | 0.284 | 0.265 | 0.297 | 0.270 | 0.300 | 0.256 | 0.288 |
| **Average** | | **0.335** | 0.370 | 0.343 | 0.382 | 0.355 | 0.387 | 0.342 | 0.369 |
| **1$^{st}$ Count** | | 21 | | 3 | | 7 | | **22** | |
| **Total Param.#** | | **28.42 M** | | 113 M | | 453 M | | 2.4 B | |
| **Training Data** | | $\sim$**231B** | | 300B | | 300B | | 300B | |
| **Information Gain Per Cost** | | $4.545 \times 10^{-19}$ | | $8.605 \times 10^{-20}$ | | $2.073 \times 10^{-20}$ | | $4.061 \times 10^{-21}$ | |

## C.3 SHORT-TERM FORECASTING

The full short-term forecasting results are presented in **Table 21** and **Table 22**. Specifically, **Table 21** compares our FeDaL-pretrained TSFM with advanced deep time series models, while **Table 22** focuses on comparisons among FL-based TSFM pretraining methods. Across all settings, FeDaL consistently outperforms all baselines, including task-specific deep models, general-purpose foundation models, and alternative federated pretraining strategies, highlighting its effectiveness in learning robust and generalizable temporal representations under decentralization.

Table 21: Short-term forecasting results. **Bold**: the best, Underline: the second best.

| Intervals | Methods | **FeDaL (Ours)** | Time-LLM | GPT4TS | TimesNet | PatchTST | N-HiTS | N-BEATS | ETS.* | LightTS | DLinear | FED.* | Stationary | Auto.* | In.* | Re.* |
|---|---|---|---|---|---|---|---|---|---|---|---|---|---|---|---|---|
| Yearly | SMAPE | **13.102** | 13.419 | 15.11 | 15.378 | 13.477 | 13.422 | 13.487 | 18.009 | 14.247 | 16.965 | 14.021 | 13.717 | 13.974 | 14.727 | 16.169 |
| | MASE | **2.812** | 3.005 | 3.565 | 3.554 | 3.019 | 3.056 | 3.036 | 4.487 | 3.109 | 4.283 | 3.036 | 3.078 | 3.134 | 3.418 | 3.800 |
| | OWA | **0.748** | 0.789 | 0.911 | 0.918 | 0.792 | 0.795 | 0.795 | 1.115 | 0.827 | 1.058 | 0.811 | 0.807 | 0.822 | 0.881 | 0.973 |
| Quarterly | SMAPE | **9.808** | 10.110 | 10.597 | 10.465 | 10.38 | 10.185 | 10.564 | 13.376 | 11.364 | 12.145 | 11.1 | 10.958 | 11.338 | 11.360 | 13.313 |
| | MASE | **1.112** | 1.178 | 1.253 | 1.227 | 1.233 | 1.18 | 1.252 | 1.906 | 1.328 | 1.520 | 1.35 | 1.325 | 1.365 | 1.401 | 1.775 |
| | OWA | **0.847** | 0.889 | 0.938 | 0.923 | 0.921 | 0.893 | 0.936 | 1.302 | 1.000 | 1.106 | 0.996 | 0.981 | 1.012 | 1.027 | 1.252 |
| Monthly | SMAPE | **12.124** | 12.980 | 13.258 | 13.513 | 12.959 | 13.059 | 13.089 | 14.588 | 14.014 | 13.514 | 14.403 | 13.917 | 13.958 | 14.062 | 20.128 |
| | MASE | **0.898** | 0.963 | 1.003 | 1.039 | 0.970 | 1.013 | 0.996 | 1.368 | 1.053 | 1.037 | 1.147 | 1.097 | 1.103 | 1.141 | 2.614 |
| | OWA | **0.820** | 0.903 | 0.931 | 0.957 | 0.905 | 0.929 | 0.922 | 1.149 | 0.981 | 0.956 | 1.038 | 0.998 | 1.002 | 1.024 | 1.927 |
| Others | SMAPE | **4.508** | 4.795 | 6.124 | 6.913 | 4.952 | 4.711 | 6.599 | 7.267 | 15.880 | 6.709 | 7.148 | 6.302 | 5.485 | 24.460 | 32.491 |
| | MASE | **2.890** | 3.178 | 4.116 | 4.507 | 3.347 | 3.054 | 4.43 | 5.240 | 11.434 | 4.953 | 4.041 | 4.064 | 3.865 | 20.960 | 33.355 |
| | OWA | **0.973** | 1.006 | 1.259 | 1.438 | 1.049 | 0.977 | 1.393 | 1.591 | 3.474 | 1.487 | 1.389 | 1.304 | 1.187 | 5.879 | 8.679 |
| Average | SMAPE | **11.412** | 11.983 | 12.69 | 12.88 | 12.059 | 12.035 | 12.25 | 14.718 | 13.525 | 13.639 | 13.16 | 12.780 | 12.909 | 14.086 | 18.200 |
| | MASE | **1.489** | 1.595 | 1.808 | 1.836 | 1.623 | 1.625 | 1.698 | 2.408 | 2.111 | 2.095 | 1.775 | 1.756 | 1.771 | 2.718 | 4.223 |
| | OWA | **0.818** | 0.859 | 0.94 | 0.955 | 0.869 | 0.869 | 0.896 | 1.172 | 1.051 | 1.051 | 0.949 | 0.930 | 0.939 | 1.230 | 1.775 |

Table 22: Short-term forecasting results based on FL. **Bold**: the best, Underline: the second best.

| Setting | | Federated Foundation Models | | | Foundation Models |
|---|---|---|---|---|---|
| Intervals | Methods | **FeDaL (Ours)** | FFTS | FedAvg | MOMENT |
| Yearly | SMAPE | **13.102** | 13.289 | 14.784 | 20.649 |
| | MASE | **2.812** | 2.909 | 3.257 | 4.757 |
| | OWA | **0.748** | 0.781 | 0.866 | 1.230 |
| Quarterly | SMAPE | **9.808** | 10.005 | 10.920 | 10.849 |
| | MASE | **1.112** | 1.190 | 1.367 | 1.305 |
| | OWA | **0.847** | 0.877 | 0.957 | 0.968 |
| Monthly | SMAPE | 12.124 | **11.920** | 13.048 | 14.497 |
| | MASE | 0.898 | **0.879** | 1.027 | 1.143 |
| | OWA | 0.820 | **0.815** | 0.916 | 1.040 |
| Others | SMAPE | 4.508 | **4.490** | 5.210 | 5.634 |
| | MASE | **2.890** | 2.907 | 3.657 | 4.102 |
| | OWA | **0.973** | 0.994 | 1.154 | 1.240 |
| Average | SMAPE | 11.412 | **11.404** | 12.342 | 14.593 |
| | MASE | **1.489** | 1.522 | 1.753 | 2.161 |
| | OWA | **0.818** | 0.831 | 0.926 | 1.103 |

Table 23: Full results on imputation task. **Bold**: the best, Underline: the second best.

| Dataset | Metric | FeDaL (Ours) MSE | MAE | FFTS MSE | MAE | FedAvg MSE | MAE | GPT4TS MSE | MAE | DLinear MSE | MAE | PatchTST MSE | MAE | TimesNet MSE | MAE | FED. MSE | MAE | Auto. MSE | MAE | Stationary MSE | MAE | ETS. MSE | MAE | LightTS MSE | MAE | In. MSE | MAE | Re. MSE | MAE |
|---|---|---|---|---|---|---|---|---|---|---|---|---|---|---|---|---|---|---|---|---|---|---|---|---|---|---|---|---|---|
| ETTh1 | 12.5% | **0.013** | **0.073** | 0.015 | 0.075 | 0.020 | 0.105 | 0.017 | 0.085 | 0.023 | 0.101 | 0.041 | 0.130 | 0.096 | 0.229 | 0.093 | 0.206 | 0.080 | 0.193 | 0.052 | 0.166 | 0.032 | 0.119 | 0.046 | 0.144 | 0.063 | 0.180 | 0.042 | 0.146 |
| | 25% | **0.013** | **0.071** | 0.015 | 0.081 | 0.022 | 0.109 | 0.022 | 0.096 | 0.023 | 0.101 | 0.044 | 0.135 | 0.096 | 0.229 | 0.093 | 0.206 | 0.080 | 0.193 | 0.052 | 0.166 | 0.032 | 0.119 | 0.046 | 0.144 | 0.063 | 0.180 | 0.042 | 0.146 |
| | 37.5% | 0.033 | **0.091** | 0.036 | 0.094 | 0.046 | 0.131 | **0.029** | 0.111 | **0.029** | 0.111 | 0.049 | 0.143 | 0.133 | 0.271 | 0.113 | 0.231 | 0.103 | 0.219 | 0.069 | 0.191 | 0.039 | 0.131 | 0.057 | 0.161 | 0.079 | 0.200 | 0.063 | 0.182 |
| | 50% | **0.030** | **0.110** | **0.030** | 0.111 | 0.048 | 0.135 | 0.040 | 0.128 | 0.036 | 0.124 | 0.055 | 0.151 | 0.186 | 0.323 | 0.134 | 0.255 | 0.132 | 0.248 | 0.089 | 0.218 | 0.047 | 0.145 | 0.067 | 0.174 | 0.093 | 0.218 | 0.082 | 0.208 |
| | Avg | **0.022** | **0.090** | 0.024 | 0.093 | 0.034 | 0.120 | 0.028 | 0.105 | 0.027 | 0.107 | 0.047 | 0.140 | 0.120 | 0.253 | 0.104 | 0.218 | 0.093 | 0.206 | 0.062 | 0.177 | 0.036 | 0.126 | 0.051 | 0.150 | 0.071 | 0.188 | 0.055 | 0.166 |
| ETTh2 | 12.5% | **0.014** | **0.062** | **0.014** | 0.070 | 0.020 | 0.085 | 0.017 | 0.076 | 0.018 | 0.080 | 0.026 | 0.094 | 0.108 | 0.239 | 0.034 | 0.127 | 0.062 | 0.166 | 0.056 | 0.159 | 0.021 | 0.088 | 0.023 | 0.092 | 0.133 | 0.270 | 0.108 | 0.228 |
| | 25% | **0.017** | **0.070** | **0.017** | 0.074 | 0.024 | 0.090 | 0.020 | 0.080 | 0.020 | 0.085 | 0.028 | 0.099 | 0.164 | 0.294 | 0.042 | 0.143 | 0.085 | 0.196 | 0.080 | 0.195 | 0.024 | 0.096 | 0.026 | 0.101 | 0.135 | 0.272 | 0.136 | 0.262 |
| | 37.5% | 0.018 | **0.070** | **0.017** | 0.078 | 0.028 | 0.106 | 0.022 | 0.087 | 0.023 | 0.091 | 0.030 | 0.104 | 0.237 | 0.356 | 0.051 | 0.159 | 0.106 | 0.222 | 0.110 | 0.231 | 0.027 | 0.103 | 0.030 | 0.108 | 0.155 | 0.293 | 0.175 | 0.300 |
| | 50% | 0.022 | **0.079** | **0.019** | 0.086 | 0.032 | 0.111 | 0.025 | 0.095 | 0.026 | 0.098 | 0.034 | 0.110 | 0.323 | 0.421 | 0.059 | 0.174 | 0.131 | 0.247 | 0.156 | 0.276 | 0.030 | 0.108 | 0.035 | 0.119 | 0.200 | 0.333 | 0.211 | 0.329 |
| | Avg | 0.018 | **0.071** | **0.017** | 0.074 | 0.026 | 0.098 | 0.021 | 0.084 | 0.022 | 0.088 | 0.029 | 0.102 | 0.208 | 0.327 | 0.046 | 0.151 | 0.096 | 0.208 | 0.101 | 0.215 | 0.026 | 0.099 | 0.029 | 0.105 | 0.156 | 0.292 | 0.157 | 0.280 |
| ETTm1 | 12.5% | **0.030** | **0.116** | 0.034 | 0.132 | 0.038 | 0.132 | 0.043 | 0.140 | 0.057 | 0.159 | 0.093 | 0.201 | 0.126 | 0.263 | 0.240 | 0.345 | 0.151 | 0.267 | 0.070 | 0.190 | 0.060 | 0.165 | 0.074 | 0.182 | 0.114 | 0.234 | 0.074 | 0.194 |
| | 25% | **0.038** | **0.130** | 0.041 | 0.147 | 0.046 | 0.142 | 0.054 | 0.156 | 0.069 | 0.178 | 0.107 | 0.217 | 0.169 | 0.304 | 0.265 | 0.364 | 0.180 | 0.292 | 0.106 | 0.236 | 0.080 | 0.189 | 0.090 | 0.203 | 0.140 | 0.262 | 0.102 | 0.227 |
| | 37.5% | 0.061 | **0.150** | 0.065 | 0.160 | **0.058** | 0.162 | 0.072 | 0.180 | 0.084 | 0.196 | 0.120 | 0.230 | 0.296 | 0.347 | 0.296 | 0.382 | 0.215 | 0.318 | 0.124 | 0.258 | 0.102 | 0.212 | 0.109 | 0.222 | 0.174 | 0.293 | 0.135 | 0.261 |
| | 50% | 0.086 | 0.192 | 0.091 | 0.202 | **0.074** | **0.180** | 0.107 | 0.216 | 0.102 | 0.215 | 0.141 | 0.248 | 0.293 | 0.402 | 0.334 | 0.404 | 0.257 | 0.347 | 0.165 | 0.299 | 0.133 | 0.240 | 0.137 | 0.248 | 0.215 | 0.325 | 0.179 | 0.298 |
| | Avg | **0.054** | **0.147** | 0.058 | 0.160 | **0.054** | 0.154 | 0.069 | 0.173 | 0.078 | 0.187 | 0.115 | 0.224 | 0.202 | 0.329 | 0.284 | 0.373 | 0.201 | 0.306 | 0.117 | 0.246 | 0.094 | 0.201 | 0.103 | 0.214 | 0.161 | 0.279 | 0.122 | 0.245 |
| ETTm2 | 12.5% | **0.020** | **0.100** | 0.028 | 0.102 | 0.045 | 0.128 | 0.039 | 0.125 | 0.040 | 0.130 | 0.057 | 0.152 | 0.187 | 0.319 | 0.101 | 0.231 | 0.100 | 0.216 | 0.095 | 0.212 | 0.042 | 0.133 | 0.044 | 0.138 | 0.305 | 0.431 | 0.163 | 0.289 |
| | 25% | **0.021** | **0.104** | 0.034 | 0.133 | 0.052 | 0.153 | 0.044 | 0.135 | 0.046 | 0.141 | 0.061 | 0.158 | 0.279 | 0.390 | 0.115 | 0.246 | 0.127 | 0.247 | 0.137 | 0.258 | 0.049 | 0.147 | 0.050 | 0.149 | 0.322 | 0.444 | 0.206 | 0.331 |
| | 37.5% | **0.048** | **0.110** | 0.063 | 0.153 | 0.077 | 0.175 | 0.051 | 0.147 | 0.052 | 0.151 | 0.067 | 0.166 | 0.400 | 0.465 | 0.126 | 0.257 | 0.158 | 0.276 | 0.187 | 0.304 | 0.056 | 0.158 | 0.060 | 0.163 | 0.353 | 0.462 | 0.252 | 0.370 |
| | 50% | 0.048 | **0.118** | **0.033** | 0.151 | 0.075 | 0.160 | 0.059 | 0.158 | 0.060 | 0.162 | 0.073 | 0.174 | 0.602 | 0.572 | 0.136 | 0.268 | 0.183 | 0.299 | 0.232 | 0.341 | 0.065 | 0.170 | 0.068 | 0.173 | 0.369 | 0.472 | 0.316 | 0.419 |
| | Avg | **0.034** | **0.108** | 0.046 | 0.135 | 0.062 | 0.154 | 0.048 | 0.141 | 0.049 | 0.146 | 0.065 | 0.163 | 0.367 | 0.436 | 0.119 | 0.250 | 0.142 | 0.259 | 0.163 | 0.279 | 0.053 | 0.152 | 0.055 | 0.156 | 0.337 | 0.452 | 0.234 | 0.352 |
| Weather | 12.5% | **0.013** | **0.036** | 0.020 | 0.053 | 0.020 | 0.053 | 0.026 | 0.049 | 0.025 | 0.045 | 0.029 | 0.049 | 0.057 | 0.141 | 0.047 | 0.101 | 0.039 | 0.084 | 0.041 | 0.107 | 0.027 | 0.051 | 0.026 | 0.047 | 0.037 | 0.093 | 0.031 | 0.076 |
| | 25% | **0.018** | **0.025** | 0.024 | 0.057 | 0.024 | 0.057 | 0.028 | 0.052 | 0.029 | 0.052 | 0.031 | 0.053 | 0.065 | 0.155 | 0.052 | 0.111 | 0.048 | 0.103 | 0.064 | 0.163 | 0.029 | 0.056 | 0.030 | 0.054 | 0.042 | 0.100 | 0.035 | 0.082 |
| | 37.5% | **0.025** | 0.060 | 0.034 | 0.063 | 0.034 | 0.063 | 0.033 | 0.060 | **0.031** | 0.057 | 0.035 | 0.058 | 0.081 | 0.180 | 0.058 | 0.121 | 0.057 | 0.117 | 0.107 | 0.229 | 0.033 | 0.062 | 0.032 | 0.060 | 0.049 | 0.111 | 0.040 | 0.091 |
| | 50% | **0.027** | **0.050** | 0.038 | 0.061 | 0.038 | 0.061 | 0.037 | 0.065 | 0.034 | 0.062 | 0.038 | 0.063 | 0.102 | 0.207 | 0.065 | 0.133 | 0.066 | 0.134 | 0.183 | 0.312 | 0.037 | 0.068 | 0.037 | 0.067 | 0.053 | 0.114 | 0.038 | 0.087 |
| | Avg | **0.024** | **0.048** | 0.029 | 0.058 | 0.034 | 0.050 | 0.031 | 0.056 | 0.030 | 0.054 | 0.060 | 0.144 | 0.076 | 0.171 | 0.055 | 0.117 | 0.052 | 0.110 | 0.099 | 0.203 | 0.032 | 0.059 | 0.031 | 0.057 | 0.045 | 0.104 | 0.038 | 0.087 |
| 1st Count | | **40** | | 7 | | 4 | | 1 | | 2 | | 0 | | 0 | | 0 | | 0 | | 0 | | 0 | | 0 | | 0 | | 0 | |

## C.4 Time Series Imputation

The full imputation results are presented in **Table 23**. Our proposed FeDaL consistently outperforms both advanced deep time series models and federated TSFM pretraining baselines, demonstrating superior generalization across heterogeneous input gaps.

## C.5 Time Series Anomaly Detection

The full anomaly detection results are presented in **Table 24**. FeDaL again achieves the best performance among all evaluated methods, surpassing both advanced deep time series models and existing federated TSFM pretraining approaches.

## C.6 Time Series Classification

The full classification results are presented in **Table 25 (UCR)** and **Table 26 (UEA)**. FeDaL consistently outperforms all baselines, including deep time series models, classical machine learning and statistical classifiers, general-purpose TSFMs (GPT4TS, MOMENT), and federated TSFM pretraining methods.

Table 24: Full results of the anomaly detection task. The P, R, and F1 represent the precision, recall, and F1-score (%) respectively. F1-score is the harmonic mean of precision and recall. A higher value of P, R, and F1 indicates a better performance. **Bold**: the best, Underline: the second best.

| Dataset | | SMD | | | MSL | | | SMAP | | | SWaT | | | PSM | | | Avg. F1 |
|---|---|---|---|---|---|---|---|---|---|---|---|---|---|---|---|---|---|---|
| Metric | | P | R | F1 | P | R | F1 | P | R | F1 | P | R | F1 | P | R | F1 | (%) |
| LSTM | (1997) | 78.52 | 65.47 | 71.41 | 78.04 | 86.22 | 81.93 | 91.06 | 57.49 | 70.48 | 78.06 | 91.72 | 84.34 | 69.24 | 99.53 | 81.67 | 77.97 |
| Transformer | (2017) | 83.58 | 76.13 | 79.56 | 71.57 | 87.37 | 78.68 | 89.37 | 57.12 | 69.70 | 68.84 | 96.53 | 80.37 | 62.75 | 96.56 | 76.07 | 76.88 |
| LogTrans | (2019) | 83.46 | 70.13 | 76.21 | 73.05 | 87.37 | 79.57 | 89.15 | 57.59 | 69.97 | 68.67 | 97.32 | 80.52 | 63.06 | 98.00 | 76.74 | 76.60 |
| TCN | (2019) | 84.06 | 79.07 | 81.49 | 75.11 | 82.44 | 78.60 | 86.90 | 59.23 | 70.45 | 76.59 | 95.71 | 85.09 | 54.59 | 99.77 | 70.57 | 77.24 |
| Reformer | (2020) | 82.58 | 69.24 | 75.32 | 85.51 | 83.31 | 84.40 | 90.91 | 57.44 | 70.40 | 72.50 | 96.53 | 82.80 | 59.93 | 95.38 | 73.61 | 77.31 |
| Informer | (2021) | 86.60 | 77.23 | 81.65 | 81.77 | 86.48 | 84.06 | 90.11 | 57.13 | 69.92 | 70.29 | 96.75 | 81.43 | 64.27 | 96.33 | 77.10 | 78.83 |
| Anomaly* | (2022) | 88.91 | 82.23 | 85.49 | 79.61 | 87.37 | 83.31 | 91.85 | 58.11 | 71.18 | 72.51 | 97.32 | 83.10 | 68.35 | 94.72 | 79.40 | 80.50 |
| Pyraformer | (2021) | 85.61 | 80.61 | 83.04 | 83.81 | 85.93 | 84.86 | 92.54 | 57.71 | 71.09 | 87.92 | 96.00 | 91.78 | 71.67 | 96.02 | 82.08 | 82.57 |
| Autoformer | (2021) | 88.06 | 82.35 | 85.11 | 77.27 | 80.92 | 79.05 | 90.40 | 58.62 | 71.12 | 89.85 | 95.81 | 92.74 | 99.08 | 88.15 | 93.29 | 84.26 |
| LSSL | (2021) | 78.51 | 65.32 | 71.31 | 77.55 | 88.18 | 82.53 | 89.43 | 53.43 | 66.90 | 79.05 | 93.72 | 85.76 | 66.02 | 92.93 | 77.20 | 76.74 |
| NSformer | (2022) | 88.33 | 81.21 | 84.62 | 68.55 | 89.14 | 77.50 | 89.37 | 59.02 | 71.09 | 68.03 | 96.75 | 79.88 | 97.82 | 96.76 | 97.29 | 82.08 |
| DLinear | (2023) | 83.62 | 71.52 | 77.10 | 84.34 | 85.42 | 84.88 | 92.32 | 55.41 | 69.26 | 80.91 | 95.30 | 87.52 | 98.28 | 89.26 | 93.55 | 82.46 |
| ETSformer | (2022) | 87.44 | 79.23 | 83.13 | 85.13 | 84.93 | 85.03 | 92.25 | 55.75 | 69.50 | 90.02 | 80.36 | 84.91 | 99.31 | 85.28 | 91.76 | 82.87 |
| LightTS | (2022) | 87.10 | 78.42 | 82.53 | 82.40 | 75.78 | 78.95 | 92.58 | 55.27 | 69.21 | 91.98 | 94.72 | 93.33 | 98.37 | 95.97 | 97.15 | 84.23 |
| FEDformer | (2022) | 87.95 | 82.39 | 85.08 | 77.14 | 80.07 | 78.57 | 90.47 | 58.10 | 70.76 | 90.17 | 96.42 | 93.19 | 97.31 | 97.16 | 97.23 | 84.97 |
| TimesNet | (2022a) | 88.66 | 83.14 | 85.81 | 83.92 | 86.42 | 85.15 | 92.52 | 58.29 | 71.52 | 86.76 | 97.32 | 91.74 | 98.19 | 96.76 | 97.47 | 86.34 |
| ModernTCN | (2024) | 87.86 | 83.85 | 85.81 | 83.94 | 85.93 | 84.92 | 93.17 | 57.69 | 71.26 | 91.83 | 95.98 | 93.86 | 98.09 | 96.38 | 97.23 | 86.62 |
| MOMENT | (2024) | 78.88 | 92.01 | 84.94 | 88.98 | 75.10 | 81.45 | 90.02 | 56.51 | 69.43 | 92.13 | 91.67 | 91.90 | 98.82 | 89.55 | 93.96 | 84.34 |
| GPT4TS | (2023) | 88.89 | 84.98 | 86.89 | 82.00 | 82.91 | 82.45 | 90.60 | 60.95 | 72.88 | 92.20 | 96.34 | 94.23 | 98.62 | 95.68 | 97.13 | 86.72 |
| FedAvg | (2017) | 87.88 | 89.21 | 88.44 | 81.69 | 83.00 | 82.32 | 89.26 | 58.66 | 70.78 | 90.27 | 90.43 | 90.32 | 96.56 | 95.08 | 95.86 | 85.54 |
| FFTS | (2025) | 89.26 | 90.48 | 89.88 | 89.23 | 87.64 | 88.42 | 90.64 | 58.88 | 71.38 | 91.22 | 90.99 | 91.12 | 99.00 | 97.85 | 98.54 | 87.86 |
| **FeDaL** | **(Ours)** | 89.15 | 87.78 | **88.46** | 88.87 | 89.23 | **89.05** | 88.46 | 60.29 | **71.70** | 92.56 | 98.49 | **95.40** | 99.29 | 98.46 | **98.88** | **88.70** |

## C.7 FEDERATED SCALING BEHAVIORS

While prior work explores scaling laws of centralized TSFMs in terms of model and data size, we instead investigate **scaling behaviors** in the federated setting, focusing on three key factors: (i) pre-training data size, (ii) number of clients, and (iii) client participation rate. Specifically, we vary: (i) Data size from $\{40B, 80B, 120B, 160B, 200B, 231B\}$ with a fixed client count of 174; (ii) Number of clients from $\{30, 70, 110, 140, 174\}$ using the full 231B dataset; (iii) Client participation rate from $\{10\%, 30\%, 50\%, 70\%, 100\%\}$ with the original pretraining setup. All experiments follow the training protocol in Section 4.3. Results are presented in **Figure 10**. We observe that: (1) Increasing data size consistently improves downstream performance, even with a fixed client count; (2) More clients (with constant total data size) lead to better representations, suggesting improved learning of diverse, domain-specific patterns; (3) Higher client participation rates yield stronger results, likely due to more effective aggregation and reduced drift. These findings suggest that federated TSFM pretraining benefits from scaling in data and client dimensions, much like centralized pretraining. However, unlike the "scale-is-all" mindset focused on model size, our results highlight the importance of expanding dataset coverage and client participation as more efficient and federated-aligned strategies for improving generalization.

## C.8 VARIANCE ESTIMATES

All experiments (except the zero-shot forecasting setting) were conducted three times, and we report the averaged results. Variance estimates for representative configurations are now provided in **Table 27-Table 30.**

## THE USE OF LARGE LANGUAGE MODELS

In preparing this paper, we used a Large Language Model as a general-purpose writing assistant. Its role was limited to language refinement, such as improving grammar, sentence flow, and vocabulary usage to meet academic writing standards. All substantive aspects of the research (including conceptualization, experimental design, data analysis, and drawing conclusions) were carried out

Table 25: Full results of the classification task on UCR Archive Dau et al. (2019). **Bold**: the best, Underline: the second best.

| Datasets | DTW | TS2Vec | T-Loss | TNC | TS-TCC | TST | CNN | Encoder | FCN | MCDNN | ResNet | TimesNet | MTCN | t-LeNet | TWIESN | MLP | DLinear | PatchTST | MOMENT | GPT4TS | FedAvg | FFTS | **FeDaL** |
|---|---|---|---|---|---|---|---|---|---|---|---|---|---|---|---|---|---|---|---|---|---|---|---|
| AllGestureWiimoteX | 71.6 | 77.7 | 76.3 | 69.7 | 69.7 | 25.9 | 41.1 | 47.5 | 71.3 | 26.1 | 74.1 | 52.7 | 49.1 | 10.0 | 52.2 | 47.7 | 31.7 | 55.6 | 71.7 | 23.7 | 78.2 | 70.1 | 74.6 |
| AllGestureWiimoteY | 72.9 | 79.3 | 72.6 | 74.1 | 74.1 | 42.3 | 47.9 | 50.9 | 78.4 | 42.0 | 79.4 | 56.3 | 50.1 | 10.0 | 60.0 | 57.1 | 34.1 | 62.3 | 76.7 | 16.0 | 79.5 | 72.6 | 73.5 |
| AllGestureWiimoteZ | 64.3 | 74.6 | 72.3 | 68.9 | 68.9 | 44.7 | 37.5 | 39.6 | 69.2 | 28.7 | 72.6 | 51.1 | 48.0 | 10.0 | 51.6 | 43.9 | 33.9 | 52.7 | 72.0 | 11.6 | 68.8 | 62.2 | 68.0 |
| ArrowHead | 70.3 | 85.7 | 76.6 | 73.7 | 73.7 | 77.1 | 71.7 | 63.0 | 84.3 | 67.8 | 83.8 | 78.9 | 76.6 | 30.3 | 68.9 | 78.4 | 66.9 | 66.3 | 86.9 | 42.9 | 81.1 | 79.7 | 90.4 |
| BME | 90.0 | 99.3 | 99.3 | 93.3 | 93.3 | 76.0 | 94.7 | 82.7 | 83.6 | 89.6 | 99.9 | 80.0 | 66.7 | 33.3 | 81.9 | 90.5 | 70.0 | 90.0 | 90.0 | 36.7 | 94.0 | 92.8 | 94.4 |
| Beef | 63.3 | 76.7 | 66.7 | 60.0 | 60.0 | 60.0 | 50.0 | 76.7 | 70.7 | 68.0 | 50.7 | 75.3 | 90.0 | 20.0 | 52.7 | 71.3 | 90.0 | 73.3 | 90.0 | 16.7 | 82.3 | 83.3 | 90.0 |
| BeetleFly | 70.0 | 90.0 | 80.0 | 80.0 | 80.0 | 100.0 | 90.0 | 62.0 | 91.0 | 63.0 | 85.0 | 70.0 | 70.0 | 50.0 | 79.0 | 88.0 | 80.0 | 75.0 | 75.0 | 70.0 | 88.6 | 87.5 | 95.0 |
| BirdChicken | 75.0 | 80.0 | 85.0 | 65.0 | 65.0 | 65.0 | 71.0 | 51.0 | 94.0 | 54.0 | 88.0 | 93.3 | 80.0 | 50.0 | 62.0 | 74.0 | 90.0 | 80.0 | 99.3 | 55.0 | 85.2 | 87.1 | 95.0 |
| CBF | 99.7 | 100.0 | 98.3 | 99.8 | 99.8 | 89.8 | 95.9 | 97.7 | 99.4 | 90.8 | 99.6 | 96.1 | 83.4 | 33.2 | 89.6 | 86.9 | 77.8 | 96.3 | 97.9 | 83.0 | 95.1 | 93.8 | 96.2 |
| Chinatown | 95.7 | 96.5 | 95.1 | 98.3 | 98.3 | 93.6 | 97.7 | 96.6 | 98.0 | 94.5 | 97.8 | 98.3 | 96.5 | 72.6 | 82.5 | 87.2 | 83.1 | 98.3 | 98.5 | 85.7 | 96.8 | 98.2 | 99.0 |
| ChlorineConcentration | 64.8 | 83.2 | 74.9 | 75.3 | 75.3 | 56.2 | 60.8 | 58.3 | 81.7 | 66.2 | 85.3 | 69.2 | 62.6 | 53.3 | 55.4 | 80.0 | 58.8 | 62.2 | 73.9 | 56.5 | 72.4 | 66.6 | 65.2 |
| Coffee | 100.0 | 100.0 | 100.0 | 100.0 | 100.0 | 82.1 | 100.0 | 88.6 | 100.0 | 97.9 | 100.0 | 92.9 | 100.0 | 50.7 | 97.9 | 99.3 | 100.0 | 100.0 | 100.0 | 67.9 | 100.0 | 100.0 | 100.0 |
| CricketX | 75.4 | 78.2 | 71.3 | 73.1 | 73.1 | 38.5 | 53.5 | 64.4 | 79.4 | 51.3 | 79.9 | 58.5 | 54.4 | 7.4 | 62.7 | 59.1 | 32.1 | 64.6 | 77.4 | 53.1 | 76.5 | 70.4 | 78.2 |
| CricketY | 74.4 | 74.9 | 72.8 | 71.8 | 71.8 | 46.7 | 58.2 | 63.9 | 79.3 | 52.1 | 81.0 | 58.7 | 57.4 | 8.5 | 65.2 | 59.8 | 39.0 | 67.9 | 81.5 | 52.1 | 75.9 | 73.1 | 76.0 |
| CricketZ | 75.4 | 79.2 | 70.8 | 71.3 | 71.3 | 40.3 | 50.1 | 65.1 | 81.0 | 48.4 | 80.9 | 57.2 | 56.7 | 6.2 | 64.3 | 62.9 | 31.8 | 68.7 | 77.9 | 39.7 | 74.2 | 68.3 | 70.0 |
| Crop | 66.5 | 75.6 | 72.2 | 74.2 | 74.2 | 71.0 | 67.0 | 76.0 | 73.8 | 68.7 | 74.3 | 77.5 | 76.4 | 4.2 | 48.9 | 61.8 | 68.1 | 72.5 | 74.8 | 34.1 | 77.8 | 77.7 | 78.2 |
| DiatomSizeReduction | 96.7 | 98.4 | 98.4 | 97.7 | 97.7 | 96.1 | 95.4 | 88.0 | 34.6 | 64.6 | 30.1 | 95.8 | 87.9 | 30.1 | 91.4 | 90.9 | 88.9 | 88.6 | 96.7 | 98.7 | 98.0 | 98.9 | 98.7 |
| DistalPhalanxOutlineAgeGroup | 77.0 | 72.7 | 72.7 | 75.5 | 75.5 | 74.1 | 75.8 | 76.1 | 71.8 | 72.9 | 71.8 | 78.4 | 79.9 | 43.3 | 70.5 | 64.7 | 66.9 | 75.5 | 79.1 | 48.9 | 80.3 | 80.5 | 81.1 |
| DistalPhalanxOutlineCorrect | 71.7 | 76.1 | 77.5 | 75.4 | 75.4 | 72.8 | 77.2 | 72.4 | 76.0 | 75.9 | 77.0 | 77.5 | 78.6 | 58.3 | 71.1 | 72.7 | 69.9 | 75.7 | 76.4 | 65.9 | 78.7 | 75.9 | 79.0 |
| DistalPhalanxTW | 59.0 | 69.8 | 67.6 | 67.6 | 67.6 | 56.8 | 67.1 | 69.4 | 69.5 | 68.5 | 66.3 | 72.7 | 73.4 | 28.5 | 59.1 | 61.0 | 69.8 | 73.4 | 70.5 | 61.9 | 71.9 | 71.2 | 72.5 |
| DodgerLoopDay | 50.0 | 56.2 | NaN | NaN | NaN | 20.0 | 31.2 | 48.7 | 14.3 | 30.5 | 15.0 | 56.3 | 57.5 | 16.0 | 59.3 | 16.0 | 57.5 | 51.3 | 58.8 | 20.0 | 58.2 | 59.7 | 60.0 |
| DodgerLoopGame | 87.7 | 84.1 | NaN | NaN | NaN | 69.6 | 81.6 | 81.0 | 76.8 | 87.7 | 71.0 | 87.0 | 81.9 | 47.8 | 71.6 | 86.5 | 79.0 | 83.3 | 88.4 | 71.7 | 83.7 | 82.9 | 84.3 |
| DodgerLoopWeekend | 94.9 | 96.4 | NaN | NaN | NaN | 73.2 | 97.4 | 98.3 | 90.4 | 97.8 | 95.2 | 98.6 | 97.8 | 73.9 | 95.4 | 97.8 | 97.1 | 98.6 | 98.6 | 80.4 | 97.5 | 98.0 | 98.0 |
| ECG200 | 77.0 | 92.0 | 94.0 | 88.0 | 88.0 | 83.0 | 81.6 | 88.4 | 83.8 | 83.8 | 87.4 | 74.8 | 74.8 | 64.0 | 87.4 | 91.4 | 64.7 | 91.0 | 75.5 | 79.0 | 89.6 | 89.2 | 91.4 |
| ECG5000 | 92.4 | 93.5 | 93.3 | 94.1 | 94.1 | 92.8 | 92.8 | 94.1 | 94.0 | 93.3 | 93.5 | 89.0 | 86.0 | 58.4 | 92.2 | 93.0 | 83.0 | 94.1 | 94.0 | 58.4 | 93.1 | 91.8 | 93.8 |
| ECGFiveDays | 76.8 | 100.0 | 100.0 | 87.8 | 87.8 | 76.3 | 87.4 | 84.2 | 98.5 | 80.0 | 96.6 | 94.0 | 94.3 | 49.7 | 72.3 | 97.3 | 94.0 | 95.8 | 94.8 | 56.1 | 95.4 | 93.5 | 97.0 |
| Earthquakes | 71.9 | 74.8 | 74.8 | 74.8 | 74.8 | 74.8 | 70.9 | 74.0 | 72.5 | 74.8 | 71.2 | 90.0 | 88.2 | 74.8 | 74.8 | 72.7 | 94.3 | 74.8 | 100.0 | 74.8 | 73.8 | 71.0 | 74.2 |
| ElectricDevices | 60.2 | 72.1 | 70.7 | 68.6 | 68.6 | 67.6 | 68.6 | 70.2 | 70.6 | 65.3 | 72.8 | 70.9 | 63.8 | 24.2 | 60.5 | 59.3 | 47.6 | 64.8 | 69.0 | 50.0 | 77.2 | 70.5 | 78.5 |
| FaceAll | 80.8 | 77.1 | 78.6 | 81.3 | 81.3 | 50.4 | 77.4 | 79.4 | 93.8 | 72.0 | 86.7 | 76.9 | 75.3 | 8.0 | 67.3 | 79.4 | 82.4 | 76.2 | 78.3 | 14.7 | 90.1 | 91.0 | 92.0 |
| FaceFour | 83.0 | 93.2 | 92.0 | 77.3 | 77.3 | 51.1 | 90.5 | 85.2 | 93.0 | 71.1 | 95.5 | 89.8 | 54.5 | 29.5 | 85.7 | 83.6 | 79.5 | 83.0 | 89.8 | 65.9 | 92.3 | 90.0 | 95.0 |
| FacesUCR | 90.5 | 92.4 | 88.4 | 86.3 | 86.3 | 54.3 | 87.3 | 86.7 | 94.3 | 77.5 | 95.4 | 84.9 | 81.4 | 14.3 | 64.1 | 83.1 | 75.9 | 83.1 | 84.2 | 46.2 | 88.9 | 88.4 | 90.5 |
| FiftyWords | 69.0 | 77.1 | 73.2 | 65.3 | 65.3 | 52.5 | 62.4 | 65.8 | 64.6 | 61.1 | 74.0 | 66.2 | 69.5 | 12.5 | 51.8 | 70.8 | 59.1 | 71.4 | 80.0 | 49.2 | 70.5 | 70.0 | 72.0 |
| Fish | 82.3 | 92.6 | 89.1 | 81.7 | 81.7 | 72.0 | 85.5 | 73.4 | 96.1 | 72.0 | 98.1 | 84.0 | 85.7 | 12.6 | 87.8 | 84.8 | 82.3 | 86.3 | 92.0 | 73.1 | 97.8 | 98.1 | 99.4 |
| FordA | 55.5 | 93.6 | 92.8 | 93.0 | 93.0 | 56.8 | 89.6 | 92.8 | 91.4 | 86.3 | 93.7 | 92.9 | 94.3 | 51.0 | 55.5 | 81.6 | 51.8 | 93.1 | 94.5 | 91.4 | 91.5 | 91.2 | 92.8 |
| FordB | 62.0 | 79.4 | 79.3 | 81.5 | 81.5 | 50.7 | 74.9 | 77.7 | 77.2 | 69.8 | 81.3 | 77.0 | 80.4 | 50.3 | 51.2 | 70.7 | 52.8 | 79.3 | 82.3 | 67.7 | 78.9 | 77.1 | 79.6 |
| FreezerRegularTrain | 89.9 | 98.6 | 95.6 | 98.9 | 98.9 | 92.2 | 98.7 | 76.0 | 99.7 | 97.3 | 99.8 | 97.1 | 92.5 | 50.0 | 94.6 | 90.6 | 78.3 | 99.4 | 99.6 | 82.9 | 97.6 | 95.4 | 98.2 |
| FreezerSmallTrain | 75.3 | 87.0 | 93.3 | 97.9 | 97.9 | 92.0 | 73.9 | 67.6 | 68.3 | 68.8 | 83.2 | 76.5 | 75.8 | 50.0 | 91.7 | 68.6 | 76.7 | 76.4 | 83.5 | 50.0 | 76.8 | 76.7 | 79.9 |
| Fungi | 83.9 | 95.7 | 100.0 | 75.3 | 75.3 | 36.6 | 96.1 | 93.4 | 1.8 | 5.1 | 17.7 | 96.2 | 84.9 | 6.3 | 43.9 | 86.3 | 84.4 | 85.5 | 91.4 | 5.4 | 99.2 | 100.0 | 100.0 |
| GestureMidAirD1 | 56.9 | 60.8 | 60.8 | 36.9 | 36.9 | 20.8 | 53.4 | 52.8 | 69.5 | 51.8 | 69.8 | 71.5 | 72.3 | 3.8 | 54.9 | 57.5 | 57.7 | 63.1 | 71.5 | 29.2 | 78.4 | 80.0 | 80.2 |
| GestureMidAirD2 | 60.8 | 46.9 | 54.6 | 25.4 | 25.4 | 13.8 | 51.8 | 48.0 | 63.1 | 50.0 | 66.8 | 54.6 | 59.2 | 3.8 | 57.5 | 54.5 | 52.3 | 53.8 | 59.2 | 20.0 | 64.1 | 63.0 | 65.4 |
| GestureMidAirD3 | 32.3 | 29.2 | 28.5 | 17.7 | 17.7 | 15.4 | 31.7 | 36.8 | 32.6 | 27.8 | 34.0 | 42.3 | 46.9 | 3.8 | 27.5 | 38.2 | 36.2 | 41.5 | 50.0 | 16.2 | 52.3 | 52.0 | 53.8 |
| GesturePebbleZ1 | 79.1 | 93.0 | 91.9 | 39.5 | 39.5 | 50.0 | 84.4 | 82.1 | 88.0 | 76.9 | 90.1 | 86.6 | 82.0 | 16.3 | 84.3 | 79.2 | 70.9 | 86.0 | 89.5 | 60.5 | 87.1 | 88.0 | 88.5 |
| GesturePebbleZ2 | 67.1 | 87.3 | 89.9 | 43.0 | 43.0 | 38.0 | 77.8 | 79.6 | 78.1 | 72.0 | 77.7 | 84.2 | 77.2 | 18.4 | 84.3 | 70.1 | 61.4 | 85.4 | 91.1 | 28.5 | 89.6 | 87.0 | 91.2 |
| GunPoint | 90.7 | 98.0 | 98.0 | 99.3 | 99.3 | 82.7 | 94.8 | 78.4 | 100.0 | 90.7 | 99.1 | 95.3 | 86.0 | 49.3 | 98.9 | 92.8 | 87.0 | 84.0 | 100.0 | 84.7 | 95.2 | 95.0 | 96.0 |
| GunPointAgeSpan | 91.8 | 98.7 | 99.4 | 99.4 | 99.4 | 99.1 | 91.2 | 89.0 | 99.6 | 88.7 | 99.7 | 96.2 | 88.3 | 49.4 | 96.5 | 93.4 | 88.0 | 94.0 | 98.1 | 49.4 | 98.5 | 96.5 | 99.2 |
| GunPointMaleVersusFemale | 99.7 | 100.0 | 99.7 | 99.7 | 99.7 | 100.0 | 97.7 | 97.8 | 99.7 | 95.2 | 99.4 | 100.0 | 99.7 | 52.5 | 98.8 | 98.0 | 91.5 | 99.4 | 99.1 | 47.5 | 99.8 | 100.0 | 100.0 |
| GunPointOldVersusYoung | 83.8 | 100.0 | 100.0 | 100.0 | 100.0 | 100.0 | 92.2 | 92.3 | 98.9 | 92.6 | 98.9 | 100.0 | 100.0 | 52.4 | 97.5 | 94.1 | 100.0 | 92.1 | 97.5 | 52.4 | 99.5 | 100.0 | 100.0 |
| Ham | 46.7 | 71.4 | 72.4 | 74.3 | 74.3 | 52.4 | 72.0 | 68.2 | 70.7 | 71.8 | 75.8 | 77.1 | 79.0 | 51.4 | 76.8 | 69.9 | 80.0 | 83.8 | 78.1 | 78.1 | 77.6 | 72.8 | 79.5 |
| Herring | 53.1 | 64.1 | 59.4 | 59.4 | 59.4 | 59.4 | 53.1 | 51.2 | 64.4 | 57.2 | 60.0 | 62.5 | 67.2 | 59.4 | 62.5 | 49.1 | 64.1 | 67.2 | 70.3 | 57.8 | 68.9 | 70.7 | 79.2 |
| InsectWingbeatSound | 35.5 | 63.0 | 59.7 | 41.5 | 41.5 | 26.6 | 58.5 | 63.0 | 39.2 | 58.7 | 49.9 | 63.6 | 65.8 | 9.1 | 43.5 | 60.4 | 63.4 | 65.1 | 65.9 | 59.8 | 66.7 | 61.4 | 68.4 |
| ItalyPowerDemand | 95.0 | 92.5 | 95.4 | 95.5 | 95.5 | 84.5 | 95.4 | 96.4 | 96.3 | 96.6 | 96.2 | 96.9 | 96.1 | 49.9 | 87.1 | 95.3 | 94.2 | 97.2 | 95.8 | 88.0 | 94.5 | 95.4 | 94.1 |
| Lightning7 | 72.6 | 86.3 | 79.5 | 68.5 | 68.5 | 41.1 | 64.7 | 69.6 | 82.5 | 55.9 | 82.7 | 80.8 | 68.5 | 26.0 | 60.8 | 61.6 | 61.1 | 72.6 | 82.2 | 56.2 | 86.3 | 87.2 | 89.2 |
| Meat | 93.3 | 95.0 | 95.0 | 88.3 | 88.3 | 90.0 | 91.3 | 78.7 | 80.3 | 78.7 | 99.0 | 86.7 | 60.0 | 33.3 | 97.0 | 89.3 | 96.7 | 66.7 | 96.7 | 16.7 | 96.2 | 88.0 | 98.5 |
| MedicalImages | 73.7 | 78.9 | 75.0 | 74.7 | 74.7 | 63.2 | 67.1 | 66.4 | 77.8 | 62.7 | 77.0 | 71.2 | 72.8 | 51.4 | 64.9 | 71.9 | 56.8 | 75.4 | 75.8 | 49.6 | 76.5 | 77.4 | 78.0 |
| MelbournePedestrian | 79.1 | 95.9 | 94.4 | 94.9 | 94.9 | 74.1 | 81.3 | 88.4 | 91.2 | 84.0 | 90.9 | 96.4 | 91.1 | 10.0 | 73.0 | 86.3 | 83.9 | 88.7 | 88.3 | 20.7 | 95.1 | 86.2 | 96.7 |
| MiddlePhalanxOutlineAgeGroup | 50.0 | 63.6 | 65.6 | 63.0 | 63.0 | 61.7 | 53.4 | 57.7 | 53.5 | 55.8 | 54.5 | 63.6 | 63.0 | 57.1 | 57.8 | 52.2 | 65.6 | 66.2 | 64.3 | 52.6 | 52.8 | 61.0 | 62.3 |
| MiddlePhalanxOutlineCorrect | 69.8 | 83.8 | 82.5 | 81.8 | 81.8 | 75.3 | 74.4 | 75.2 | 79.5 | 79.6 | 82.6 | 81.8 | 82.1 | 57.0 | 74.3 | 75.6 | 61.2 | 79.4 | 84.5 | 51.9 | 81.5 | 79.8 | 82.2 |
| MiddlePhalanxTW | 50.6 | 58.4 | 59.1 | 61.0 | 61.0 | 50.6 | 55.1 | 59.7 | 50.1 | 56.2 | 49.5 | 63.0 | 59.7 | 28.6 | 56.9 | 53.6 | 61.7 | 62.3 | 61.0 | 57.1 | 67.4 | 68.0 | 68.2 |
| MoteStrain | 83.5 | 86.1 | 85.1 | 84.3 | 84.3 | 76.8 | 88.5 | 87.2 | 93.6 | 69.1 | 92.4 | 91.4 | 83.9 | 53.9 | 80.9 | 85.5 | 86.6 | 86.7 | 89.7 | 68.1 | 89.2 | 82.4 | 90.0 |
| OSULeaf | 59.1 | 85.1 | 76.0 | 72.3 | 72.3 | 54.5 | 48.2 | 55.4 | 97.9 | 41.9 | 98.0 | 54.5 | 56.2 | 18.2 | 62.8 | 56.0 | 43.0 | 56.2 | 83.5 | 23.1 | 79.6 | 71.9 | 91.0 |
| PhalangesOutlinesCorrect | 72.8 | 80.9 | 78.4 | 80.4 | 80.4 | 77.3 | 79.9 | 74.5 | 81.8 | 79.5 | 84.5 | 82.8 | 81.2 | 61.3 | 65.6 | 75.6 | 67.2 | 75.9 | 82.1 | 66.3 | 76.8 | 79.3 | 77.2 |
| PickupGestureWiimoteZ | 66.0 | 82.0 | 74.0 | 60.0 | 60.0 | 24.0 | 60.8 | 49.6 | 74.4 | 41.2 | 70.4 | 76.0 | 70.0 | 10.0 | 61.6 | 60.4 | 64.0 | 80.0 | 84.0 | 8.0 | 85.3 | 88.8 | 88.0 |
| Plane | 100.0 | 100.0 | 99.0 | 100.0 | 100.0 | 93.3 | 96.2 | 96.4 | 100.0 | 95.2 | 100.0 | 98.1 | 98.1 | 14.3 | 100.0 | 97.7 | 99.0 | 98.1 | 99.0 | 92.4 | 99.8 | 100.0 | 100.0 |
| PowerCons | 87.8 | 96.1 | 90.0 | 96.1 | 96.1 | 91.1 | 96.0 | 97.1 | 86.3 | 92.9 | 87.9 | 100.0 | 100.0 | 50.0 | 85.2 | 97.7 | 98.9 | 99.4 | 96.7 | 98.9 | 99.6 | 100.0 | 100.0 |
| ProximalPhalanxOutlineAgeGroup | 80.5 | 83.4 | 84.4 | 83.9 | 83.9 | 85.4 | 81.2 | 87.2 | 82.5 | 83.9 | 84.7 | 86.3 | 87.3 | 48.8 | 83.9 | 84.9 | 85.9 | 87.8 | 85.3 | 83.9 | 83.1 | 80.4 | 83.6 |
| ProximalPhalanxOutlineCorrect | 78.4 | 88.7 | 85.9 | 87.3 | 87.3 | 77.0 | 80.7 | 76.9 | 90.7 | 86.6 | 92.0 | 88.3 | 88.3 | 68.4 | 81.7 | 73.0 | 79.7 | 82.5 | 85.6 | 80.1 | 87.6 | 87.3 | 88.4 |
| ProximalPhalanxTW | 76.1 | 82.4 | 77.1 | 80.0 | 80.0 | 78.0 | 77.7 | 79.1 | 76.1 | 77.5 | 77.3 | 81.5 | 82.4 | 34.1 | 78.4 | 76.7 | 80.0 | 81.0 | 82.0 | 61.2 | 71.2 | 84.3 | 83.2 | 86.0 |
| ShakeGestureWiimoteZ | 86.0 | 94.0 | 92.0 | 86.0 | 86.0 | 76.0 | 58.0 | 75.6 | 88.4 | 52.4 | 88.0 | 82.0 | 80.0 | 10.0 | 86.4 | 54.8 | 62.0 | 82.0 | 86.0 | 8.0 | 86.7 | 81.6 | 89.0 |
| ShapeletSim | 65.0 | 100.0 | 67.2 | 68.3 | 68.3 | 48.9 | 49.7 | 51.0 | 70.6 | 49.8 | 78.2 | 57.8 | 50.0 | 50.0 | 54.6 | 51.3 | 47.2 | 52.8 | 100.0 | 48.9 | 92.1 | 82.7 | 93.7 |
| ShapesAll | 76.8 | 90.2 | 84.8 | 77.3 | 77.3 | 73.3 | 61.7 | 67.9 | 89.4 | 59.9 | 92.6 | 71.0 | 73.2 | 1.7 | 64.3 | 77.6 | 63.3 | 74.2 | 82.8 | 23.7 | 81.9 | 82.0 | 82.5 |
| SmoothSubspace | 82.7 | 98.0 | 96.0 | 95.3 | 95.3 | 82.7 | 97.6 | 96.4 | 96.3 | 98.0 | 98.0 | 86.7 | 33.3 | 84.9 | 98.0 | 86.0 | 45.3 | NaN | 86.0 | 45.3 | 89.7 | 98.3 | 98.3 |
| SonyAIBORobotSurface1 | 72.5 | 90.3 | 90.2 | 89.9 | 89.9 | 72.4 | 69.0 | 72.9 | 95.8 | 65.5 | 96.1 | 80.4 | 73.5 | 42.9 | 72.5 | 69.2 | 65.6 | 85.9 | 89.5 | 58.9 | 92.7 | 93.8 | 94.2 |
| SonyAIBORobotSurface2 | 83.1 | 87.1 | 88.9 | 90.7 | 90.7 | 74.5 | 83.1 | 84.4 | 98.0 | 80.4 | 97.5 | 84.2 | 83.1 | 61.7 | 63.5 | 83.1 | 82.2 | 84.1 | 89.4 | 65.0 | 90.5 | 88.7 | 91.2 |
| Strawberry | 94.1 | 96.2 | 95.4 | 96.5 | 96.5 | 91.6 | 95.2 | 95.9 | 97.5 | 95.8 | 98.0 | 97.0 | 95.9 | 64.3 | 91.1 | 95.9 | 93.5 | 94.1 | 96.5 | 93.5 | 93.2 | 89.4 | 92.8 |
| SwedishLeaf | 79.2 | 94.1 | 91.4 | 92.3 | 92.3 | 73.8 | 88.4 | 90.2 | 96.7 | 84.1 | 96.3 | 90.4 | 90.1 | 6.4 | 83.7 | 84.5 | 83.2 | 89.1 | 95.4 | 89.9 | 90.7 | 90.0 | 91.0 |
| Symbols | 95.0 | 97.6 | 96.3 | 91.6 | 91.6 | 78.6 | 80.8 | 75.4 | 95.5 | 64.4 | 89.3 | 88.8 | 84.8 | 17.4 | 79.8 | 83.6 | 82.0 | 87.0 | 93.7 | 69.4 | 98.5 | 99.0 | 99.2 |
| SyntheticControl | 99.3 | 99.7 | 98.7 | 99.0 | 99.0 | 49.0 | 98.7 | 97.3 | 99.7 | 95.3 | 99.7 | 99.3 | 92.7 | 16.7 | 87.9 | 97.3 | 88.7 | 98.3 | 99.0 | 43.7 | 94.8 | 91.2 | 95.3 |
| ToeSegmentation1 | 77.2 | 91.7 | 93.9 | 93.0 | 93.0 | 80.7 | 59.8 | 70.6 | 96.1 | 55.9 | 95.7 | 66.2 | 61.4 | 52.6 | 88.2 | 58.9 | 61.4 | 64.0 | 96.5 | 56.1 | 90.2 | 91.9 | 91.0 |
| ToeSegmentation2 | 83.8 | 89.2 | 90.0 | 87.7 | 87.7 | 61.5 | 75.2 | 70.2 | 88.9 | 64.9 | 89.4 | 83.1 | 63.1 | 81.5 | 79.4 | 74.5 | 51.5 | 81.5 | 94.6 | 73.1 | 94.8 | 92.3 | 96.2 |
| Trace | 100.0 | 100.0 | 99.0 | 100.0 | 100.0 | 100.0 | 95.2 | 74.0 | 100.0 | 90.2 | 100.0 | 92.0 | 78.0 | 24.0 | 93.4 | 80.6 | 68.0 | 84.0 | 100.0 | 71.0 | 99.0 | 100.0 | 100.0 |
| TwoLeadECG | 90.5 | 98.6 | 99.9 | 97.6 | 97.6 | 87.1 | 87.7 | 78.4 | 99.9 | 80.6 | 100.0 | 78.2 | 63.6 | 50.0 | 94.9 | 75.3 | 75.2 | 60.7 | 96.8 | 65.8 | 99.5 | 100.0 | 100.0 |
| TwoPatterns | 100.0 | 100.0 | 99.9 | 99.9 | 99.9 | 46.6 | 99.1 | 100.0 | 87.6 | 97.6 | 100.0 | 99.4 | 94.8 | 25.9 | 87.5 | 94.8 | 85.3 | 99.1 | 99.9 | 92.3 | 93.8 | 93.4 | 94.5 |
| UMD | 99.3 | 100.0 | 99.3 | 98.6 | 98.6 | 91.0 | 96.0 | 77.1 | 98.8 | 84.2 | 99.0 | 100.0 | 88.9 | 33.3 | 83.5 | 94.9 | 95.1 | 93.8 | 97.9 | 36.8 | 99.2 | 100.0 | 100.0 |
| UWaveGestureLibraryX | 72.8 | 79.5 | 78.5 | 73.3 | 73.3 | 56.9 | 72.1 | 77.1 | 75.4 | 72.9 | 78.1 | 72.9 | 78.4 | 12.0 | 53.1 | 74.4 | 77.9 | 78.2 | 72.4 | 80.5 | 80.5 | | 80.5 |
| UWaveGestureLibraryY | 63.4 | 71.9 | 71.0 | 64.1 | 64.1 | 34.8 | 62.6 | 67.6 | 64.2 | 63.9 | 66.6 | 62.6 | 69.9 | 12.1 | 49.7 | 69.9 | 59.3 | 70.8 | 76.3 | 64.8 | 73.6 | 74.0 | 74.1 |
| UWaveGestureLibraryZ | 65.8 | 77.0 | 75.7 | 69.0 | 69.0 | 65.5 | 63.0 | 68.4 | 72.7 | 64.5 | 74.3 | 64.8 | 69.6 | 12.1 | 57.3 | 69.7 | 55.6 | 72.8 | 75.7 | 64.3 | 78.3 | 78.0 | 79.9 |
| Wafer | 98.0 | 99.8 | 99.2 | 99.4 | 99.4 | 99.1 | 96.1 | 99.8 | 99.7 | 99.2 | 99.8 | 99.8 | 99.3 | 89.2 | 91.6 | 99.6 | 94.9 | 99.3 | 99.9 | 99.4 | 99.7 | 99.7 | 99.9 |
| Wine | 57.4 | 87.0 | 81.5 | 77.8 | 77.8 | 50.0 | 51.9 | 55.6 | 61.1 | 50.0 | 72.2 | 70.4 | 64.8 | 50.0 | 74.4 | 54.1 | 66.7 | 50.0 | 63.0 | 61.1 | 70.8 | 70.4 | 72.1 |
| WordSynonyms | 64.9 | 67.6 | 69.1 | 53.1 | 53.1 | 42.2 | 56.8 | 55.7 | 56.1 | 47.0 | 61.7 | 57.7 | 57.8 | 21.9 | 50.6 | 59.9 | 45.3 | 59.1 | 68.3 | 45.1 | 62.4 | 63.0 | 63.9 |
| Yoga | 83.7 | 88.7 | 83.7 | 79.1 | 79.1 | 83.0 | 78.6 | 75.3 | 83.7 | 74.1 | 86.7 | 79.3 | 76.1 | 53.6 | 62.6 | 85.6 | 65.7 | 79.5 | 88.0 | 69.1 | 81.9 | 84.1 | 82.7 |
| Avg. | 76.4 | 85.2 | 83.4 | 79.3 | 79.3 | 65.9 | 75.2 | 74.3 | 80.9 | 70.2 | 82.5 | 80.0 | 76.8 | 34.8 | 72.7 | 75.1 | 71.0 | 78.4 | 85.3 | 56.7 | 78.9 | 83.7 | **86.4** |

Table 26: Full results for the classification task on UEA Archive (2018). ∗. in the Transformers indicates the name of ∗former. We report the classification accuracy (%) as the result. The standard deviation is within 0.1%. **Bold**: the best, Underline: the second best.

| Datasets / Models | Classical methods | | | RNN | | | TCN | Transformers | | | | | | | | | MLP | | CNN | | Pre-trained FMs | | | |
|---|---|---|---|---|---|---|---|---|---|---|---|---|---|---|---|---|---|---|---|---|---|---|---|---|
| | DTW | XGBoost | Rocket | LSTM | LSTNet | LSSL | TCN | Trans. | Re. | In. | Pyra. | Auto. | Station. | FED. | ETS. | Flow. | DLinear | LightTS. | TimesNet | MTCN | MOMENT | FedAvg | FFTS | **FeDaL** |
| | (2017) | (2016) | (2020) | (1997) | (2018) | (2021) | (2019) | (2017) | (2020) | (2021) | (2021) | (2021) | (2022) | (2022) | (2022) | (2022b) | (2023) | (2022) | (2022a) | (2024) | (2024) | (2017) | (2025) | (Ours) |
| EthanolConcentration | 32.3 | 43.7 | 45.2 | 32.3 | 39.9 | 31.1 | 28.9 | 32.7 | 31.9 | 31.6 | 30.8 | 31.6 | 32.7 | 31.2 | 28.1 | 33.8 | 32.6 | 29.7 | 35.7 | 36.3 | 30.0 | 27.7 | 32.1 | 35.4 |
| FaceDetection | 52.9 | 63.3 | 64.7 | 57.7 | 65.7 | 66.7 | 52.8 | 67.3 | 68.6 | 67.0 | 65.7 | 68.4 | 68.0 | 66.0 | 66.3 | 67.6 | 68.0 | 67.5 | 68.6 | 70.8 | 68.9 | 65.2 | 69.0 | 70.0 |
| Handwriting | 28.6 | 15.8 | 58.8 | 15.2 | 25.8 | 24.6 | 53.3 | 32.0 | 27.4 | 32.8 | 29.4 | 36.7 | 31.6 | 28.0 | 32.5 | 33.8 | 27.0 | 26.1 | 32.1 | 30.6 | 35.1 | 35.0 | 37.2 | 37.3 |
| Heartbeat | 71.7 | 73.2 | 75.6 | 72.2 | 77.1 | 72.7 | 75.6 | 76.1 | 77.1 | 80.5 | 75.6 | 74.6 | 73.7 | 73.7 | 71.2 | 77.6 | 75.1 | 75.1 | 78.0 | 77.2 | 73.7 | 74.0 | 75.2 | 80.2 |
| JapaneseVowels | 94.9 | 86.5 | 96.2 | 79.7 | 98.1 | 98.4 | 98.9 | 98.7 | 97.8 | 98.9 | 98.4 | 96.2 | 99.2 | 98.4 | 95.9 | 98.9 | 96.2 | 96.2 | 98.4 | 98.8 | 95.7 | 94.2 | 94.7 | 99.6 |
| PEMS-SF | 71.1 | 98.3 | 75.1 | 39.9 | 86.7 | 86.1 | 68.8 | 82.1 | 82.7 | 81.5 | 83.2 | 82.7 | 87.3 | 80.9 | 86.0 | 83.8 | 75.1 | 88.4 | 89.6 | 89.1 | 85.5 | 86.0 | 84.5 | 85.6 |
| SelfRegulationSCP1 | 77.7 | 84.6 | 90.8 | 68.9 | 84.0 | 90.8 | 84.6 | 92.2 | 90.4 | 90.1 | 88.1 | 84.0 | 89.4 | 88.7 | 89.6 | 92.5 | 87.3 | 89.8 | 91.8 | 93.4 | 88.7 | 89.1 | 90.5 | 95.1 |
| SelfRegulationSCP2 | 53.9 | 48.9 | 53.3 | 46.6 | 52.8 | 52.2 | 55.6 | 53.9 | 56.7 | 53.3 | 53.3 | 50.6 | 57.2 | 54.4 | 55.0 | 56.1 | 50.5 | 51.1 | 57.2 | 60.3 | 55.0 | 56.2 | 55.7 | 59.5 |
| SpokenArabicDigits | 96.3 | 69.6 | 71.2 | 31.9 | 100.0 | 100.0 | 95.6 | 98.4 | 97.0 | 100.0 | 99.6 | 100.0 | 100.0 | 100.0 | 100.0 | 98.8 | 81.4 | 100.0 | 99.0 | 98.7 | 99.4 | 98.0 | 99.5 | 100.0 |
| UWaveGestureLibrary | 90.3 | 75.9 | 94.4 | 41.2 | 87.8 | 85.9 | 88.4 | 85.6 | 85.6 | 85.6 | 83.4 | 85.9 | 87.5 | 85.3 | 85.0 | 86.6 | 82.1 | 80.3 | 85.3 | 86.7 | 90.0 | 87.6 | 80.6 | 98.5 |
| Average Accuracy | 67.0 | 66.0 | 72.5 | 48.6 | 71.8 | 70.9 | 70.3 | 71.9 | 71.5 | 72.1 | 70.8 | 71.1 | 72.7 | 70.7 | 71.0 | 73.0 | 67.5 | 70.4 | 73.6 | 74.2 | 72.2 | 71.5 | 71.7 | **76.1** |

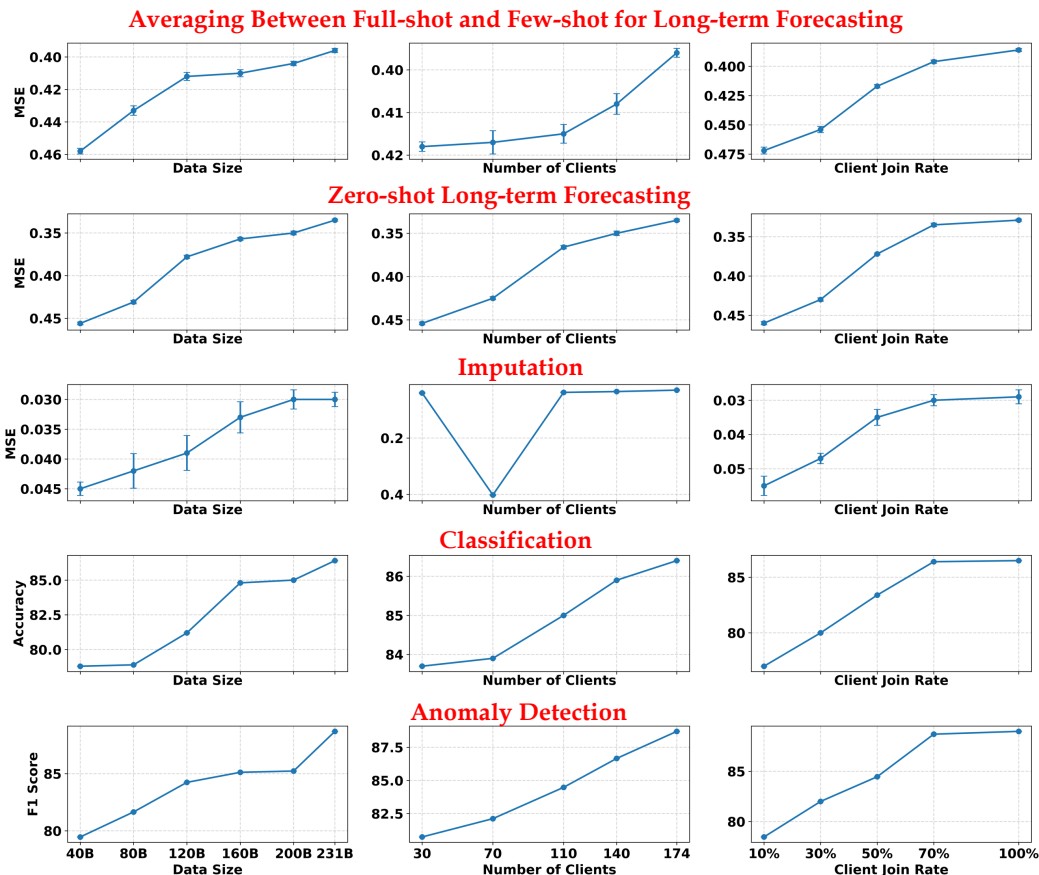

Figure 10: Scaling behavior from a federated learning perspective. Note that for forecasting and imputation, the y-axis is inverted for better visualization (lower values indicate better performance).

independently by the authors. The LLM was not employed for generating scientific content. We take full responsibility for the authenticity, originality, and integrity of this work.

Table 27: Federated representation learning results with variance estimates. For USTD, only the H2 setting is reported as a representative case.

| Method | USTD | | | | | CTSD | | | | |
|---|---|---|---|---|---|---|---|---|---|---|
| | 20% | 35% | 50% | 75% | 90% | 20% | 35% | 50% | 75% | 90% |
| FedAvg | 0.404 (± 0.007) | 0.489 (± 0.011) | 0.565 (± 0.004) | 0.602 (± 0.007) | 0.901 (± 0.003) | 0.350 (± 0.002) | 0.388 (± 0.008) | 0.405 (± 0.011) | 0.480 (± 0.005) | 0.652 (± 0.005) |
| FedProx | 0.401 (± 0.004) | 0.486 (± 0.010) | 0.553 (± 0.013) | 0.603 (± 0.009) | 0.889 (± 0.011) | 0.336 (± 0.009) | 0.390 (± 0.004) | 0.400 (± 0.009) | 0.454 (± 0.019) | 0.638 (± 0.004) |
| FedPer | 0.385 (± 0.002) | 0.578 (± 0.008) | 0.533 (± 0.010) | 0.580 (± 0.007) | 0.863 (± 0.012) | 0.330 (± 0.001) | 0.381 (± 0.004) | 0.395 (± 0.004) | 0.439 (± 0.008) | 0.606 (± 0.015) |
| FedRep | 0.413 (± 0.007) | 0.560 (± 0.003) | 0.537 (± 0.005) | 0.592 (± 0.009) | 0.860 (± 0.007) | 0.352 (± 0.003) | 0.373 (± 0.003) | 0.386 (± 0.009) | 0.440 (± 0.010) | 0.598 (± 0.006) |
| FFTS | 0.327 (± 0.001) | 0.410 (± 0.010) | 0.510 (± 0.011) | 0.584 (± 0.007) | 0.823 (± 0.008) | 0.300 (± 0.002) | 0.357 (± 0.004) | 0.386 (± 0.002) | 0.440 (± 0.007) | 0.598 (± 0.005) |
| Stand-alone | 0.380 (± 0.002) | 0.450 (± 0.013) | 0.521 (± 0.001) | 0.616 (± 0.013) | 0.870 (± 0.006) | 0.342 (± 0.012) | 0.381 (± 0.004) | 0.395 (± 0.006) | 0.470 (± 0.014) | 0.646 (± 0.018) |
| FeDaL (Ours) | 0.300 (± 0.005) | 0.422 (± 0.003) | 0.489 (± 0.009) | 0.549 (± 0.005) | 0.795 (± 0.008) | 0.310 (± 0.007) | 0.319 (± 0.007) | 0.343 (± 0.002) | 0.405 (± 0.005) | 0.560 (± 0.011) |

Table 28: Long-term forecasting and imputation Results with Variance Estimates. We use the forecasting horizon of 720 and masking ratio of 50% with MSE as the representative case.

| Datasets | FeDaL (Long-term Forecasting) | FeDaL (Imputation) |
|---|---|---|
| ETTh1 | 0.457 (± 0.003) | 0.030 (± 0.001) |
| ETTh2 | 0.401 (± 0.004) | 0.022 (± 0.002) |
| ETTm1 | 0.464 (± 0.004) | 0.086 (± 0.001) |
| ETTm2 | 0.397 (± 0.002) | 0.048 (± 0.002) |
| Weather | 0.359 (± 0.005) | 0.024 (± 0.003) |

Table 29: Short-term forecasting results with standard deviations (SMAPE Report).

| Intervals | Yearly | Quarterly | Monthly | Others |
|---|---|---|---|---|
| FeDaL | 13.102 (± 0.11) | 9.808 (± 0.10) | 12.124 (± 0.19) | 4.508 (± 0.07) |

Table 30: Anomaly detection results with standard deviations (F1-score Report).

| Datasets | SMD | MSL | SMAP | SWaT | PSM |
|---|---|---|---|---|---|
| FeDaL | 88.46 (± 0.26) | 89.05 (± 0.12) | 71.70 (± 0.09) | 95.40 (± 0.15) | 98.88 (± 0.21) |

