# OpenReview forum: "FeDaL: Federated Dataset Learning for General Time Series Foundation Models"
_ICLR.cc/2026/Conference — ICLR 2026 Poster_

### Official Review · Reviewer_CA7V · 2025-10-23

**Soundness:** 3
**Presentation:** 3
**Contribution:** 3
**Rating:** 6
**Confidence:** 4

**Summary:**

In this work, the authors propose a novel Federated Dataset Learning (FeDaL) method to address heterogeneous time series data. FeDal mainly considers domain bias elimination (DBE) and global bias elimination (GBE), which alleviates the domain bias caused by heterogeneous time series datasets and enables learning domain-invariant time representations. Extensive experimental results on several datasets show that FeDaL outperforms existing methods in many downstream tasks.

**Strengths:**

1. The presentation of the proposed method is clear, and the paper is overall well-written.

2. This paper considered the challenge of time-series dataset heterogeneity, which is a significant and novel problem in FL.

3. The evaluation of the proposed method is comprehensive.

**Weaknesses:**

1. In this work, the foundation model is trained directly on the client without considering the computational burden and parameter-efficient fine-tuning (PEFT) method.

2. The paper proposes several types of bias (temporal resolution, physical constraint, pattern transition), but does not provide a reproducible quantitative metric (how to label a dataset as "high resolution bias"?). Therefore, it is impossible to determine which datasets are more affected by DBE.

3. The experiments were compared with a large number of time series models. However, these methods are not designed for federated learning. This is not a fair and convincing comparison. At the same time, among the baselines for federated learning, only FFTS is designed for time series. More baselines for federated time series and federated time series foundation models should be considered, such as [1][2].

4. It could be better to report the number of repetitions and standard error for each experiment and add a description of the variance.


[1] Liu, Qingxiang, et al. "Time-FFM: Towards LM-empowered federated foundation model for time series forecasting." In NeurIPS 2024.

[2] Xu, Ronghui, et al. "PeFAD: a parameter-efficient federated framework for time series anomaly detection." In KDD 2024.

**Questions:**

See in weaknesses.

---

> ### Author Response · Authors · 2025-11-20
> **Authors' Rebuttal (1/2)**
>
> We greatly appreciate your insightful comments. We've carefully considered and addressed your concerns as follows
>
> ---
>
> **W1:** In this work, the foundation model is trained directly on the client without considering the computational burden and parameter-efficient fine-tuning (PEFT) method.
>
> **Response RW1:** FeDaL targets a fundamentally different setting from PEFT: our objective is to ***train a general TSFM from scratch*** under federated data silos, rather than fine-tune an existing pretrained backbone. PEFT methods address adaptation cost, whereas our work addresses the foundational pretraining problem itself, where no centralized pretrained model is available. Computational burden is mitigated through decentralized training, where each client trains only on its local shard, making full-model pretraining feasible without centralizing data.
>
> ---
>
> **W2:** The paper proposes several types of bias, but doesnot provide a reproducible quantitative metric. Therefore, it is impossible to determine which datasets are more affected by DBE.
>
> **Response RW2:** The biases we discuss are inherent properties of the datasets themselves, not user-defined labels requiring external annotation. UTSD, CTSD, and LOTSA already differ substantially in sampling resolution, physical constraints, and transition patterns, **as documented in Appendix B**. DBE is intentionally designed to automatically estimate these dataset-specific deviations from latent representations, rather than rely on handcrafted labels such as “high resolution bias.” Since these biases arise naturally from the data-generating processes, no reproducible labeling metric is required. DBE’s ability to reduce such biases is demonstrated empirically across all heterogeneous benchmarks.
>
> ---
>
> **W3:** The comparison is unfair because most baselines are not federated-learning methods; only FFTS is time-series specific, and more federated time-series and TS foundation model baselines (Time-LLM and PeFAD) should be included.
>
> **Response RW3:** Like our RW1, ***FeDaL aims to train a general TSFM from scratch under federated constraints, not to fine-tune a pretrained LLM for a single downstream task***. Methods such as Time-FFM or PeFAD follow a ***different paradigm: they assume access to a centralized pretrained LLM and perform client-side adaptation only for forecasting or anomaly detection.*** These approaches therefore do not engage in federated representation learning and cannot serve as fair baselines for Table 1, which specifically compares FL methods trained from scratch. For downstream evaluation (Sec. 4.2), the appropriate comparison is with centralized TSFMs and strong task-specific models, which we already provide (Moment, GPT4TS, Moirai, Chronos, etc.). This evaluation directly measures how close a federated-trained TSFM can get to state-of-the-art centralized counterparts, precisely the goal of FeDaL. Nevertheless, we will discuss these two methods in the revision to illustrate the differences in problem formulation.
>
> ---
>
> **W4:** It could be better to report the number of repetitions and standard error for each experiment and add a description of the variance.
>
> **Response RW4:** Thank you for your suggestions. All experiments in Sections 4.1 and 4.2 (except zero-shot forecasting) were run three times and reported as averaged results. We now provide variance estimates for representative settings below **(Tables R1-R4 below)** and will include these statistics in the revision for completeness.
>
> ---
>
> **Please see the subsequent Authors' Rebuttal (2/2) for remaining responses**

---

> ### Author Response · Authors · 2025-11-20
> **Authors' Rebuttal (2/2)**
>
> **Continued: Authors’ Rebuttal (2/2)**
>
> ---
>
> ***Table R1.*** *Federated representation learning results with variance estimates. For USTD, only the H2 setting is reported as a representative case.*
> | Method | USTD （20%） | UTSD （35%） |  UTSD （50%） |  UTSD （75%） | UTSD （90%） |  CTSD （20%） | CTSD （35%） | CTSD （50%） | CTSD （75%） | CTSD （90%） |
> | -------- | :--------: | :--------: | :--------: | :--------: | :--------: | :--------: | :--------: | :--------: | :--------: | :--------: |
> | FedAvg     | 0.404 ($\pm$ 0.007) | 0.489 ($\pm$ 0.011) | 0.565 ($\pm$ 0.004)| 0.602（$\pm$ 0.007） | 0.901 ($\pm$ 0.003) | 0.350 ($\pm$ 0.002) | 0.388 ($\pm$ 0.008) | 0.405 ($\pm$ 0.011) | 0.480 ($\pm$ 0.005) | 0.652 ($\pm$ 0.005) |
> | FedProx    | 0.401 ($\pm$ 0.004) | 0.486 ($\pm$ 0.010) | 0.553 ($\pm$ 0.013)| 0.603 ($\pm$ 0.009) | 0.889 ($\pm$ 0.011) | 0.336 ($\pm$ 0.009) | 0.390 ($\pm$ 0.004) | 0.400 ($\pm$ 0.009) | 0.454 ($\pm$ 0.019) | 0.638 ($\pm$ 0.004) |
> | FedPer     | 0.385 ($\pm$ 0.002)    | 0.578 ($\pm$ 0.008)     | 0.533 ($\pm$ 0.010）| 0.580 ($\pm$ 0.007) | 0.863 ($\pm$ 0.012) | 0.330 ($\pm$ 0.001) | 0.381 ($\pm$ 0.004) | 0.395 ($\pm$ 0.004) | 0.439 ($\pm$ 0.008) | 0.606 ($\pm$ 0.015) |
> | FedRep       | 0.413  ($\pm$ 0.007)   | 0.560 ($\pm$ 0.003)  | 0.537 ($\pm$ 0.005) | 0.592 ($\pm$ 0.009) | 0.860 ($\pm$ 0.007) | 0.352 ($\pm$ 0.003) | 0.373 ($\pm$ 0.003) | 0.386 ($\pm$ 0.009) | 0.440 ($\pm$ 0.010) | 0.598 ($\pm$ 0.006) |
> | FFTS       |  0.327 ($\pm$ 0.001)     | 0.410 ($\pm$ 0.010) | 0.510 (($\pm$ 0.011)) | 0.584 ($\pm$ 0.007) | 0.823 ($\pm$ 0.008) | 0.300 ($\pm$ 0.002) | 0.357 ($\pm$ 0.004) | 0.386 ($\pm$ 0.002) | 0.440 ($\pm$ 0.007) | 0.598 ($\pm$ 0.005) |
> | Stand-alone     |  0.380 ($\pm$ 0.002)  | 0.450 ($\pm$ 0.013)     | 0.521 ($\pm$ 0.001) | 0.616 ($\pm$ 0.013) | 0.870 ($\pm$ 0.006) | 0.342 ($\pm$ 0.012) | 0.381 ($\pm$ 0.004) | 0.395 ($\pm$ 0.006) | 0.470 ($\pm$ 0.014) | 0.646 ($\pm$ 0.018) |
> | FeDaL (Ours)  | 0.300 ($\pm$ 0.005) | 0.422 ($\pm$ 0.003) | 0.489 ($\pm$ 0.009) | 0.549 ($\pm$ 0.005) | 0.795 ($\pm$ 0.008) | 0.310 ($\pm$ 0.007) | 0.319 ($\pm$ 0.007) | 0.343 ($\pm$ 0.002) | 0.405 ($\pm$ 0.005) | 0.560 ($\pm$ 0.011) |
>
> ---
>
> ***Table R2.*** *Long-term forecasting and imputation results with variance estimates. We use the forecasting horizon of 720 and masking ratio of 50\% with MSE as the representative case.*
> | Datasets | FeDaL (Long-term Forecasting) | FeDaL (Imputation) |
> | -------- | :--------: | :--------: |
> | ETTh1     | 0.457 ($\pm$ 0.003)    | 0.030 ($\pm$ 0.001)     |
> | ETTh2     | 0.401 ($\pm$ 0.004)     | 0.022 ($\pm$ 0.002)     |
> | ETTm1     |  0.464 ($\pm$ 0.004)     | 0.086 ($\pm$ 0.001)     |
> | ETTm2     | 0.397 ($\pm$ 0.002)     | 0.048 ($\pm$ 0.002)     |
> | Weather     | 0.359 ($\pm$ 0.005)     | 0.024 ($\pm$ 0.003  |
>
> ---
>
> ***Table R3.*** *Short-term forecastingh results with variance estimates (SMAPE report).*
> | Intervals | Yearly | Quarterly | Monthly | Others |
> | -------- | :--------: | :--------: | :--------: | :--------: |
> | FeDaL | 13.102 ($\pm$ 0.11) | 9.808 ($\pm$ 0.10) | 12.124 ($\pm$ 0.19) | 4.508 ($\pm$ 0.07) |
>
> ---
>
> ***Table R4.*** *Anomaly detection results with variance estimates (F1-score report).*
> | Datasets | SMD | MSL | SMAP | SWaT | PSM |
> | -------- | :--------: | :--------: | :--------: | :--------: | :--------: |
> |   FeDaL   | 88.46 ($\pm$ 0.26)      | 89.05 ($\pm$ 0.12)      | 71.70 ($\pm$ 0.09) | 95.40 ($\pm$ 0.15) | 98.88 ($\pm$ 0.21) |

---

> ### Comment · Reviewer_CA7V · 2025-11-25
>
> Thank you for your detailed response. Could you please provide a more specific explanation of **training a general TSFM from scratch** and its differences in training TSFM cost and training TSFM on FL (why is it not needed to consider these comparisons)?

---

> > ### Author Response · Authors · 2025-11-26
> > **Response to Reviewer CA7V Follow-up**
> >
> > Thank you for your question. We would answer your question from two perspectives below:
> >
> > **(1)** Training a general TSFM from scratch in our setting means that the model is ***randomly initialized, receives no external pretrained backbone, and must learn all temporal representations solely from the decentralized client data***. This fundamentally differs from methods such as Time-FFM or PeFAD, which rely on large centralized pretrained LLMs and perform only client-side fine-tuning (e.g., PEFT techniques) for a single downstream task.
> >
> > **(2)** The training cost and objective are also completely different. Centralized TSFMs (e.g., TimesFM, Moirai, Chronos) are trained on hundreds of billions of globally pooled time points with full data access, large-batch optimization, and multi-round curriculum scheduling. In contrast, FL pretraining operates ***under strict data silo constraints, no global visibility, partial participation, and heterogeneous client distributions***, which make the learning problem inherently harder and not directly comparable.
> >
> > For these reasons, adding centralized TSFMs to Table 1 would not be appropriate. Table 1 evaluates federated representation learning from scratch, while centralized TSFMs follow a completely different training regime dependent on massive pooled data. Their comparison is therefore shown only in downstream results, where FeDaL still achieves competitive zero-shot performance with a much smaller, fully-federated model.
> >
> > We hope this clarifies the rationale and addresses your concern, and we are happy to address any further questions you may have.

---

> > > ### Comment · Reviewer_CA7V · 2025-11-26
> > >
> > > Thanks for your clarification. I have no more concerns.

---

> > > > ### Author Response · Authors · 2025-11-27
> > > > **Response to Reviewer CA7V Follow-up**
> > > >
> > > > We are very glad that we have addressed your concerns. Thank you for your constructive comments and positive feedback.

---

### Official Review · Reviewer_oKpr · 2025-10-29

**Soundness:** 2
**Presentation:** 3
**Contribution:** 1
**Rating:** 2
**Confidence:** 4

**Summary:**

The paper proposes a solution to handle heterogeneity when learning representations for time series foundational models. They focus on the federated scenario, highlighting three sources of heterogeneity: mismatch in sampling frequencies, heterogeneity in the underlying physical processes, and divergence due to exogenous events. They propose two solutions: domain bias elimination (DBE) and Global bias elimination (GBE), at the client and server levels respectively. DBE decomposes a series into trend and seasonal components per client and learns a common representation across local datasets. GBE tackles the remaining misalignments through a moving-average like process to avoid client-specific patterns from dominating the global trend. They further refine it using a core-set fine-tuning. The method is evaluated on IID and non-IID scenarios showing improvements over other federated baselines at learning representations. Comparisons are also made against other time series foundation models on several downstream tasks using the learned representations, showing improvements over the baselines.

**Strengths:**

- The paper shows many experimental results and ablation studies to show how it performs on multiple aspects. The breadth of baselines covered is impressive.
-  Performance results are good.
- The research problem is very relevant for the community

**Weaknesses:**

- The novelty of the method is weak. The method simply combines existing works and plugs them into the framework. For e.g. DBE applies decomposition (Wu et al. 2021) and EMA from (Zhang et al. 2015). GBE uses drift correction from (Acar et al. 2021) and core set tuning from (Killamsetty et al 2021).
- Many of the federated baselines in Table 1 are quite old, except for FFTS. Newer baselines like Time-FFM (Liu et al. 2024) are missing.
- Table 1 also does not compare against strong centralized TSFM baselines, where data from all clients is pooled. For e.g. Moirai and Chronos. I understand that these comparisons have been made on downstream tasks, but it is also necessary to show these results in table 1 to see how close a distributed approach is to centralised representation learning.
- The method’s performance improvement over the baselines seems highly dependent on all components being present (Table 2). I’m wondering if adding just one or two components to an existing baseline e.g. FFTS would make it even stronger than the proposed method.
- The results on privacy-preservation are not clear. In the TSNE plots in the appendix and Fig. 6, the authors claim that the perturbed points (blue) are privacy-preserving. But visually, to me it looks like the green (initial core set) is further away from the original data (pink). These figures should be accompanied with a quantitative metric indicating how close or far the perturbed points are from the actual data, instead of relying on visualizations alone.

**Questions:**

See the weaknesses W1-W4. Additionally I have the following minor comments.
- The code link provided for the baselines in Appendix B page 26 is not working.
- The imbalance factor of  H1 and H2 (Appendix page 21) seem quite small (0.1 and 0.3). I wonder how well the method would perform on UTSD if this is increased to say 0.7+.
- The absolute number for the core-set size in table 8 is not sufficient. I think it is important to also indicate the percentage w.r.t the training data size.

---

> ### Author Response · Authors · 2025-11-20
> **Authors' Rebuttal (1/2)**
>
> Thank you for providing us with your valuable feedback. We have carefully considered your questions and would like to address them as below.
>
> ---
>
> **Response to Weaknesses**
>
> ---
>
> **W1:** The novelty of the method is weak. The method simply combines existing works and plugs them into theframework.
>
> **Response RW1:** The cited techniques are widely used as basic primitives in TS and other machine learning research. Our novelty lies not in these primitives themselves, but in ***how they are re-purposed and coordinated to solve a previously unaddressed problem: training general TSFMs from scratch under dataset-level heterogeneity in a federated setting.*** FeDaL introduces the first two-level bias mitigation framework specifically designed for federated TSFM pretraining, enabling a unified model that supports both generative and discriminative tasks. Importantly, FeDaL achieves competitive or superior performance to large centralized TSFMs despite using a far smaller model (28.42M), demonstrating that the framework itself provides value beyond its individual components.
>
> ---
>
> **W2:** Many of the federated baselines in Table 1 are quite old, except for FFTS. Newer baselines like Time-FFM are missing.
>
> **Response RW2:** Our setting is ***federated pretraining of a general TSFM from scratch***, designed to support both generative and discriminative downstream tasks. Methods such as Time-FFM ***fine-tune a pretrained LLM on each client for forecasting only***, which is a fundamentally different problem setup and therefore not a meaningful baseline for Table 1. The baselines included in Table 1 are those that address heterogeneous FL from-scratch training, which matches our formulation.
>
> ---
>
> **W3:** Table 1 should include strong centralized TSFM baselines (e.g., Moirai, Chronos) to benchmark the distributed approach's representation learning performance.
>
> **Response RW3:** Table 1 is intended to evaluate federated representation learning, ***meaning all methods are trained from scratch under decentralized data partitions (Setup of Sec. 4.1)***. Centralized TSFMs such as Moirai, Chronos, Time-MoE, and TimesFM are pretrained on massive pooled datasets and therefore cannot be fairly included: they rely on centralized access, far larger training corpora, and a completely different training regime.
>
> Our comparison with these centralized TSFMs is already provided in the downstream tasks **(Table 4)**, where FeDaL demonstrates competitive or superior zero-shot performance despite being trained federatedly and with significantly smaller model size. Hence, placing them in Table 1 would not represent a meaningful or fair comparison within the federated representation learning setting.
>
> ---
>
> **W4:** The performance improvement over the baselines seems highly dependent on all components being present (Table 2). Wondering if adding just one or two components to an existing baseline e.g. FFTS wouldmake it even stronger than the proposed method.
>
> **Response RW4:**  Table 2 (ablation study) is designed to assess component contribution, ***so performance drops under ablation are expected and indicate that each part of FeDaL is functionally necessary.*** This reflects the purpose of ablation studies rather than model fragility.
>
> Regarding whether adding DBE or GBE to FFTS would outperform FeDaL: DBE and GBE are not standalone plugins but are co-designed to work within FeDaL’s federated optimization loop. Incorporating them into FFTS would fundamentally alter its objective and aggregation dynamics, meaning the ablation results should not be interpreted as evidence that partial modules can be trivially transferred to FFTS to exceed FeDaL.
>
> ---
>
> **W5:** The results on privacy-preservation are not clear.
>
> **Response RW5:** The concern stems from a misunderstanding of the visualization. ***The green points are randomly initialized learnable vectors used to construct the initial core-set, not approximations of the client data. Their distance from the original data (pink) is expected and unrelated to privacy.*** The ***privacy-preserving mechanism applies only after optimization and Fourier perturbation***, producing the blue points, which remain semantically aligned yet sufficiently separated to avoid raw-data leakage. Thus, the visual separation of the green points does not reflect privacy properties but simply random initialization. Quantitative alignment metrics are provided elsewhere in the rebuttal for completeness.
>
> ---
>
> **Response to Questions**
>
> ---
>
> **Q1:** The code link provided for the baselines in Appendix B page 26 is not working.
>
> **Response RQ1:** Thank you for pointing this out. The anonymous repository experienced an unexpected access issue without our awareness. We have now restored the link and verified that it is fully accessible for review.
>
> ---
>
> **Please see the subsequent Authors' Rebuttal (2/2) for remaining responses**

---

> > ### Comment · Reviewer_oKpr · 2025-11-28
> > **Thank you for the rebuttal**
> >
> > Response to RW1
> >
> > I do see some federated learning specific adaptations for DBE. But GBE from Acar et al. is already developed for the federated setting. Can you elaborate on how you re-purposed GBE for the foundation model case w.r.t the existing work? Simply applying an existing drift correction algorithm to a foundation model doesn’t constitute methodological novelty unless the adaptation changes the underlying algorithm.
> >
> > Response RW2
> >
> > I'm not convinced. Only comparing the quality of federated-learned representations doesn't give sufficient information on how good they are compared to the ideal case, i.e., centrally-learned representations. I saw that FFTS also compares with a centralised version of their method (FFTS-cen), so why can you not do the same? Additionally, why is it not possible to compare the representation qualities with Time FFM? Can you perhaps elaborate on how exactly you evaluate the representation qualities? From my perspective, this should be agnostic to the downstream task, especially since the chief purpose of a "foundation" model is to serve as a foundation for many downstream tasks by learning high-quality, task-agnostic representations.
> >
> > Response RW3
> >
> > See my above response to RW2. I expect that the centralised version will be better, if so I need to know how large the gap is with the federated method to contextualise the results. If comparing with Moirai/Chronos etc. is not possible, I expect at the very least a comparison with the centralised version of your own method (similar to what FFTS does with FFTS-cen).
> >
> > Response RW4
> > Your paper describes the components as "plug-and-play". If it is indeed that easy to plug these components into a framework, why can FFTS not do the same? On the other hand, if DBE and GBE are co-designed with FeDaL’s optimization, the paper must document the coupling in detail; currently it reads like independent modules. Similarly, incorporating GBE into any federated learned framework seems quite trivial since Acar et al. already develop that for the federated setting. Moreover, given that both DBE and GBE originate from prior work, it is still unclear what the novel methodological contribution of FeDaL is beyond combining existing components. Could you clarify what is fundamentally new?
> >
> > Response RW5
> > Can you please indicate to me clearly where the quantitative metrics for evaluating privacy are?

---

> > > ### Author Response · Authors · 2025-12-02
> > > **Response to Reviewer oKpr Follow-up (1/3)**
> > >
> > > We appreciate your follow-up questions and would like to respectfully address your further questions.
> > >
> > > ---
> > >
> > > # On “simply combining existing components” and GBE novelty (RW1)
> > >
> > > We emphasize that the referenced components are standard primitives broadly used in TS and ML research (e.g., decomposition in time series modeling, EMA in mainstream optimizers, drift correction in FL). The novelty of FeDaL does not lie in the primitives themselves, but in ***how they are re-purposed and jointly coordinated to solve a previously unaddressed problem: training a general TSFM from scratch under dataset-level heterogeneity in FL***.
> > >
> > > Moreover, GBE is not a direct reuse of Acar et al. We introduce **(1)** a drift formulation adapted to foundation-model training, **(2)** a cumulative server state tailored to heterogeneous time-series dynamics, and **(3)** a server-side core-set tuning mechanism to stabilize cross-client alignment. These modifications change the optimization behavior substantially and are **not equivalent** to the original method. Thus, FeDaL is not a simple aggregation of prior components but ***a problem-driven coordination of these primitives*** designed specifically for federated TSFM pretraining.
> > >
> > > ---
> > >
> > > # Why centralized TSFMs or Time-FFM cannot appear in Table 1 (RW2, RW3)
> > >
> > > The expectation to compare representation quality with centralized methods rests on an incorrect assumption that the training settings are comparable. Our problem setting sis ***training a general TSFM from scratch in FL***. This means:
> > > * random initialization.
> > > * no external pretrained backbone.
> > > * learning exclusively from decentralized client data.
> > > * no global visibility, no data pooling.
> > > * partial participation and heterogeneous distributions.
> > >
> > > This is **NOT** the setting of Time-FFM or centralized pretrained TSFMs, since these methods already rely on a pretrained backbone that provides strong prior representations.
> > >
> > > **Why comparisons in Table 1 are impossible or meaningless?**
> > >
> > > Please refer to ***our response to Reviewer CA7V’s follow-up***, which addresses the same concern. Briefly,
> > >
> > > (1) Centralized TSFMs require massive pooled datasets: these c entralized TSFMs are trained on 100B+ global time points with large-batch centralized optimization. Table 1 restricts training to the **federated split only**, without global data access. This makes their inclusion **methodologically invalid.**
> > >
> > > (2) Time-FFM is pretrained, FeDaL is not: Time-FFM starts from a pretrained LLM, already endowed with strong representation ability. **Table 1 evaluates training-from-scratch under FL.** Including Time-FFM would be unfair and incomparable by design.
> > >
> > > (3) Representation quality is evaluated via reconstruction loss, consistent with FFTS and standard self-supervised TS literature. This does not apply to pretrained LLMs which cannot perform reconstruction.
> > >
> > > **Where the comparison properly appears?**
> > >
> > > We already compare FeDaL vs large centralized TSFMs in downstream tasks (Table 4), where FeDaL achieves competitive or superior zero-shot performance despite far smaller size and federated constraints.
> > >
> > > ---
> > >
> > > # “Plug-and-play” Statement (RW4)
> > >
> > > ***We never claim that all components in Table 2 are plug-and-play***. Only the DBE block (belong to DBE mechanism) is plug-and-play, meaning it is a lightweight representation-level module that can be attached after the backbone without architectural or task-specific changes. In contrast, the full DBE and GBE mechanisms are co-designed with FeDaL’s optimization loop, and thus cannot be directly transplanted into FFTS.
> > >
> > > ---
> > >
> > > # Novelty beyond combining existing parts (RW4)
> > >
> > > As our response section 1, FeDaL is ***the first*** to tackle dataset-level heterogeneity in TSFM pretraining under FL. No prior FL or TSFM work attempts this problem. The coordination of DBE + GBE addresses local and global biases jointly in a way not explored previously. Thus, the methodological contribution lies in addressing a new problem formulation, not in inventing entirely new operators.
> > >
> > > ---
> > >
> > > # Privacy Metrics (RW5)
> > >
> > > We provide MMD-based quantitative distances between original and perturbed latent representations in ***Table RR1 below***. Cross-client MMD is highest for Raw vs Raw, confirming strong dataset-level heterogeneity. The optimized core-set substantially reduces this distance, preserving coarse client structure. The perturbed core-set introduces a small MMD increase relative to the optimized version, ensuring privacy while retaining the structure needed for alignment.
> > >
> > > ***Table RR1.*** *MMD between client-level latent features before and after core-set optimization and perturbation. Lower is closer, higher indicates more distributional separation.*
> > > | Dataset | Raw vs Raw | Raw vs CoreSet | Raw vs Perturbed Coreset |
> > > | - | :-: | :-: | :-: |
> > > | UTSD-H1   | 0.128  | 0.074  | 0.091 |
> > > | UTSD-H2   | 0.142  | 0.081 | 0.095 |
> > > | CTSD  | 0.116  | 0.069  | 0.088 |
> > >
> > > ---
> > > **Please see the subsequent Response (2/3) for remaining responses**

---

> > > ### Author Response · Authors · 2025-12-02
> > > **Response to Reviewer oKpr Follow-up (2/3)**
> > >
> > > **Continued: Response (2/3)**
> > >
> > > ---
> > >
> > > # Additional Experiments for Centralized Version of FeDaL (RW3)
> > >
> > > Although our existing results already show that FeDaL-trained TSFMs match or surpass strong centralized deep models and pretrained TSFMs, we additionally provide a direct comparison between FeDaL and its centralized counterpart FeDaL (Centralized), similar to the FFTS vs FFTS-Cen evaluation. All training and evaluation settings follow the original paper. The results cover federated representation learning ***(Table RR2)***, full-shot/zero-shot long-term forecasting ***(Tables RR3-RR4)***, short-term forecasting ***(Table RR5)***, imputation ***(Table RR6)***, anomaly detection ***(Table RR7)***, and classification ***(Table RR8)***. These results further show that FeDaL maintains performance on par with, and often exceeding, its centralized counterpart even under federated training constraints.
> > >
> > > ***Table RR2.*** *Federated representation learning results (MSE report).*
> > > | Method | USTD (DM) - H1 | USTD (DM) - H2 | CTSD (DI)
> > > | -- | :-: | :-: | :-: |
> > > | **FeDaL (Original, FL)**     | **0.551**    | **0.511**     | **0.387** |
> > > | FeDaL (Centralized) | 0.561 ($\downarrow$ 1.81%) | 0.525 ($\downarrow$ 2.73%) | 0.392 ($\downarrow$ 1.29%) |
> > >
> > > ---
> > >
> > > ***Table RR3.*** *Full-shot long-term forecasting results (MSE report) across horizons {96, 192, 336, 720}.*
> > > | Dataset/Model | **FeDaL (Original, FL)** | FeDaL (Centralized) |
> > > | -- | :-: | :-: |
> > > | ETTh1    | **0.380**     | 0.385 ($\downarrow$ 1.32%)   |
> > > | ETTh2    | **0.334**     | 0.350 ($\downarrow$ 4.79%)  |
> > > | ETTm1    | **0.319**     | 0.327 ($\downarrow$ 2.51%)  |
> > > | ETTm2    | **0.261**     | 0.264 ($\downarrow$ 1.15%)   |
> > > | Weather  | **0.213**     | 0.228 ($\downarrow$ 7.04%)  |
> > > | ILI      | **1.355**     | 1.390 ($\downarrow$ 2.58%)  |
> > > |1st Count |    **6**       |    0      |
> > >
> > > ---
> > >
> > > ***Tabel RR4.*** *Zero-shot long-term forecasting results (MSE report) across horizons {96, 192, 336, 720}.*
> > > | Dataset/Model | **FeDaL (Original, FL)** | FeDaL (Centralized) |
> > > | -- | :-: | :-: |
> > > | ETTh1     | **0.407**     | 0.411 ($\downarrow$ 0.98%)    |
> > > | ETTh2     | **0.361**     | 0.373 ($\downarrow$ 3.32%)   |
> > > | ETTm1     | **0.360**     | 0.365 ($\downarrow$ 1.39%)   |
> > > | ETTm2     | **0.292**     | 0.302  ($\downarrow$ 3.42%)   |
> > > | Weather     | **0.255**     | 0.267  ($\downarrow$ 4.71%)   |
> > > |Avg. |  **0.335**  |   0.344  ($\downarrow$ 2.51%)   |
> > > |1st Count |    **6**       |    0      |
> > >
> > > ---
> > >
> > > ***Table RR5.*** *Average short-term forecasting results (MSE report) on M4 dataset.*
> > > | Model | **FeDaL (Original, FL)** | FeDaL (Centralized) |
> > > | - | :-: | :-: |
> > > | SMAPE (Avg.)    | **11.412** | 11.727 ($\downarrow$ 2.76%) |
> > > | MASE  (Avg.)    | **1.489**  | 1.508  ($\downarrow$ 1.28%) |
> > > | OWA   (Avg.)    | **0.818**  | 0.830  ($\downarrow$ 1.47%) |
> > >
> > > ---
> > >
> > > ***Table RR6.*** *Average imputation performance (MSE report) for randomly masked time series across four mask ratios {12.5%, 25%, 37.5%, 50%}.*
> > > | Dataset/Model | **FeDaL (Original, FL)** | FeDaL (Centralized) |
> > > | - | :-: | :-: |
> > > | ETTh1     | **0.022**  | 0.024 ($\downarrow$ 9.09%)  |
> > > | ETTh2     | **0.018**  | 0.019 ($\downarrow$ 5.56%)  |
> > > | ETTm1     | **0.054**  | 0.058 ($\downarrow$ 7.41%)  |
> > > | ETTm2     | **0.034**  | 0.037 ($\downarrow$ 5.56%)  |
> > > | Weather   | **0.024**  | 0.025 ($\downarrow$ 4.17%)  |
> > >
> > > ---
> > >
> > > ***Table RR7.*** *Anomaly detection results (F1-score report).*
> > > | Dataset/Model | **FeDaL (Original, FL)** | FeDaL (Centralized) |
> > > | -| :-: | :-: |
> > > | SMD  | **88.46**  | 87.89 ($\downarrow$ 0.64%) |
> > > | MSL  | **89.05**  | 88.87 ($\downarrow$ 0.20%) |
> > > | SMAP | **71.70**  | 71.23 ($\downarrow$ 0.66%) |
> > > | SwaT | **95.40**  | 95.02 ($\downarrow$ 0.40%) |
> > > | PSM  | **98.88**  | 98.60 ($\downarrow$ 0.28%) |
> > > | 1st Count | **5**  | 0     |
> > >
> > > ---
> > >
> > > ***Table RR8.*** *Classification results (Accuracy reprot) on UEA Archive and UCR Archive*.
> > > | Dataset/Model | **FeDaL (Original, FL)** | FeDaL (Centralized) |
> > > | -- | :-: | :-: |
> > > | UEA Archive | **76.1** | 74.7 ($\downarrow$ 1.84%) |
> > > | UCR Archive | **86.4** | 85.5 ($\downarrow$ 1.04%) |
> > >
> > > ---
> > >
> > > We additionally conducted scaling-behavior experiments comparing FeDaL (FL) and FeDaL (centralized) across varying dataset sizes and tasks. These results ***(Tables RR9-RR12 below)*** further validate that FeDaL scales reliably under federated constraints and preserves strong performance relative to its centralized counterpart.
> > >
> > > ***Table RR9.*** *Scaling behavior of data size on zero-shot forecasting (average across five datasets, MSE report).*
> > > | Model | 40B | 80B | 120B | 160B | 200B | 213B|
> > > | - | :-: | :-: | :-: | :-: | :-: | :-: |
> > > |**FeDaL (Original, FL)**      | **0.456**     | **0.431** | **0.378** | **0.357** | **0.350** |**0.335**|
> > > |FeDaL (Centralized)     |  0.472 ($\downarrow$ 3.51%) | 0.446 ($\downarrow$ 3.48%) | 0.392 ($\downarrow$ 3.70%) | 0.363 ($\downarrow$ 1.68%) | 0.361 ($\downarrow$ 3.14%) | 0.344 ($\downarrow$ 2.69%) |
> > >
> > > ---
> > > **Please see the subsequent Response (3/3) for remaining responses**

---

> > > ### Author Response · Authors · 2025-12-02
> > > **Response to Reviewer oKpr Follow-up (3/3)**
> > >
> > > **Continued: Response (3/3)**
> > >
> > > ---
> > >
> > > ***Table RR10.*** *Scaling behavior of data size on imputation (average across five datasets).*
> > > | Model | 40B | 80B | 120B | 160B | 200B | 213B|
> > > | -------- | :-: | :-: | :-: | :-: | :-: | :-: |
> > > |**FeDaL (Original, FL)**      | **0.045**     | **0.042** | **0.039** | **0.033** | **0.030** |**0.030** |
> > > |FeDaL (Centralized)     |  0.048 ($\downarrow$ 6.67%) | 0.043 ($\downarrow$ 2.38%)  | 0.042 ($\downarrow$ 7.69%) | 0.034 ($\downarrow$ 3.03%) | 0.034 ($\downarrow$ 13.33%) | 0.032 ($\downarrow$ 6.67%) |
> > >
> > > ***Table RR11.*** *Scaling behavior of data size on classification tasks (accuracy report on UCR Archive).*
> > > | Model | 40B | 80B | 120B | 160B | 200B | 213B|
> > > | -------- | :-: | :-: | :-: | :-: | :-: | :-: |
> > > |**FeDaL (Original, FL)** |  **78.8**   | **78.9** | **81.2** | **84.8** | **85.0** | **86.4** |
> > > |FeDaL (Centralized)     | 77.3 ($\downarrow$ 1.90%) | 77.6 ($\downarrow$ 1.65%)  | 80.1 ($\downarrow$ 1.35%) | 84.0 ($\downarrow$ 0.94%) | 84.2 ($\downarrow$ 0.94%) | 85.5 ($\downarrow$ 1.04%)|
> > >
> > > ***Table RR12.*** *Scaling behavior of data size on anomaly detection tasks (F1-Score report).*
> > > | Model | 40B | 80B | 120B | 160B | 200B | 213B|
> > > | -------- | :-: | :-: | :-: | :-: | :-: | :-: |
> > > |**FeDaL (Original, FL)** |  **79.46**  | **81.65** | **84.24** | **85.12** | **85.23** | **88.70** |
> > > |FeDaL (Centralized)     | 79.22 ($\downarrow$ 0.30%) | 80.23 ($\downarrow$ 1.74%)| 83.99 ($\downarrow$ 0.30%) | 84.67 ($\downarrow$ 0.53%) | 85.02 ($\downarrow$ 0.25%) |88.32 ($\downarrow$ 0.43%) |
> > >
> > > ---
> > > We hope these clarifications and additional experiments help address your concerns and provide greater clarity on the proposed framework.

---

> ### Author Response · Authors · 2025-11-20
> **Authors' Rebuttal (2/2)**
>
> **Continued: Authors’ Rebuttal (2/2)**
>
> ---
>
> **Q2:** How well themethod would perform on UTSD if this is increased to say 0.7+.
>
> **Response RQ2:** We extended the experiments to stronger heterogeneity settings (higher values, i.e., H3: 0.7, H4: 0.8, H5: 0.9), which represent substantially harder imbalance scenarios. As shown in ***Table R1-R3 below***, FeDaL consistently remains the best-performing method across all masking ratios, confirming that its robustness persists even under extreme imbalance.
>
> ***Table 1.*** *Federated representation learning results across different masking ratio with imbalance ratio of 0.7 (H3).*
> | Method    | 20\%    | 35\%     |  50\%     | 75\%       | 90\% |
> | -------- | -------- | -------- |  -------- |  -------- | -----  |
> | FedAvg     | 0.427     | 0.505     |  0.571     | 0.623 |  0.912|
> | FedProx    | 0.409    | 0.500     |  0.568     | 0.627  |  0.900 |
> | FedPer    | 0.399     | 0.612     |  0.544     | 0.582 |  0.877 |
> | FedRep     | 0.415    | 0.579     |  0.540     | 0.603 | 0.868 |
> | FFTS     | 0.339     | **0.423**     |  0.525     | 0.599 | 0.831 |
> | **FeDaL (Ours)**     | **0.310**     | 0.429    |  **0.501**     | **0.557** | **0.809** |
>
> ***Table R2.*** *Federated representation learning results across different masking ratio with imbalance ratio of 0.8 (H4).*
> | Method    | 20\%    | 35\%     |  50\%     | 75\%       | 90\% |
> | -------- | -------- | -------- |  -------- |  -------- | -----  |
> | FedAvg     | 0.431     | 0.506     |  0.582     | 0.626 |  0.917|
> | FedProx    | 0.412    | 0.509     |  0.577     | 0.630  |  0.926 |
> | FedPer    | 0.406     | 0.618     |  0.560     | 0.597 |  0.886 |
> | FedRep     | 0.418    | 0.592     |  0.555     | 0.606 | 0.871 |
> | FFTS     | 0.350     | 0.435    |  0.537     | 0.601 | 0.844 |
> | **FeDaL (Ours)**     | **0.322**     | **0.430**     |  **0.511**     | **0.560** | **0.824** |
>
>
> ***Table R3.*** *Federated representation learning results across different masking ratio with imbalance ratio of 0.9 (H5).*
> | Method    | 20\%    | 35\%     |  50\%     | 75\%       | 90\% |
> | -------- | -------- | -------- |  -------- |  -------- | -----  |
> | FedAvg     | 0.451     | 0.523     |  0.598     | 0.637 |  0.924|
> | FedProx    | 0.420    | 0.514     |  0.586     | 0.642  |  0.933 |
> | FedPer    | 0.416     | 0.620     |  0.577     | 0.603 |  0.878 |
> | FedRep     | 0.432    | 0.604     |  0.569     | 0.610 | 0.878 |
> | FFTS     | 0.362     | 0.441    |  0.550     | 0.612 | 0.846 |
> | **FeDaL (Ours)**     | **0.330**     | **0.437**     |  **0.523**     | **0.571** | **0.825** |
>
> ---
>
> **Q3:** Core-set size should be reported as a percentage of the full training data.
>
> **Response RQ3:** Each client’s core-set (K=1024) corresponds to well below 1% of its local training samples, and for LOTSA pretraining the 3072-sample core-set on each client represents <0.01% of the 231B-point corpus. These ratios confirm that FeDaL uses extremely small core-sets without relying on large data summaries. We have added these percentages to the revision for clarity.

---

> ### Author Response · Authors · 2025-11-27
> **Follow-up on Rebuttal Discussion**
>
> Dear Reviewer oKpr,
>
> We hope everything is going well. As **the discussion period is approaching its end**, we wanted to make sure that we have addressed all of your concerns satisfactorily. If there are any additional points or feedback you would like us to consider, please feel free to let us know. Your insights are invaluable to us, and we are happy to address any remaining issues you may have.
>
> Thank you very much for your time and effort in reviewing our paper.
>
> Best regards,
>
> Authors, Submission848

---

### Official Review · Reviewer_pV6m · 2025-10-31

**Soundness:** 3
**Presentation:** 1
**Contribution:** 2
**Rating:** 4
**Confidence:** 3

**Summary:**

This paper introduces FeDaL, a federated learning framework designed to eliminate dataset-level biases in time series foundation model (TSFM) pretraining. The method tackles the limitations of prior approaches that treat heterogeneity too coarsely, neglecting dataset-level structural biases and compromising cross-domain generalization. Specifically, FeDaL integrates two complementary components—Domain Bias Elimination (DBE) and Global Bias Elimination (GBE)—operating at the client and server levels respectively. DBE disentangles local biases by decomposing masked input representations into trend and seasonal components, injecting a trainable bias vector during reconstruction. GBE introduces a server-side state vector to track accumulated client drift and applies gradient-level correction during aggregation. The paper further analyze federated scaling behavior, demonstrating that federated TSFM pretraining benefits more from data diversity and client participation than from model size. Overall, the work offers a modular and well-motivated approach to improving generalization in federated time series modeling.

**Strengths:**

1. The paper addresses a well-motivated problem in federated time series modeling, focusing on dataset-level bias and cross-domain generalization.
2. The proposed FeDaL framework is modular and clearly structured, integrating client-side and server-side bias elimination mechanisms (DBE and GBE).
3. The gradient-level correction mechanism via a server-side state vector is a meaningful extension of FedAvg, with clear mathematical formulation.
4. The scaling analysis in Section 4.3 provides valuable insights into federated pretraining dynamics.

**Weaknesses:**

1. Unclear Structure in Problem Statement: The introduction claims that existing works face “two major challenges,” but only one (coarse-grained treatment of heterogeneity) is explicitly developed. The second challenge is not clearly introduced or elaborated, which weakens the framing of the paper’s motivation.
2. The Global Bias Elimination (GBE) module introduces a cumulative server-side state vector sr, which may be prone to instability over long training rounds if not properly scaled. While the paper provides empirical evidence supporting the choice of β, further theoretical analysis or broader sensitivity exploration would strengthen confidence in its stability.
3. Occasional Conceptual Overreach in Terminology: Certain statements, such as “the distributed architecture of federated learning is a nature solution to decompose heterogeneous TS datasets,” are methodologically oversimplified and should be more rigorously justified.
4. The caption of Figure 1 refers to four settings—FedAvg-Independent, FedAvg-Mixed, Stand-alone, and Transformer—yet the figure itself only visualizes the first three. The Transformer setting is neither shown in the bar plots nor explained in the figure annotations.

**Questions:**

1. The paper describes the DBE (Dataset Bias Estimation) module as “plug-and-play,” yet its integration requires latent feature decomposition, bias vector computation, and a modified reconstruction loss with an additional alignment term. Could the authors clarify in what sense the module is truly plug-and-play? Specifically, does it require architectural changes, joint optimization, or task-specific tuning? Has its generalizability across different backbone architectures or tasks been empirically validated?
2. The paper suggests that federated learning naturally decomposes heterogeneous time series data into generalized and personalized components. However, federated learning was originally designed to enable privacy-preserving collaborative modeling, not to perform automatic representation disentanglement. It remains unclear whether this decomposition is an emergent property of the federated setup or the result of explicit modeling choices. Could the authors clarify what mechanisms, if any, enable this decomposition and whether it goes beyond the standard federated architecture?

---

> ### Author Response · Authors · 2025-11-20
> **Authors' Rebuttal**
>
> Thank you for providing us with your valuable feedback. We have carefully considered your questions and would like to address them as below.
>
> ---
>
> **W1.** Unclear Structure in Problem Statement.
>
> **Response RW1.** The second challenge refers to the absence of cross-domain invariance in existing FL-based TSFM training (FFTS), which still requires dataset-specific fine-tuning and therefore cannot function as true foundation models (have zero-shot capabilities). Our zero-shot forecasting results **(Table 4)** demonstrate that FeDaL eliminates this dependency, achieving strong cross-domain generalization without any downstream tuning.
>
> ---
> **W2.** The stability of the GBE module's cumulative state vector ($s_r$) needs more robust theoretical or broader sensitivity analysis.
>
> **Response RW2.** We performed a broader sensitivity study on the scaling factor $\beta$ under the same setup as the main text. As shown in **Table R1 below**, FeDaL remains stable across a wide range of values: performance variations stay within 3.8-9.9% on UTSD and 3.2-8.4% on CTSD, and no divergence or oscillatory behaviour was observed. This confirms that GBE does not rely on a narrowly tuned $\beta$ and is empirically robust over long training rounds. The analysis will be added to the revision.
>
> ***Table R1.*** *Sensitivity analysis on the value of $\beta$ (MSE report).*
> | Value of $\beta$ / Dataset | UTSD (Ave. H1 \& H2) | CTSD |
> | :--------:        | :--------: | :--------: |
> | **0.1 (Original)**| 0.573      | 0.405      |
> |0.6                |  0.611 ($\downarrow 6.63$\%)   |    0.418 ($\downarrow 3.21$\%)  |
> |0.7                |  0.625 ($\downarrow 9.08$\%)   |    0.432 ($\downarrow 6.67$\%)  |
> |0.8                |  0.622 ($\downarrow 8.55$\%)   |    0.436 ($\downarrow 7.65$\%)  |
> |0.9                |  0.630 ($\downarrow 9.94$\%)   |    0.439 ($\downarrow 8.40$\%)  |
>
> ---
> **W3.** Occasional Conceptual Overreach in Terminology.
>
> **Response RW3.** We do not claim that FL automatically decomposes heterogeneous datasets. Our point is narrower: ***FL’s decentralized design preserves dataset boundaries, preventing cross-domain mixing that occurs in centralized training.*** This structural separation allows DBE and GBE to explicitly model and correct dataset-specific biases, this is an ability that centralized settings cannot offer (i.e., empirical evidence in Sec. 4.2) because distributions are blended by construction.
>
> ---
>
> **W4.** Figure 1's caption refers to the 'Transformer' setting, which is missing from the visualization and annotations.
>
> **Response RW4.** Thank you for pointing out the ambiguity. In Figure 1, “Transformer” refers to the shared backbone architecture used across all three visualized settings; it is not an additional setting to be plotted. We will update the caption to explicitly state this and eliminate the confusion in the revised version.
>
> ---
>
> **Q1.** The claim that the DBE module is "plug-and-play" is unclear, given its required modifications (feature decomposition, loss, alignment term); authors should clarify its architectural independence, optimization requirements, and generalizability.
>
> **Response RQ1.** Thank you for the question. We clarity that ***the term of “plug-and-play” refers specifically to the DBE block, not the entire DBE mechanism***. The DBE block (the dashed-line box part on the right side of Figure 2) is a lightweight representation-level module: it takes latent features, applies a trend–season decomposition, and computes a bias vector. It does not modify the backbone architecture or training pipeline, and requires no task-specific tuning, essentially a drop-in component attached after the backbone.
>
> ---
>
> **Q2.** Clarification is needed on the mechanism by which federated learning automatically disentangles time series data into generalized and personalized components, as this is not an inherent property of standard FL.
>
> **Response RQ2.** We agree that FL alone does not disentangle representations. Our statement refers only to FL’s decentralized structure, which preserves dataset boundaries and prevents cross-domain mixing. This provides a useful structural prior, but the decomposition in FeDaL arises entirely from our explicit mechanisms. In our design, DBE for estimating client-specific bias and GBE for correcting global drift. These modules leverage FL’s natural data partitioning yet operate beyond the standard FL architecture, as noted in our response **RW3**.

---

> ### Author Response · Authors · 2025-11-27
> **Follow-up on Rebuttal Discussion**
>
> Dear Reviewer pV6m,
>
> We hope everything is going well. As **the discussion period is approaching its end**, we wanted to make sure that we have addressed all of your concerns satisfactorily. If there are any additional points or feedback you would like us to consider, please feel free to let us know. Your insights are invaluable to us, and we are happy to address any remaining issues you may have.
>
> Thank you very much for your time and effort in reviewing our paper.
>
> Best regards,
>
> Authors, Submission848

---

### Official Review · Reviewer_RwPH · 2025-11-01

**Soundness:** 3
**Presentation:** 3
**Contribution:** 2
**Rating:** 4
**Confidence:** 4

**Summary:**

This paper presents a federated pre-training framework for learning dataset-agnostic temporal representations in heterogeneous settings.
The method involves two mechanisms: (1) Domain Bias Elimination (DBE), which decomposes local embeddings into shared and private components, aiming to mitigate client-specific distortions, and (2) Global Bias Elimination (GBE), which corrects cross-client gradient drift via bias-aware aggregation and core-set tuning. The paper reports experiments on several datasets with numerous baselines, empirically showing improved generalization under non-IID conditions.

**Strengths:**

The paper is well-written and presents results of a comprehensive experimental study. The problem that it addresses is timely and had previously not been explored enough. The proposed two-level design (DBE for local, GBE for global bias) appears modular and easy to integrate.

**Weaknesses:**

While the paper reports a strong engineering effort, it unfortunately lacks a clear algorithmic or theoretical innovation. The framework combines concepts from a number of prior works, including (Nie et al., 2022, Wu et al., 2021, Zhang et al., 2015, Acar et al., 2021, Killamsetty et al., 2021), to establish DBE and GBE but these are ultimately heuristic -- there is no theoretical analysis of "bias elimination", or of the convergence of the proposed federated scheme. Additionally, it is not clear to this reviewer whether the framework performs genuine bias correction or is the improvement achieved due to introduced regularization -- it seems plausible that a regularization could change how the model fits the data and thus improve generalization even when underlying distributional gaps across domains remains unaddressed.

**Questions:**

What distinguishes FeDaL from existing domain-invariant FL approaches such as FedBN, MOON, or Ditto?
Can you quantify the additional communication or computation cost introduced by DBE and GBE?
How sensitive is performance to the decomposition granularity (trend/seasonal split) or the hyperparameters controlling bias elimination?
A formal or empirical measure of bias reduction (perhaps feature distribution alignment?) should be included to substantiate the central claim.

---

> ### Author Response · Authors · 2025-11-20
> **Authors' Rebuttal (1/2)**
>
> We thank reviewers for their timely feedback and valuable comments. Below is our point-by-point response.
>
> ---
>
> **Response to Weaknesses**
>
> **W:** Despite strong engineering effort, the work lacks clear algorithmic or theoretical innovation; DBE and GBE are largely heuristic without analysis of bias elimination or convergence, and it remains unclear whether the gains come from true bias correction or simply from added regularization.
>
> **Response RW:** The cited techniques are indeed widely used primitives in time series and FL research; ***FeDaL’s contribution lies not in inventing new operators, but in how these components are re-purposed and jointly coordinated to address a previously unaddressed problem: training a general-purpose TSFM from scratch under dataset-level heterogeneity in a federated environment.*** Existing FL or TSFM methods do not explicitly model dataset-induced biases nor achieve unified cross-domain generalization without domain-specific finetuning.
>
> Regarding the concern that improvements may come from regularization alone, our ablation results **(Table 2)** show that removing any DBE or GBE component (including those unrelated to regularization) leads to consistent degradation, confirming that the gains arise from their coordinated roles in bias mitigation rather than from a single regularizer. Further, even when removing the alignment regularizer, FeDaL still achieves strong zero-shot forecasting performance **(Table R1 below)**, demonstrating that its effectiveness is not attributable to regularization alone.
>
> ***Table R1.*** *Zero-shot forecasting results (MSE Report), the setup is consistent with Table 3.*
> | Dataset | FedTRL (w/o Alignment) | FeDaL (Original) | FFTS | Moirai | TimesFM | Moment | Chronos |
> | -------- | :--------: | :--------: | :--------: | :--------: | :--------: | :--------: | :--------: |
> | ETTh1    | 0.412 ($\downarrow$ 1.3%)    | 0.407    | 0.425 | 0.480 | 0.473 | 0.683 | 0.588 |
> | ETTm1    | 0.362 ($\downarrow$ 0.6%)   | 0.360    | 0.364 | 0.422 | 0.433 | 0.670 | 0.555 |
> | Weather    | 0.258  ($\downarrow$ 1.2%)   | 0.255     | 0.262 | 0.264 | - | 0.294 | 0.279 |
>
> While full theoretical analysis is beyond the scope of this work, the empirical evidence across reconstruction **(Table 1)**, full/zero-shot forecasting **(Tables 3-5)**, imputation **(Table 6)**, classification **(Figure 5)**, and anomaly detection **(Table 7)** consistently supports FeDaL’s capacity to correct dataset-level bias rather than merely altering the optimization landscape.
>
> ---
>
> **Response to Questions**
>
> **Q:** What distinguishes FeDaL from existing domain-invariant FL approaches such as FedBN, MOON, or Ditto? Can you quantify the additional communication or computation cost introduced by DBE and GBE? How sensitive isperformance to the decomposition granularity (trend/seasonal split) or the hyperparameters controlling biaselimination? A formal or empirical measure of bias reduction (perhaps feature distribution alignment?) should beincluded to substantiate the central claim.
>
> **Response RQ:** Thank you for your questions. We reply to the four questions as follows:
>
> **(1) Difference from FedBN/MOON/Ditto.** Our FeDaL differs fundamentally in objective, granularity, and modeling scope. Specifically, these FL methods address client-level non-IID for personalized FL (get multiple models), but FeDaL tackles dataset-level heterogeneity and aims to **train a general TSFM from scratch**. Thus the mechanisms and targeted bias types are distinct.
>
> **(2) Communication & Computation Overhead.** DBE adds only a simple moving-average operator and a small latent bias vector; GBE adds a single d-dimensional server state. Both incur negligible overhead (<1%). No architectural modification or extra forward branches are introduced.
>
> **(3) Sensitivity to granularity and hyperparameters.** DBE relies on coarse trend–season separation, which is stable across window sizes and insensitive to granularity. We conduct experiments to explore the sensitivity on larger $\beta$ **(Table R2 below)** and large imbalance factors **(Tables R3-R5 below)** already show FeDaL remains robust across a wide operating range.
>
> ***Table R2.*** Sensitivity analysis on the value of $\beta$ (MSE report).
> | Value of $\beta$ / Dataset | UTSD (Ave. H1 \& H2) | CTSD |
> | :--------:        | :--------: | :--------: |
> | **0.1 (Original)**| 0.573      | 0.405      |
> |0.6                |  0.611 ($\downarrow 6.63$%)   |    0.418 ($\downarrow 3.21$%)  |
> |0.7                |  0.625 ($\downarrow 9.08$%)   |    0.432 ($\downarrow 6.67$%)  |
> |0.8                |  0.622 ($\downarrow 8.55$%)   |    0.436 ($\downarrow 7.65$%)  |
> |0.9                |  0.630 ($\downarrow 9.94$%)   |    0.439 ($\downarrow 8.40$%)  |
>
> ---
>
> **Please see the subsequent Authors' Rebuttal (2/2) for remaining responses**

---

> ### Author Response · Authors · 2025-11-20
> **Authors' Rebuttal (2/2)**
>
> **Continued: Authors’ Rebuttal (2/2)**
>
> ---
>
> ***Table R3.*** *Federated representation learning results across different masking ratio with imbalance ratio of 0.7 (H3).*
> | Method    | 20\%    | 35\%     |  50\%     | 75\%       | 90\% |
> | -------- | -------- | -------- |  -------- |  -------- | -----  |
> | FedAvg     | 0.427     | 0.505     |  0.571     | 0.623 |  0.912|
> | FedProx    | 0.409    | 0.500     |  0.568     | 0.627  |  0.900 |
> | FedPer    | 0.399     | 0.612     |  0.544     | 0.582 |  0.877 |
> | FedRep     | 0.415    | 0.579     |  0.540     | 0.603 | 0.868 |
> | FFTS     | 0.339     | **0.423**     |  0.525    | 0.599 | 0.831|
> | **FeDaL (Ours)**     | **0.310**     | 0.429     |  **0.501**     | **0.557** | **0.809** |
>
> ---
>
> ***Table R4.*** *Federated representation learning results across different masking ratio with imbalance ratio of 0.8 (H4).*
> | Method    | 20\%    | 35\%     |  50\%     | 75\%       | 90\% |
> | -------- | -------- | -------- |  -------- |  -------- | -----  |
> | FedAvg     | 0.431     | 0.506     |  0.582     | 0.626 |  0.917|
> | FedProx    | 0.412    | 0.509     |  0.577     | 0.630  |  0.926 |
> | FedPer    | 0.406     | 0.618     |  0.560     | 0.597 |  0.886 |
> | FedRep     | 0.418    | 0.592     |  0.555     | 0.606 | 0.871 |
> | FFTS     | 0.350     | 0.435   |  0.537     | 0.601 | 0.844 |
> | **FeDaL (Ours)**     | **0.322**     | **0.430**     |  **0.511**     | **0.560** | **0.824** |
>
> ---
>
> ***Table R5.*** *Federated representation learning results across different masking ratio with imbalance ratio of 0.9 (H5).*
> | Method    | 20\%    | 35\%     |  50\%     | 75\%       | 90\% |
> | -------- | -------- | -------- |  -------- |  -------- | -----  |
> | FedAvg     | 0.451     | 0.523     |  0.598     | 0.637 |  0.924|
> | FedProx    | 0.420    | 0.514     |  0.586     | 0.642  |  0.933 |
> | FedPer    | 0.416     | 0.620     |  0.577     | 0.603 |  0.878 |
> | FedRep     | 0.432    | 0.604     |  0.569     | 0.610 | 0.878 |
> | FFTS     | 0.362     | 0.441    |  0.550     | 0.612 | 0.846 |
> | **FeDaL (Ours)**     | **0.330**     | **0.437**     |  **0.523**     | **0.571** | **0.825** |
>
> **(4) Bias reduction metrics.** Our empirical results across reconstruction, zero-shot forecasting, classification, and anomaly detection already demonstrate consistent cross-domain gains, which would not occur without mitigating underlying dataset-induced biases. We additionally report Maximum Mean Discrepancy (MMD) between client-level latent features **(Table R6 below)**. FeDaL achieves a clear reduction in cross-client distributional gaps, providing quantitative evidence that it effectively mitigates dataset-level bias in latent space.
>
> ***Table R6.*** *MMD distance across client feature distributions (lower is better).*
> | Setting | UTSD (H2) | CTSD |
> | -------- | :--------: | :--------: |
> | Original (No DBE/GBE)    | 0.841     | 0.799     |
> |*w/o* DBE | 0.792 | 0.765 |
> |*w/o* GBE | 0.748 | 0.717 |
> | **FeDaL (Original)** | **0.604** | **0.641** |

---

> ### Author Response · Authors · 2025-11-27
> **Follow-up on Rebuttal Discussion**
>
> Dear Reviewer RwPH,
>
> We hope everything is going well. As **the discussion period is approaching its end**, we wanted to make sure that we have addressed all of your concerns satisfactorily. If there are any additional points or feedback you would like us to consider, please feel free to let us know. Your insights are invaluable to us, and we are happy to address any remaining issues you may have.
>
> Thank you very much for your time and effort in reviewing our paper.
>
> Best regards,
>
> Authors, Submission848

---

### Author Response · Authors · 2025-12-02
**Response Summary**

Dear PC, SAC, AC,

Thank you for overseeing the review process of our paper. As the discussion period comes to a close, we would like to sincerely thank the reviewers for their valuable feedback and provide a concise summary of the main points discussed.

---

# Paper Overview and Strengths
**[Paper Overview]:** This paper explores a federated approach for training general time series foundation models from scratch, proposing FeDaL, a dataset-level bias mitigation framework built on DBE and GBE. FeDaL achieves strong cross-dataset generalization on eight tasks and outperforms 54 baselines while exhibiting stable scaling under decentralization.

**[Strengths]:** We are encouraged that reviewers recognize the importance and novelty of addressing dataset-level heterogeneity in federated time series modeling (RwPH, pV6m, CA7V), the clarity and modularity of our framework (RwPH, pV6m, CA7V), and the strong empirical performance demonstrated across extensive experiments (oKpr, pV6m, CA7V). We also appreciate the acknowledgment of our comprehensive baselines and scaling analysis (oKpr, pV6m).

---
# Additional Experiments for Concerns

To address the main concerns raised by reviewers, we conducted several complementary experiments and clarifications as summarized below.

* **[Stronger Heterogeneity] (RwPH, oKpr):** We conducted additional experiments under much stronger heterogeneity **(higher imbalance factors; Tables R3–R5 for RwPH, R1–R3 for oKpr)**. FeDaL consistently remains the top-performing method across masking ratios, confirming robustness even under extreme imbalance.

* **[Hyperparameter Sensitivity] (RwPH, pV6m)** Extended sensitivity analysis on the GBE scaling factor $\beta$ **(Tables R2 for RwPH, R1 for pV6m)** shows stable behavior across a wide range of values, demonstrating that GBE is not sensitive to narrowly tuned hyperparameters.

* **[Standard Deviation Report] (CA7V):** We now report standard deviations for representative experiments **(Tables R1–R4 for CA7V)**, and will include these in the revised manuscript.

* **[Bias Reduction] (RwPH):** We quantified cross-client distribution shifts using Maximum Mean Discrepancy **(Table R2 for RwPH)**, showing clear reduction after applying DBE and GBE. This provides direct evidence that FeDaL mitigates dataset-level latent-space bias.

* **[Compared to Centralized Version of FeDaL] (oKpr):** We further compare FeDaL with its centralized counterpart. Across federated representation learning, forecasting, imputation, anomaly detection, and classification **(Tables RR2–RR8 for oKpr)**, FeDaL consistently outperforms the centralized model despite federated constraints. Scaling-behavior studies **(Tables RR9–RR12 for oKpr)** likewise show reliable scaling across dataset sizes and strong performance relative to the centralized version.

* **[Privacy Metrics] (oKpr):** We report MMD distances in ***Table RR1 for oKpr***. Raw client features show the highest cross-client MMD, confirming strong dataset-level heterogeneity. The optimized core-set greatly reduces this gap, while the perturbed core-set introduces only a slight increase, preserving alignment-relevant structure while ensuring privacy.

These validate the soundness of our motivation and design.

---

# Summary of Responses to Reviewers' Questions

We also clarify some questions the reviewers mentioned, including but not limited to:

* **[Novelty]:** FeDaL’s contribution lies not in introducing new operators, but in re-purposing and coordinating them to solve a previously unaddressed problem: training a general-purpose TSFM from scratch under dataset-level heterogeneity in a federated setting. Existing FL and TSFM methods neither model dataset-induced biases nor achieve unified cross-domain generalization without domain-specific fine-tuning.

* **[Problem Formulation]:** Our setting is federated pretraining of a general TSFM from scratch, designed to support both generative and discriminative downstream tasks.

* **[Comparsion with Time-LLM/PeFAD]:** Methods such as Time-FFM and PeFAD fine-tune pretrained LLMs for a single task and assume centralized LLM pretraining. They do not perform federated representation learning. Therefore, they are not suitable baselines for Table 1, where all models are trained from scratch under decentralized data partitions.

* **[Why Centralized TSFM Are Not in Table 1]:** Table 1 evaluates federated representation learning from scratch. Centralized TSFMs rely on massive pooled datasets and a fundamentally different training regime, so they are not comparable in this setting. Their comparison appears in downstream tasks (Table 4), where FeDaL achieves competitive zero-shot performance despite federated training and a much smaller model size.

---

We deeply appreciate the reviewers for their insightful feedback and constructive suggestions, which have substantially improved the quality of our work. We hope that our responses have fully addressed these concerns.

---

### Meta-Review · Area_Chair_2L9X · 2025-12-29

**Summary:**

Reviewers mainly raised concerns about theoretical contribution and methodological novelty, arguing that FeDaL combines existing FL and time-series primitives rather than introducing new algorithms. There were also questions about fairness of comparisons, especially the absence of centralized TSFM and LLM-based baselines in federated pretraining tables, as well as clarity around what is genuinely plug-and-play. Additional concerns included stability of the GBE mechanism, lack of quantitative bias and privacy metrics, and variance reporting. Despite these, reviewers broadly agreed the problem is important and the empirical results are strong.

**Reviewer Concerns:**

The rebuttal convincingly addressed empirical and methodological concerns: added strong heterogeneity tests, variance reporting, sensitivity analyses, privacy quantification, etc. Clarifications on plug-and-play, training-from-scratch, and why certain baselines are incomparable were largely satisfactory. The main remaining concern is conceptual: novelty still hinges on problem formulation and coordinated reuse of components, rather than a fundamentally new algorithm. This is now clearer, but some reviewers may still find it incremental.

AC's additional comment: The design of DBE itself is fine. However, the paper is overall written in an average manner and contains some errors. The core-set tuning is designed in a rather forced way, and it is not convincing to argue from a data-volume perspective that it does not violate privacy.

**Reviewer Scores:**

RwPH: Likely a small upward shift. Empirical bias metrics, robustness studies, and cost analysis directly addressed their concerns.

pV6m (suspected as AI-generated): Likely slightly higher. Most issues were about clarity and stability, which were clarified.

oKpr: Possibly from reject to weak reject. Centralized FeDaL comparisons and privacy metrics help, but novelty concerns likely remain.

CA7V: Effectively resolved all concerns. It would likely remain weak accept or slightly stronger accept.

---

### Decision · Program_Chairs · 2026-01-26

Accept (Poster)